# The nuclear periphery confers repression on H3K9me2-marked genes and transposons to shape cell fate

Harold C. Marin[1], Charlie Allen[1,5], Eric Simental[2,3,5], Eric W. Martin[1,2], Barbara Panning[3], Bassem Al-Sady[2] & Abigail Buchwalter [1,4] ✉

Heterochromatic loci marked by histone H3 lysine 9 dimethylation (H3K9me2) are enriched at the nuclear periphery in metazoans, but the effect of spatial position on heterochromatin function has not been defined. Here we remove three nuclear lamins and the lamin B receptor (LBR) in mouse embryonic stem cells and show that heterochromatin detaches from the nuclear periphery. Mutant mouse embryonic stem cells sustain naive pluripotency and maintain H3K9me2 across the genome but cannot repress H3K9me2-marked genes or transposons. Further, mutant cells fail to differentiate into epiblast-like cells, a transition that requires the expansion of H3K9me2 across the genome. Mutant epiblast-like cells can silence naive pluripotency genes and activate epiblast-stage genes. However, H3K9me2 cannot repress markers of alternative fates, including primitive endoderm. We conclude that the lamins and LBR control the spatial position, dynamic remodelling and repressive capacity of H3K9me2-marked heterochromatin to shape cell fate decisions.

Eukaryotic genomes are organized into compartments of transcriptionally active euchromatin and silent heterochromatin. Heterochromatin shapes cell fate decisions by restricting gene expression and safeguards genome integrity by inhibiting transposon activity[1]. Across eukaryotes, heterochromatin is abundant at the nuclear periphery while euchromatin resides in the nuclear interior[2–5]. In mammals, the peripheral position of heterochromatin is established in embryonic nuclei shortly after fertilization and precedes other features of genome folding such as topologically associating domains[6]. The nuclear periphery has an intrinsic and conserved capacity to repress transcription. In *Schizosaccharomyces pombe*, the nuclear periphery enables the long-term epigenetic memory of cell state by antagonizing local nucleosome turnover[7]. In mammals, artificial tethering of loci to the nuclear periphery induces silencing[8–10]. Lineage-irrelevant genes are recruited to the nuclear periphery during differentiation

processes[11,12] and disease-linked mutations to the lamin A/C protein weaken both peripheral positioning and repression of such genes[13,14]. These observations have led to the prevailing model that peripheral heterochromatin positioning promotes the establishment of cell fate by repressing alternative fate genes. However, the mechanism by which the nuclear periphery confers repression on associated chromatin remains unknown.

Histone H3 lysine 9 di- and/or trimethylation (H3K9me2/3) is enriched on peripheral heterochromatin[7,15–19] and is required for the formation of this compartment[17,18,20]. H3K9me2 is highly enriched in lamina-associated domains (LADs) of chromatin in mammals[16,18,21–23]. H3K9me2 is predominantly deposited by the heterodimeric G9a/GLP enzyme (also known as EHMT2/EHMT1)[24,25]. H3K9me2 promotes the repression of both genes and transposons[24,26–30] and is essential for mammalian pre-implantation development[24,25,29]. However,

[1]Cardiovascular Research Institute, University of California, San Francisco, San Francisco, CA, USA. [2]Department of Microbiology and Immunology, University of California, San Francisco, San Francisco, CA, USA. [3]Department of Biochemistry, University of California, San Francisco, San Francisco, CA, USA. [4]Department of Physiology, University of California, San Francisco, San Francisco, CA, USA. [5]These authors contributed equally: Charlie Allen, Eric Simental. ✉e-mail: abigail.buchwalter@ucsf.edu

H3K9me2-modified chromatin is permissive to transcription in some contexts[31,32], implying that H3K9me2 alone is not sufficient to induce repression.

Receptor proteins that tether H3K9me2/3-marked loci to the nuclear periphery vary across eukaryotes and include components of the inner nuclear membrane (INM) and the nuclear lamina, a meshwork formed by lamin proteins that underlies the INM. For instance, the INM protein Amo1 tethers H3K9me3 to the nuclear periphery in *S. pombe*[7], while the INM protein CEC-4 performs this function in *Caenorhabditis elegans* embryos[15,33]. In mammals, lamin A/C and the nuclear membrane protein lamin B receptor (LBR) each contribute to heterochromatin positioning[34], although it is unknown whether they directly recognize H3K9me2/3. Lamin-bound proteins such as LAP2β, HDAC3, PRR14 and others are candidates that could promote lamin-mediated heterochromatin tethering[9,16,35–37]; separately, direct interactions between lamin A/C and histones have also been reported[38,39]. LBR binds to the H3K9me2/3-binding protein HP1 (ref. [40]) and can also interact with histone tails via its Tudor domain[41,42].

Here, we disrupt the lamins and LBR in mouse embryonic stem (mES) cells to dissect the functions of heterochromatin positioning during early mammalian development. We show that these proteins control the peripheral position of heterochromatin and discover that H3K9me2 is unable to repress either transposons or lineage-specific genes when displaced from the nuclear periphery, indicating that heterochromatin positioning enhances H3K9me2-mediated repression. Finally, we show that displacing heterochromatin from the nuclear periphery impairs the timely restriction of gene expression that shapes cell fate during differentiation. Altogether, this work reveals how the nuclear periphery sculpts the function of heterochromatin.

## Results

### The lamins and LBR recruit heterochromatin to the nuclear periphery in naive pluripotency

H3K9me2-marked chromatin strongly correlates with LADs across mammalian cell types[16,23]. In wild-type (WT) mES cells, H3K9me2 is enriched underneath the nuclear lamina (Fig. 1a and Extended Data Fig. 1a) while H3K27me3 and H3K9me3 are not (Extended Data Fig. 1c,d). Despite the lamina's anticipated role in genome organization, lamin triple knockout (TKO) mES cells lacking all lamin isoforms are viable, pluripotent and exhibit modest changes to peripheral heterochromatin positioning, genome folding and gene expression[43–45]. Both WT and TKO mES cells express high levels of LBR, which remains enriched at the nuclear periphery even in the absence of the lamins (Supplementary Fig. 1b). H3K9me2-marked chromatin also remains enriched at the nuclear periphery in TKO mES cells (Fig. 1a,b). These observations led us to hypothesize that LBR sustains heterochromatin organization in this context[34], and to propose that lamin TKO mES cells are a useful sensitized system in which to dissect the functions of heterochromatin positioning in mammals.

We generated a single lamin + LBR quadruple knockout (QKO) clone and several LBR knockout (LBR KO) clones by introducing frameshift indels into the *Lbr* locus in TKO mES cells and WT mES cells, respectively (Supplementary Fig. 1a,c–f). The survival of only one QKO clone implies that disrupting both the lamins and LBR severely impairs viability. LBR KO, TKO and QKO mES cells are each viable and express core pluripotency factors, albeit with a modest decrease in proliferation rate in QKO mES cells (Supplementary Fig. 1g–i).

We assessed the impact of lamin and LBR ablation on heterochromatin organization by quantifying the radial distribution of H3K9me2 (Fig. 1a–e,h and Supplementary Fig. 2) and compacted DNA (Extended Data Fig. 2a–e), as well as by transmission electron microscopy (TEM) (Fig. 1i–m and Extended Data Fig. 2f–i). H3K9me2-marked chromatin is displaced from the nuclear periphery and forms nucleoplasmic foci in QKO mES cells (Fig. 1d,h), but not in TKO (Fig. 1b,h) or LBR KO mES cells (Fig. 1c,h). However, the total abundance of H3K9me2

remains unchanged (Fig. 1e), implying that deposition is independent of peripheral positioning. While compacted DNA is detectable at the nuclear periphery in WT, LBR KO and TKO mES cells, it shifts into the nucleoplasm in QKO mES cells (Extended Data Fig. 2d,e). These observations were confirmed by acute CRISPR–Cas9 targeting of LBR in TKOs (Extended Data Fig. 1e–h). TEM reveals an electron-dense layer of heterochromatin underneath the nuclear envelope in WT, TKO and LBR KO mES cells (Fig. 1i–k,m), but not in QKO mES cells (Fig. 1l,m), where it is instead more prominent in the nucleoplasm (Extended Data Fig. 2i). These data indicate that the lamins and LBR control the spatial position of heterochromatin, consistent with previous work[30].

To understand the time scale of heterochromatin displacement from the nuclear periphery in the absence of the lamins and LBR, we used tetracycline-inducible RNAi[46] to deplete LBR in TKO mES cells (Extended Data Fig. 3f). Loss of LBR and disorganization of H3K9me2-marked chromatin each occurred within 48 h (Extended Data Fig. 3g–j). H3K9me2 positioning could be re-established within 48 h by expressing exogenous LBR in QKO mES cells (Fig. 1g,h), but not by expressing the nucleoplasmic domain or transmembrane domain of LBR alone (Extended Data Fig. 3c–e), indicating that the displacement of H3K9me2-marked chromatin from the nuclear periphery of lamin-depleted cells depends on LBR and is reversible.

### LAP2β and H3K9me2 delineate the lamina-associated genome in naive pluripotency

We next sought a means to evaluate the effects of lamin and LBR ablation on genome contacts with the nuclear periphery. While LADs are typically defined by detection of lamin:genome contacts using anti-lamin antibodies, we required a fiducial marker of the nuclear periphery that remains intact even without the lamins and LBR. We chose the INM protein LAP2β, which localizes to the INM in all genotypes (Fig. 1b–d). The *TMPO* gene encodes both the INM-localized LAP2β and nucleoplasmic LAP2α splice isoforms (Supplementary Fig. 3a,b); the LAP2β-encoding transcript is more highly expressed than the LAP2α-encoding transcript in mES cells (Supplementary Fig. 3c). We identified an anti-LAP2 antibody (Invitrogen, PA5-52519) targeting residues 129-269 of LAP2α/β, where the C-terminal ~70 amino acids are unique to LAP2β (Supplementary Fig. 3b). This antibody stains the INM in mES cells (Fig. 1b–d) and detects a single band corresponding to the molecular weight of LAP2β by immunoblotting (Supplementary Fig. 3d,e). We conclude that the anti-LAP2 antibody preferentially recognizes INM-localized LAP2β.

To test the ability of the LAP2 antibody to detect LADs, we performed cleavage under targets and release using nuclease (CUT&RUN) for both lamin B1 and LAP2β with a spike-in control[47] in WT mES cells (Fig. 2a, Supplementary Fig. 4 and Extended Data Fig. 4). We applied a two-state hidden Markov model (HMM)[48] to identify regions with significant enrichment of lamin B1 or LAP2β association (Fig. 2a, Supplementary Fig. 4, Extended Data Fig. 4 and Supplementary Table 1), which indicated that lamin B1 and LAP2β contact nearly identical kilobase- to megabase-long domains (~95% overlap; Extended Data Fig. 4f–h).

A strong correlation between LADs and H3K9me2/3-marked heterochromatin has been reported in various mammalian cell types[16,23]. To evaluate this correlation in mES cells, we used H3K9me2/3 antibodies with validated selectivity in genome-binding assays[16,49,50] to perform spike-in-controlled CUT&RUN followed by HMM domain calling (Fig. 2a, Methods and Supplementary Table 1). H3K9me2 forms large domains that overlap extensively with LB1 and LAP2β-bound LADs (Fig. 2a,c). H3K9me3, in contrast, is restricted to more focal sites and covers much less of the genome (Fig. 2a,b). While H3K9me3 overlaps partially with LADs, many LADs lack H3K9me3 (Fig. 2a,b), indicating that LADs are preferentially modified by H3K9me2 in naive pluripotency. This outcome aligns with a previous report in naive mES cells[16], and our LAP2β and H3K9me2 domains each overlap extensively with published datasets[16] (Extended Data Figs. 4f and 5g). LAP2β and

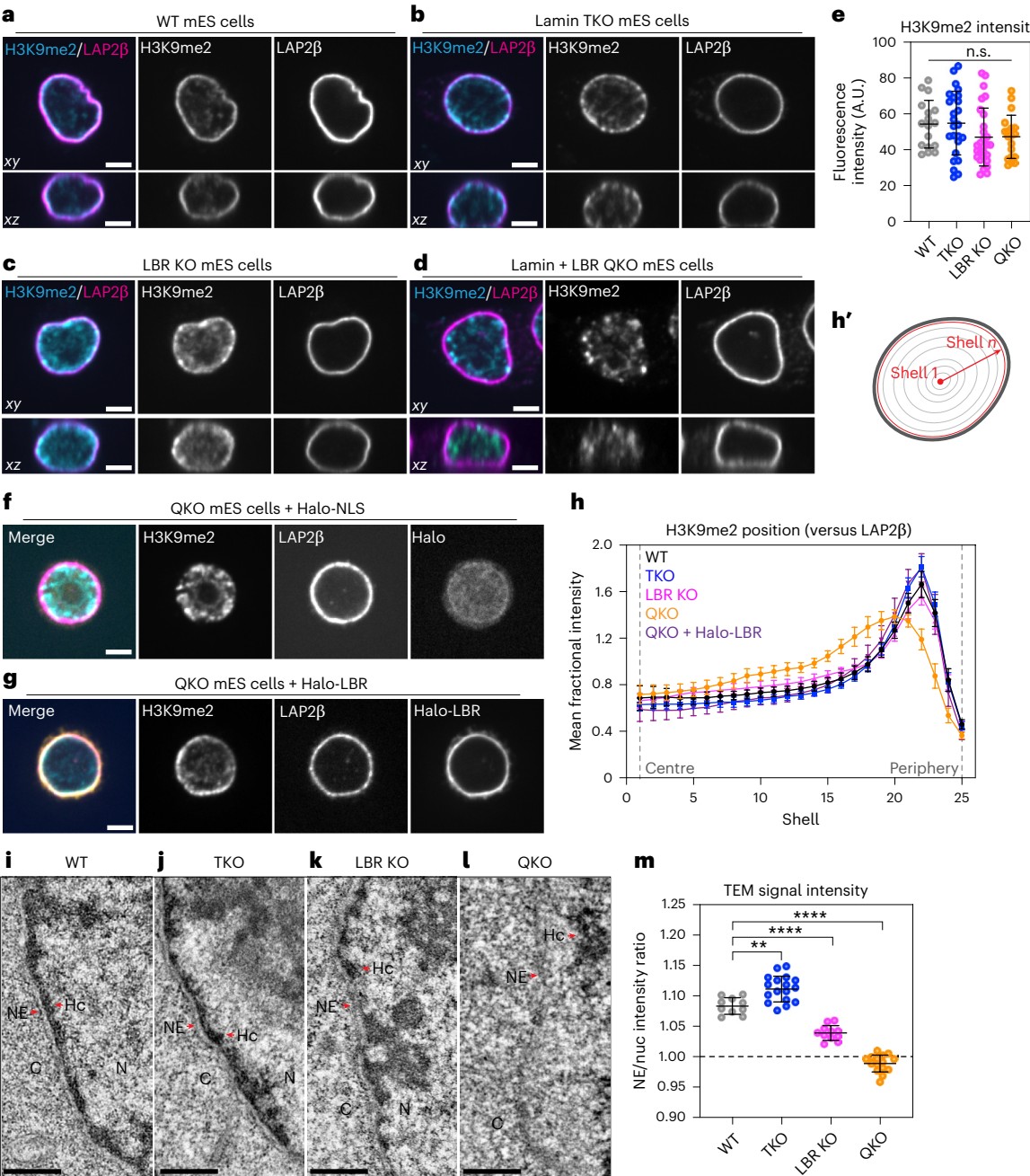

**Fig. 1 | The lamins and LBR localize H3K9me2-marked chromatin to the nuclear periphery in mES cells. a–d,** Immunofluorescence of H3K9me2 localization compared with the INM protein LAP2β in WT (**a**), lamin TKO (**b**), LBR KO (**c**) and lamin + LBR QKO (**d**) mES cells. A central *z* slice (*xy*) and a central *y* slice (*xz*) are shown. Scale bar, 5 μm. **e,** H3K9me2 total fluorescence intensity per nucleus in WT (*n* = 17), TKO (*n* = 25), LBR KO (*n* = 26) and QKO mES cells (*n* = 18), *P* = 0.1629 (n.s.) by one-way analysis of variance (ANOVA). The bars indicate mean and s.d. **f,g,** The immunofluorescence of H3K9me2 localization compared with the INM protein (LAP2β) in lamin + LBR QKO mES cells expressing Halo-NLS (**f**) or Halo-LBR (**g**) for 48 h. **h,** Analysis of H3K9me2 intensity as a function of radial nuclear position (as diagrammed in **h′**) in WT, TKO, LBR KO and QKO mES cells (*n* = 25 nuclei per condition) and in QKO mES cells expressing Halo-LBR (n = 16 nuclei). WT versus QKO, shells 9–19, *P* < 0.0001; WT versus QKO, shell 20, 0.0116; WT versus QKO, shell 21, 0.0024; WT versus QKO, shells 22–24, <0.0001; WT versus TKO, shell 21, 0.0405; WT versus TKO, shell 22, 0.0297; WT versus LBR KO, shell 14, 0.0201; WT versus LBR KO, shell 15, 0.0203; WT versus LBR KO, shell 16, 0.0332. All other comparisons n.s.; unpaired *t*-test (two-tailed). The points indicate the mean and error bars indicate 95% confidence intervals. See Extended Data Fig. 3 for the full Halo-LBR rescue analysis and statistics. **i–l,** Transmission electron microscopy showing a 1.3 μm by 2.6 μm section of the nuclear periphery in WT (**i**), lamin TKO (**j**), LBR KO (**k**) and lamin + LBR QKO (**l**) mES cells. C, cytoplasm; HC, heterochromatin; N, nucleus; NE, nuclear envelope. Scale bar, 0.5 μm. **m,** Quantification of relative TEM signal intensity ratio at the NE versus nucleoplasm (nuc), for WT (*n* = 10), LBR KO (*n* = 11), TKO (*n* = 17) and QKO (*n* = 15) mES cells. ****$P_{adj}$ < 0.0001 for WT versus LBR KO and WT versus QKO and **$P_{adj}$ = 0.001 for WT versus TKO by one-way ANOVA followed by Dunnett's T3 multiple comparisons test. The bars indicate mean and s.d.

LB1-bound, H3K9me2-modified LADs cover 1,081 Mb, or approximately 40%, of the genome (Fig. 2c,d). We also identified less numerous H3K9me2-only domains (KODs)[32] (439 Mb) and domains occupied by lamin B1 and LAP2β that lack H3K9me2 (167.6 Mb) (Fig. 2c).

Gene association with the nuclear lamina is correlated with transcriptional repression[8,9,11,52]. We found strong correlations between H3K9me2 modification, LAD residence and repression (Fig. 2e). LADs are more repressive than KODs, which are more repressive than

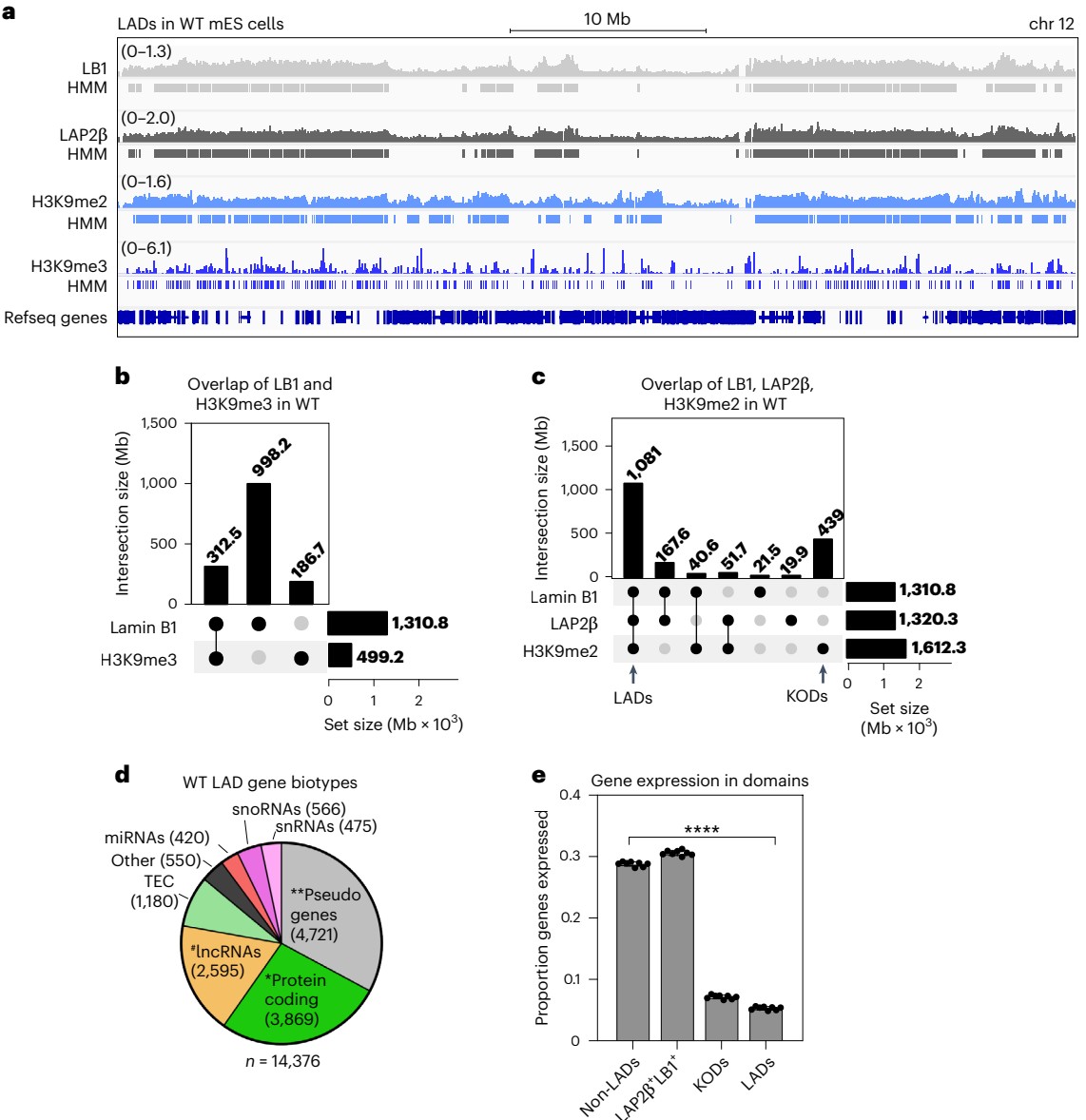

**Fig. 2 | The lamina-associated genome is modified with H3K9me2 in naive pluripotency. a**, Averaged genome tracks and HMM domain calls from spike-in controlled lamin B1, LAP2β, H3K9me2 and H3K9me3 CUT&RUN in WT mES cells (*n* = 3 replicates per condition except for H3K9me3, for which *n* = 2 replicates were performed), highlighting a 50 Mb section of chromosome 12 (chr 12). **b**, An UpSet plot showing the overlap between lamin B1 and H3K9me3 on 312.5 Mb of the genome in WT mES cells. H3K9me3 domains separately cover 186.7 Mb, while lamin B1 domains separately cover 998.2 Mb. **c**, An UpSet plot showing the extensive overlap of lamin B1, LAP2β and H3K9me2 in WT mES cells in domains covering 1081 Mb of the genome ('LADs'), H3K9me2-only domains covering 439 Mb of the genome ('KODs') and lamin B1 and LAP2β-occupied

domains covering 167.6 Mb of the genome. **d**, Analysis of LAD-resident genes, defined as being at least 90% covered by a LAD, in WT mES cells (14,376 genes in total). #long non-coding (lnc)RNAs that are significantly depleted in LADs compared with the mouse genome (*P* < 0.0001, two-sided chi-squared test). *Protein-coding genes and **pseudogenes significantly enriched in mES cell LADs (*P* = 0.0334 and *P* < 0.0001, respectively, two-sided chi-square test). **e**, The proportion of genes expressed (TPM >5) in non-LADs (*n* = 28,410 genes), LAP2β⁺LB1⁺ domains (*n* = 5,020 genes), KODs (*n* = 9,224 genes) and LADs (*n* = 14,376 genes) in WT mES cells. ****all means are significantly different (*P* < 0.0001) by one-way ANOVA. TEC, to be experimentally confirmed; snoRNA, small nucleolar RNA; snRNA, small nuclear RNA.

non-LADs. Curiously, the small group of lamin B1 and LAP2β−bound domains that lack H3K9me2 are highly expressed in mES cells (Fig. 2e).

## Global displacement of chromatin from the nuclear periphery upon loss of lamins and LBR

The strong correlation between LAP2β and lamin B1 genome contacts (Fig. 2a,c) implies that LAP2β can faithfully report the consequences of lamin and LBR ablation on chromatin positioning. We applied spike-in controlled LAP2β CUT&RUN in TKO, LBR KO and QKO mES cells (Fig. 3a–d). Notably, the ratio of total CUT&RUN-released DNA to spike-in control

DNA decreases significantly in both TKO and QKO mES cells (scale factor, Fig. 3b), indicating a global reduction in LAP2β:genome contacts. Consistently, inspection of WT LADs reveals progressive decreases in LAP2β:genome contact frequency when LBR, the lamins or both are depleted (Fig. 3c and Extended Data Fig. 6a). The progressive erosion of a LAD on chromosome 12 (Fig. 3a) reflects this genome-wide trend.

While WT LADs are diminished, new LAP2β:genome contacts emerge in mutant mES cells. This inversion is most notable in LBR KOs, where new domains form with contact frequencies comparable to WT LADs (Fig. 3a,d). These new domains correlate poorly with H3K9me2

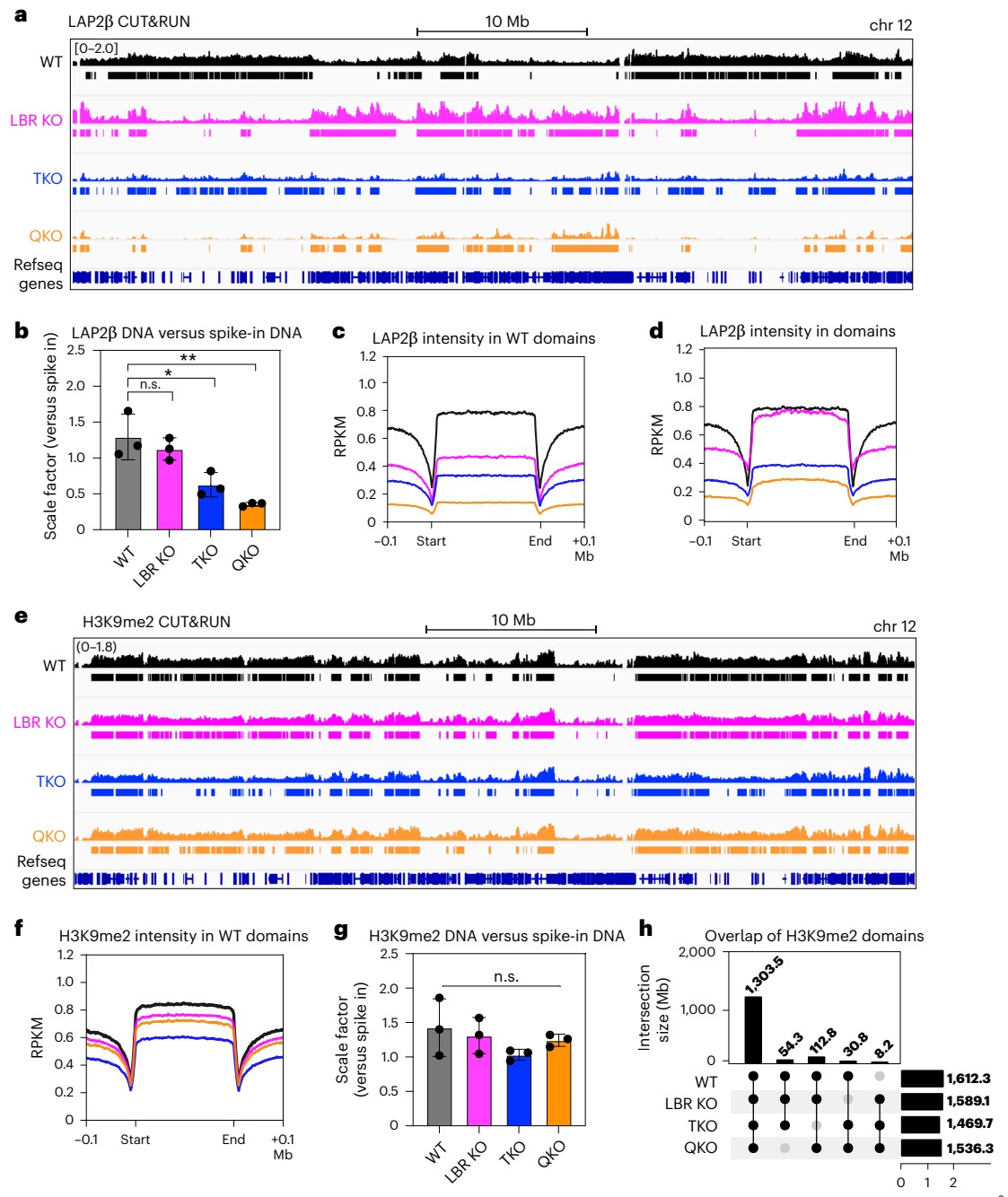

**Fig. 3 | The lamins and LBR control the spatial position but not the genomic distribution of H3K9me2-modified chromatin. a**, Averaged genome tracks and HMM domain calls from spike-in controlled LAP2β CUT&RUN in WT, LBR KO, TKO and QKO mES cells, highlighting a 50 Mb section of chromosome 12 ($n = 3$ replicates per condition). The scale shown applies to all conditions. **b**, The quantitative yield of LAP2β CUT&RUN DNA compared to *S. cerevisiae* spike-in DNA in WT, LBR KO, TKO and QKO mES cells. n.s. indicates $P_{adj} = 0.7291$ for WT versus LBR KO, *$P_{adj} = 0.0139$ for WT versus TKO and **$P_{adj} = 0.0017$ for WT versus QKO by one-way ANOVA followed by Tukey's multiple comparisons test. The bars indicate mean and s.d. **c**, Averaged spike-in controlled read density for LAP2β within domains identified in WT mES cells, compared across WT, LBR KO, TKO and QKO CUT&RUN samples. **d**, Averaged spike-in controlled read density for

LAP2β within domains identified within each genotype. **e**, Averaged genome tracks and HMM domain calls from spike-in controlled H3K9me2 CUT&RUN in WT, LBR KO, TKO and QKO mES cells over a 50 Mb section of chromosome 12 ($n = 3$ replicates per condition). **f**, Averaged spike-in controlled read density for H3K9me2 within domains identified in WT mES cells, compared across WT, LBR KO, TKO and QKO CUT&RUN samples. **g**, The quantitative yield of H3K9me2 CUT&RUN DNA compared with *S. cerevisiae* spike-in DNA in WT, LBR KO, TKO and QKO mES cells ($n = 3$ replicates per condition). n.s. indicates no significant differences between means by one-way ANOVA ($P = 0.3433$). The bars indicate mean and s.d. **h**, An UpSet plot showing extensive overlap of H3K9me2 domains between WT, LBR KO, TKO and QKO mES cells.

(compare Fig. 3a versus Fig. 3e), suggesting that their chromatin features are distinct from WT LADs. New domains are also detected in TKO and QKO mES cells, but at far lower contact frequencies (Fig. 3a,d). These data suggest that lamina-mediated tethering may enable LAD

formation in the absence of LBR, while LBR partially sustains LADs in the absence of the lamins. Without both the lamins and LBR, chromatin positioning becomes more random. Altogether, we conclude that the lamins and LBR control the peripheral positioning of chromatin in mES cells.

## The genomic distribution of H3K9me2 is preserved in the absence of the lamins and LBR

We next asked whether chromatin displacement impacts the genomic position and/or abundance of H3K9me2. H3K9me2 domains remain highly consistent across all genotypes (Fig. 3e–g and Extended Data Fig. 5h,i). Overall, 81% of WT H3K9me2 domains are maintained in all four genotypes (Fig. 3h, first column). Ablating the lamins and LBR thus has little to no effect on H3K9me2 deposition, despite the visible displacement of H3K9me2-marked chromatin from the nuclear periphery (Fig. 1d,h) and the profound disruption of LADs (Fig. 3a–d).

## Spatial displacement dysregulates both LAD and non-LAD genes

Transcriptional activation often correlates with gene displacement from the nuclear periphery[12,16,53,54]. To test whether heterochromatin displacement weakens repression, we profiled the transcriptome by total RNA sequencing (RNA-seq). Removing either LBR or the lamins dysregulated modest numbers of nonoverlapping genes (Fig. 4a,b). In contrast, removing both the lamins and LBR synergistically disrupted gene expression, with nearly twice as many genes upregulated as downregulated (2,904 genes upregulated >2-fold versus 1,595 genes downregulated >2-fold; Fig. 4a,b, Extended Data Fig. 6b and Supplementary Table 2). We applied gene set enrichment analysis (GSEA)[55] to evaluate the relationships between dysregulated genes (Fig. 4c). No significant Gene Ontology (GO) terms were identified in LBR KO mES cells (Supplementary Table 3). In TKO and QKO mES cells, development, morphogenesis and proliferation-related GO terms were enriched among downregulated genes, suggesting moderately reduced pluripotency (Fig. 4c, Extended Data Fig. 6c and Supplementary Table 3). Interestingly, very few GO terms were enriched in upregulated genes, even though they are more numerous than downregulated genes (Supplementary Table 3). We infer that genes upregulated by depleting the lamins and/or LBR are not clustered within gene regulatory networks.

We next investigated whether a correlation exists between LAD disruption and derepression. LAP2β contact frequency of WT LADs decreases progressively with the loss of LBR, the lamins or both (Fig. 3c and Extended Data Fig. 6a). This displacement is accompanied by modest up- and downregulation of LAD genes in LBR KO and TKO mES cells (Extended Data Fig. 6d,e). In QKO mES cells, in contrast, nearly twice as many LAD genes are upregulated as downregulated, indicating weakened repression (489 genes >2-fold up and 286 genes >2-fold down; Fig. 4d). However, the unchanged expression of many LAD genes (Fig. 4d) implies that displacement is not sufficient to induce transcriptional activation. derepressed LAD genes are moderately H3K9me2 modified and exhibit intermediate LAP2β contact frequency in WT mES cells, while LAD genes that are further repressed in QKOs are densely H3K9me2 modified and exhibit high LAP2β contact frequency (Fig. 4e–h). Displacement from the nuclear periphery thus has divergent effects on genes with low versus high levels of lamina contact and H3K9me2 modification. Overall, differentially expressed genes (DEGs) are distributed across LADs, KODs and non-LADs (Fig. 4i).

To assess the initial consequences of heterochromatin displacement on gene expression, we analysed the transcriptomes of TKO mES cells after acute depletion of LBR. Few genes were dysregulated at day 2 (Extended Data Fig. 6f and Supplementary Table 4), despite apparent H3K9me2 disorganization (Extended Data Fig. 3f–i). However, after 4 days of LBR depletion, 102 genes were upregulated more than twofold, while only 21 genes were downregulated more than twofold (one of which was *Lbr)* (Extended Data Fig. 6g,h and Supplementary Table 4), indicating a strong bias towards derepression. These upregulated genes overlap with those upregulated in QKO mES cells (Extended Data Fig. 6i,j) and are significantly enriched for H3K9me2 and LAP2β association (Extended Data Fig. 6k,l). Altogether, we conclude that removing the lamins and LBR displaces LAD genes from the nuclear periphery (Fig. 3a–d) and increases permissiveness to transcription (Fig. 4d). As acute disruption of LBR in TKO mES cells

induces far more upregulation than downregulation (Extended Data Fig. 6f–g), we speculate that transcriptional dysregulation extends beyond LAD gene upregulation (Fig. 4i) as a secondary consequence of heterochromatin disorganization.

## Heterochromatin displacement causes widespread derepression of transposons

Transposable elements (TEs) reside within LADs in various mammalian cell types[23,56], but the role of LADs in TE repression is unclear. H3K9me2 deposition by G9a/GLP in mES cells[19,20] represses TEs, including long interspersed elements (LINE1s) and endogenous retroviruses flanked by long terminal repeats (ERV LTRs)[26–30]. We identified several LINE1 and ERV LTR families that are strongly LAP2β associated and densely H3K9me2 modified in WT mES cells (Fig. 5a, Extended Data Fig. 7a,b and Supplementary Table 5). We investigated TE expression in the absence of the lamins and/or LBR by using TEtranscripts[57] to analyse multimapping RNA-seq reads originating from ~1,200 distinct TE families. LBR ablation had a minimal effect on TE expression (seven TE families upregulated twofold; Fig. 5b and Extended Data Fig. 7c), while lamin ablation induced the expression of 27 TE families by at least twofold (Fig. 5b, Extended Data Fig. 7d and Supplementary Table 6). TEs that are derepressed by lamin disruption alone include the LAD-resident, H3K9me2-modified LINE1s L1MdA_I and L1MdA_II, as well as other LINE1s and ERV LTRs (Extended Data Fig. 7d). Co-depletion of the lamins and LBR dramatically elevated TE expression, with 338 TE families upregulated at least twofold (Fig. 5b,c and Supplementary Table 6). Several ERV-K LTRs are upregulated in QKOs (ERVB4_2-I, ERVB4_3-I, RLTR13D6 and RLTR45-int; Fig. 5c). Notably, ERV-K family members are strongly enriched in WT LADs (Fig. 5a and Extended Data Fig. 7a) and are also derepressed when H3K9me2 is depleted via G9a/GLP inactivation in mES cells[29,30].

Each TE family exists in numerous identical copies across the genome. The expression of individual copies can be evaluated by mapping transcripts that contain unique flanking genomic sequences. We used this approach to evaluate the L1MdA_I LINE1 family, an actively retrotransposing and LAD-resident TE[58] (Fig. 5a). While L1MdA_Is are lowly transcribed in WT and LBR KOs (Extended Data Fig. 7c), 550 and 684 copies are upregulated at least tenfold in TKOs and QKOs, respectively (Fig. 5d and Extended Data Fig. 7g). The levels of the L1-encoded Orf1p protein are increased in both TKOs and QKOs (Fig. 5g–k), consistent with significantly elevated L1 activity. Inspecting individual L1MdA_I loci indicated that derepressed L1MdA_I copies are strongly associated with LAP2β in WT mES cells, but that deletion of the lamins or both the lamins and LBR displaces these loci from the nuclear periphery (Fig. 5e and Extended Data Fig. 7h). These data indicate that L1MdA_I LINE1s become derepressed by displacement from the nuclear periphery.

To determine how quickly TEs become derepressed when heterochromatin is displaced, we analysed TE expression after RNAi-mediated depletion of LBR in TKO mES cells. Strikingly, 42 uniquely mapped TE copies were potently derepressed greater than fivefold within 4 days of LBR depletion (Extended Data Fig. 7e,f), indicating that TE derepression follows H3K9me2 displacement.

The effects of heterochromatin displacement on TE expression are reminiscent of those of deleting H3K9 methyltransferases, which reduce H3K9me2/3 on TEs and permit transcription[28–30,59,60]. However, our analyses indicate that H3K9me2 is generally maintained on the same genomic loci when displaced from the nuclear periphery (Fig. 3e–h). We compared the extent of H3K9me2 modification on active versus silent L1MdA_I LINE1s and found that (1) L1MdA_I copies that become activated are highly modified by H3K9me2, while unaffected L1MdA_I copies are not (Fig. 5f and Extended Data Fig. 7i) and (2) when activated, these L1MdA_I copies remain modified by H3K9me2 (Fig. 5f and Extended Data Fig. 7i). Therefore, similarly to our analyses of genes (Fig. 4), LAD displacement permits TE transcription despite persistent H3K9me2 modification.

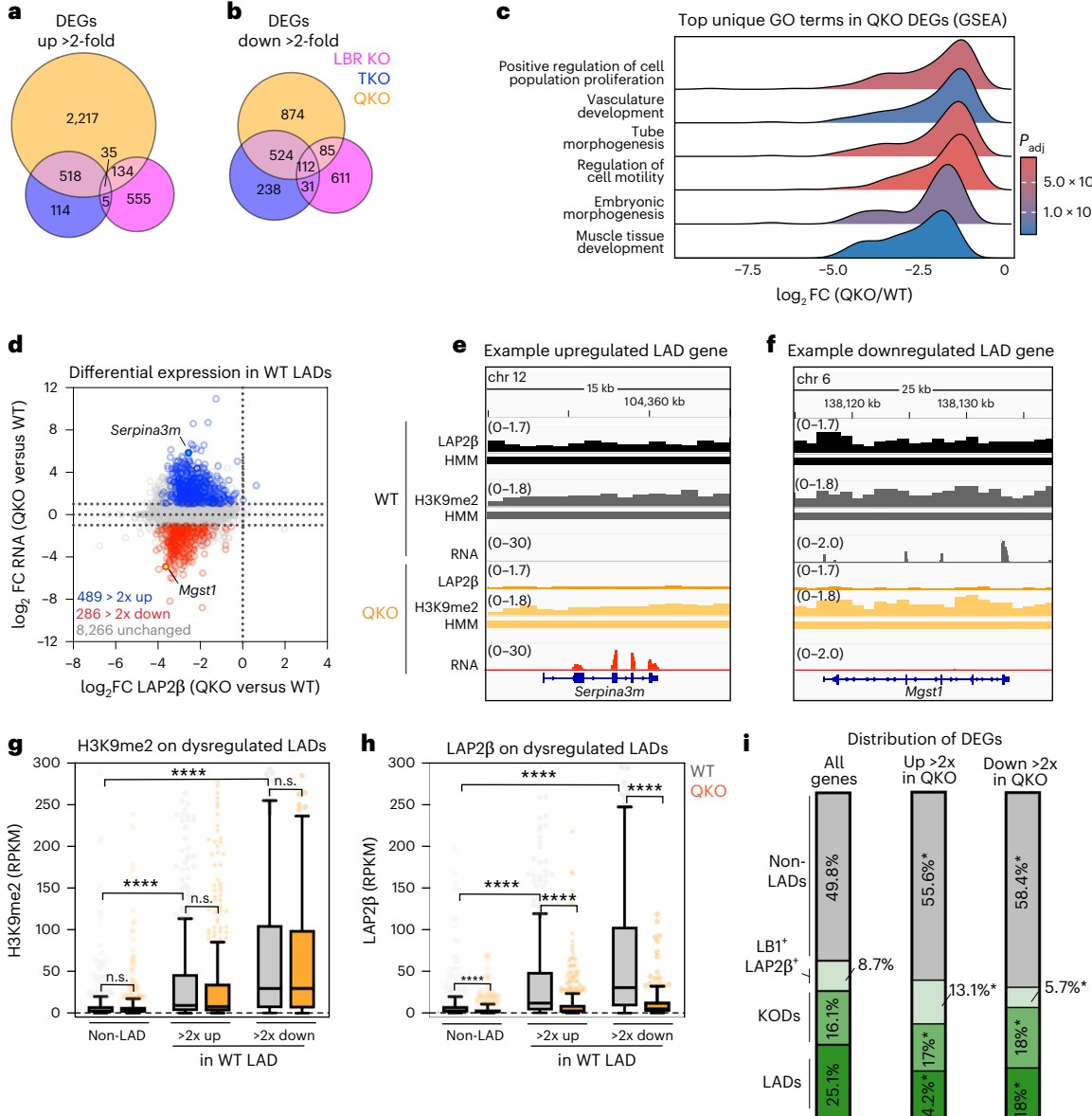

**Fig. 4 | Removal of the lamins and LBR displaces and derepresses LADs and other genes. a,b,** The number of genes significantly upregulated (**a**) and downregulated (**b**) at least twofold in LBR KO, TKO and QKO mES cells (*n* = 4 replicates per condition). **c**, The identification of top unique GO terms in genes differentially expressed between QKO and WT mES cells (GO terms identified by GSEA and reduced by ReVIGO). See also Supplementary Table 3 for the full list of GO terms. **d**, Analysis of gene expression changes within WT LADs: a comparison of fold change (FC) in gene expression with FC in lamina association (LAP2β) in QKO versus WT mES cells. Of this subset of genes, 489 are significantly upregulated >2-fold ($P_{adj}$ < 0.05), 286 are significantly downregulated >2-fold ($P_{adj}$ < 0.05) and 8,266 exhibit no significant change in expression. **e,f,** LAP2β and H3K9me2 levels (spike-in controlled RPKM) and RNA expression levels for representative upregulated WT LAD gene *Serpina3m* (**e**) and downregulated

WT LAD gene *Slamf6* (**f**) in WT and QKO mES cells. **g,h,** H3K9me2 levels (average spike-in controlled RPKM from three replicates) (**g**) and LAP2β levels (average spike-in controlled RPKM from three replicates) (**h**) on 489 WT LAD genes upregulated >2-fold in QKO mES cells (**d**, blue points) and 286 WT LAD genes downregulated >2-fold in QKO mES cells (**d**, red points), compared with a random sample of 489 non-LAD genes in WT and QKO mES cells. ****$P$ < 0.0001 by Kruskal–Wallis multiple comparisons test with Dunn's correction. n.s. indicates $P$ > 0.9999. **i**, Distribution of QKO DEGs across WT domain categories (non-LADs, lamin B1 and LAP2β-associated genes, KODs and LADs). *$P$ < 0.0001 for non-LADs up in QKO versus WT, non-LADs down in QKO versus WT, LB1⁺LAP2β⁺ genes up in QKO, LB1⁺LAP2b⁺ genes down in QKO, LADs up in QKO and LADs down in QKO. *KODs up in QKO ($P$ = 0.0051) and KODs down in QKO ($P$ = 0.0415). Analysis of significant enrichment of DEGs in a domain class by two-sided Fisher's exact test.

## Heterochromatin positioning enables the transition from naive to primed pluripotency

Our data indicate that H3K9me2 loses its capacity to repress transcription of genes (Fig. 4) and TEs (Fig. 5) when displaced from the nuclear periphery. We hypothesize that recruitment of H3K9me2-marked chromatin to the nuclear periphery enhances H3K9me2-mediated repression. To further test the influence of heterochromatin positioning on H3K9me2 function, we tested its necessity for the progression from

naive to primed pluripotency, a developmental transition that relies on G9a/GLP-mediated expansion of H3K9me2 across the genome[29,61].

In the embryo, the transition to primed pluripotency occurs when the inner cell mass begins to form the epiblast (Fig. 6a). In vitro, induction of epiblast-like cells (EpiLCs) can be achieved by treatment of mES cells with FGF and activin A[62,63] (Fig. 6a). LBR KOs and TKOs each exhibited moderately decreased EpiLC survival, while survival of QKO EpiLCs was severely impaired (Fig. 6b). In each genotype, surviving cells

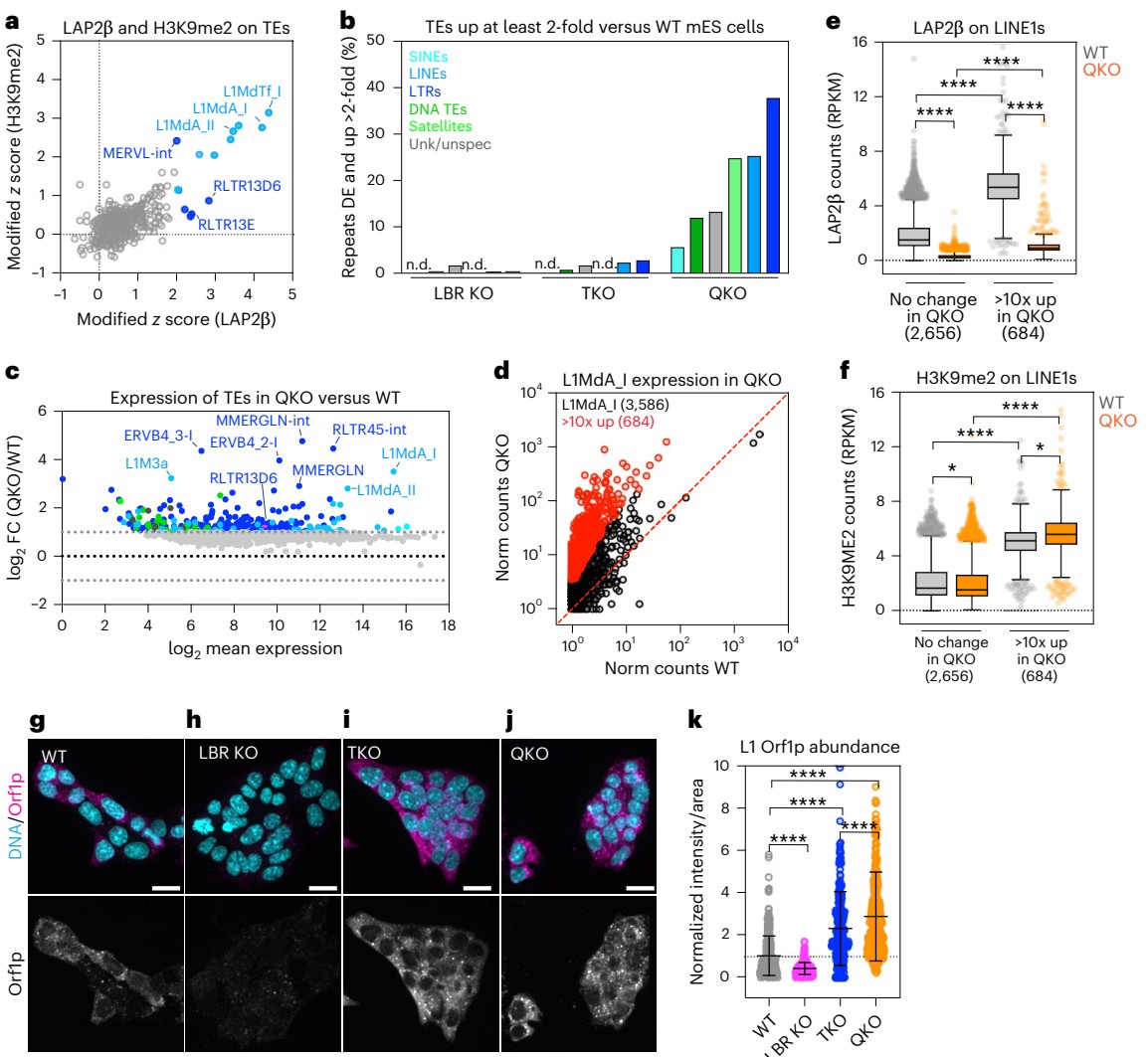

**Fig. 5 | Removal of the lamins and LBR displaces and derepresses H3K9me2-marked transposons. a**, Identification of strongly LAD-associated and/or H3K9me2-modified TE families in WT mES cells. The modified *z* score of spike-in controlled H3K9me2 (*y* axis) versus LAP2β (*x* axis) shown for ~1,200 TE families. TE families with a modified *z* score >2 for H3K9me2 and/or LAP2β signal are colour coded according to class and all others are shown in grey. **b**, The percentage of TE families significantly derepressed (DE) at least twofold ($P_{adj} < 0.05$) in all genotypes compared with WT mES cells, determined by TEtranscripts. n.d., not detected; Unk/unspec, unknown or unspecified TE type. **c**, An MA plot comparing the expression of ~1,200 TEs detected by TEtranscripts in QKO versus WT mES cells. All repeats without a significant change between genotypes are grey and significantly differentially expressed TEs (minimum two FC) are coloured correspondingly to the TE family. **d**, Normalized (norm) RNA counts for 3,586 uniquely mapped L1MdA_I LINE element genomic copies in QKO versus WT mES cells, plotted as $\log_{10}$(average + 1). L1MdA_I copies with >10-FC and significant difference in expression ($P_{adj} < 0.05$) are coloured in red. **e**, The

average spike-in normalized RPKM from uniquely mapped reads of LAP2β (*n* = 3 replicates) on L1MdA_I LINE elements with no significant change to expression versus those upregulated >10-fold in QKO mES cells. ****$P < 0.0001$. **f**, The average spike-in normalized RPKM from uniquely mapped reads of H3K9me2 (*n* = 3 replicates) on L1MdA_I LINE elements with no significant change to expression versus those upregulated >10-fold in QKO mES cells. ****$P < 0.0001$ and *$P = 0.014$ (no change, WT versus QKO); $P = 0.015$ (>10× upregulated in WT versus QKO) by one-way Kruskal–Wallis multiple comparisons test with Dunn's correction. In the box plots (Tukey), the centre line indicates the median, box limits indicate the 25th to 75th percentiles, whiskers indicate 1.5× the interquartile range and points indicate outlier values. **g–j**, Detection of L1 Orf1p expression in WT (**g**), LBR KO (**h**), TKO (**i**) and QKO (**j**) mES cells. Scale bar, 20 μm. **k**, Quantification of relative Orf1p abundance shown as intensity per unit area and normalized to WT. WT (*n* = 356), LBR KO (*n* = 357), TKO (*n* = 270) and QKO (*n* = 379) from three replicate experiments. ****$P < 0.0001$ by one-way ANOVA followed by Tukey's multiple comparisons test. The bars indicate mean and s.d.

express the EpiLC-specific transcription factor Otx2 and downregulate the naive pluripotency factor Nanog (Fig. 6c and Supplementary Fig. 6). Re-introduction of LBR into QKOs rescues EpiLC survival (Fig. 6d). Acute depletion of LBR by RNAi in TKOs impaired EpiLC differentiation, demonstrating that EpiLC loss is an immediate consequence of lamin + LBR co-depletion (Fig. 6e).

In addition to their shared role in heterochromatin positioning, LBR and the lamins have other functions that could contribute to EpiLC survival. We therefore tested the relevance of these other functions. LBR participates in cholesterol biosynthesis[64], but cholesterol levels are

unaffected by LBR ablation (Supplementary Fig. 7a), probably because of the redundant activity of the TM7SF2 enzyme[64]. The effect of deleting LBR on EpiLC survival is thus not a consequence of cholesterol depletion. Differentiation is accompanied by changes in cell morphology (Supplementary Fig. 7b). As the lamins and heterochromatin each absorb forces on the nucleus[65], we tested whether forces induced by cellular spreading impair QKO viability by seeding cells on lower-attachment and higher-attachment substrates (Supplementary Fig. 7c,d). We found no difference in cell viability between these conditions, indicating that cellular flattening itself does not impair survival of QKOs.

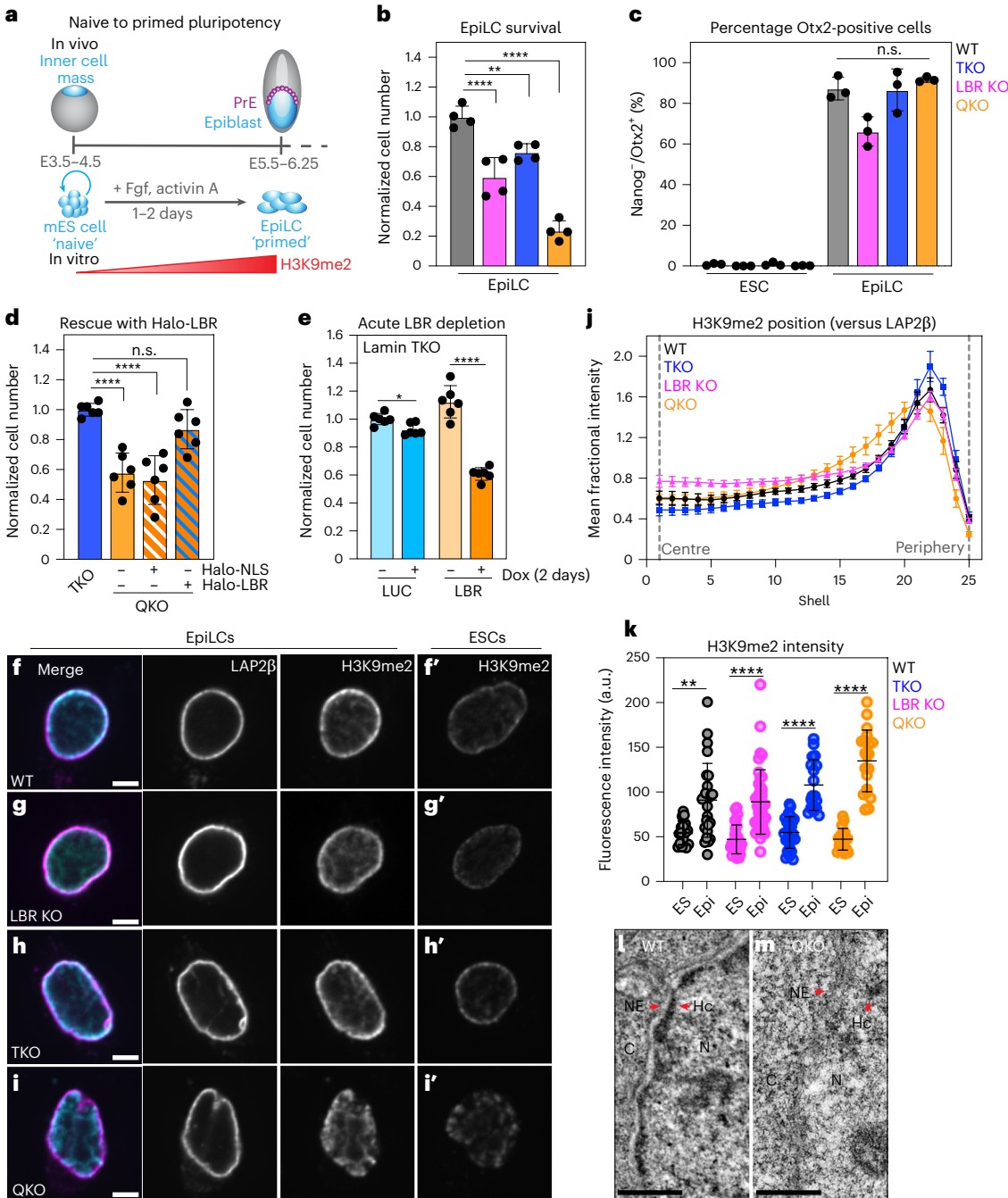

**Fig. 6 | Heterochromatin positioning by the lamins and LBR is essential for EpiLC survival. a**, Diagram of naive (inner cell mass/mES cell) to primed (epiblast/EpiLC) developmental transition in vivo and its approximation in vitro. **b**, Normalized cell numbers after 2 days of culture in EpiLC differentiation conditions (normalized to WT; $n = 4$ replicates per genotype). The columns indicate the mean and error bars indicate s.d. ****$P_{adj} < 0.0001$ for WT versus LBR KO and WT versus QKO; **$P_{adj} = 0.0058$ for WT versus TKO by one-way ANOVA followed by Dunnett's multiple comparisons test. **c**, The percentage Otx2-positive cells after 2 days of culture in EpiLC differentiation conditions ($n = 3$ replicates per genotype). The columns indicate the mean and the error bars indicate s.d. All EpiLC conditions n.s.; WT versus LBR KO EpiLC, $P = 0.0867$; WT versus TKO, $P = 0.9935$; WT versus QKO, $P = 0.5942$. **d**, Normalized cell numbers after 2 days of culture in EpiLC differentiation conditions (normalized to TKO; $n = 6$ replicates per genotype). The columns indicate the mean, error bars indicate s.d. ****$P_{adj} < 0.0001$ for TKO versus QKO and TKO versus QKO + Halo-NLS by one-way ANOVA followed by Dunnett's multiple comparisons test. n.s. indicates $P = 0.6244$. **e**, Normalized cell numbers of lamin TKO mES cells after 2 days of culture in EpiLC differentiation with (+) or without (−) co-induction of LUC or LBR miR-E. The columns indicate

the mean and error bars indicate s.d. $n = 6$ replicates from 2 independent clones shown. *$P = 0.011$ and ****$P < 0.0001$ by unpaired two-tailed $t$-test. Dox, doxycycline. **f–i**, Immunofluorescence of H3K9me2 localization compared with the INM protein LAP2β in WT (**f–f'**), LBR KO (**g–g'**), lamin TKO (**h–h'**) and lamin + LBR QKO (**i–i'**) cells; **f–i** show EpiLC samples while **f'–i'** show mES cell samples stained and imaged in parallel. Central $z$ slices ($xy$) are shown. Scale bar, 5 μm. **j**, Radial intensity analysis of H3K9me2 position versus LAP2β in WT, TKO, LBR KO and QKO EpiLCs. WT versus QKO, shell 12, P = 0.0022; shell 13, P = 0.0002; shells 14–19, $P < 0.0001$; shell 20, 0.0029; shell 22, 0.0097; shell 23, 0.0007; shells 24–25, $P < 0.0001$ by unpaired $t$-test (two tailed). **k**, H3K9me2 total fluorescence intensity per nucleus in WT mES cells ($n = 17$) versus WT EpiLCs ($n = 26$); **$P = 0.0015$. TKO mES cells ($n = 25$) versus TKO EpiLCs ($n = 18$); ****$P < 0.0001$. LBR KO mES cells ($n = 28$) versus LBR KO EpiLCs ($n = 41$); ****$P < 0.0001$. QKO mES cells ($n = 19$) versus QKO EpiLCs ($n = 20$); ****$P < 0.0001$ by one-way ANOVA followed by Dunnett's multiple comparisons test. The bars indicate mean and error bars indicate s.d. **l,m**, TEM images showing a 1.3 μm by 2.6 μm section of the nuclear periphery in WT (**l**) and lamin + LBR QKO (**m**) EpiLCs. C, cytoplasm; Hc, heterochromatin; N, nucleus; NE, nuclear envelope. Scale bar, 0.5 μm.

We re-evaluated heterochromatin positioning within EpiLC nuclei upon lamin and LBR depletion and again observed heterochromatin displacement by H3K9me2 immunostaining (Fig. 6f–j), by DNA staining (Extended Data Fig. 8d,e), and by TEM (Fig. 6l,m and Extended Data Fig. 8f–j). However, H3K9me2 abundance increases during the transition from naive to primed pluripotency in all genotypes (Fig. 6f–i,k and Supplementary Fig. 7e,f). Therefore, the lamins and LBR are required for H3K9me2 positioning but not for H3K9me2 deposition. As H3K9me2 displacement phenocopies the effect of H3K9me2 loss on EpiLC survival[29], we infer that the lamins and LBR promote the essential functions of H3K9me2 during this developmental transition.

### Heterochromatin positioning influences remodelling of H3K9me2 in primed pluripotency

We applied spike-in controlled LB1 and H3K9me2 CUT&RUN to track WT LADs during the transition from naive to primed pluripotency (Supplementary Table 7). While LB1:genome contacts remain largely constant between WT mES cells and EpiLCs, H3K9me2 is consolidated into larger (Extended Data Fig. 9f) but less numerous domains (Extended Data Fig. 9g). LB1 and H3K9me2 overlap extensively in mES cells (Fig. 2b), but their overlap decreases in EpiLCs (Fig. 7a,b) as many LADs lose H3K9me2 (8,508 genes; Fig. 7a,c) and non-LADs gain H3K9me2 (8,985 genes; Fig. 7a,c). LADs remain strongly repressive despite this remodelling. In fact, LADs that lose H3K9me2 are more strongly repressed than those that retain H3K9me2 (Fig. 7d). Interestingly, the establishment of new KODs is not accompanied by widespread repression; instead, new KODs are highly transcribed (Fig. 7d). H3K9me2 therefore does not induce repression of newly modified loci in primed pluripotency.

We next applied CUT&RUN to mutant EpiLCs. However, we chose not to rely on LAP2β as a surrogate for LADs, as LAP2β CUT&RUN produced both LAD- and non-LAD-localized signal in WT EpiLCs (compared with LB1; Extended Data Fig. 9a,b). We therefore restricted our analysis to H3K9me2 (Fig. 7e and Supplementary Table 7). Each genotype underwent remodelling of H3K9me2 into larger but less numerous domains (Extended Data Fig. 9f,g) that overlap extensively (Fig. 7f). However, H3K9me2 domains in QKO EpiLCs are moderately more fragmented and numerous than in other genotypes (Extended Data Fig. 9f,g), and their position across the genome is most dissimilar to other genotypes, with about 12% of QKO domains being unique (Fig. 7f, last column). These findings suggest that H3K9me2 remodelling is impaired in QKO EpiLCs. While many lamina-associated H3K9me2 domains are removed in WT EpiLCs (LAD to LB1; Fig. 7a,c), H3K9me2 remains significantly higher in these domains in QKOs (Fig. 7e,g). H3K9me2 intensity is also moderately higher and more variable within EpiLC-specific H3K9me2 domains in QKOs (Fig. 7e,h and Extended Data Fig. 9h). These analyses indicate that displacement of heterochromatin from the nuclear periphery alters H3K9me2 remodelling during the transition from naive to primed pluripotency.

### Heterochromatin positioning silences transposons and alternative cell fate genes in EpiLCs

Finally, we evaluated the consequences of H3K9me2 displacement on the dynamic transcriptional landscape of EpiLCs by RNA-seq. Similarly to naive mES cells (Fig. 4), LBR KO and TKO EpiLCs each exhibit modest and nonoverlapping effects on gene expression, while co-depletion of the lamins and LBR induces derepression (Fig. 8a,b, Supplementary Table 9 and Extended Data Fig. 10a). GSEA again identified strong relationships between genes downregulated in QKO EpiLCs, linked to morphogenesis and neural fate-related GO terms (Fig. 8c), but very few relationships between upregulated genes (Supplementary Table 10). Analysis with enrichGO (Supplementary Table 11) uncovered upregulated genes involved in retrotransposon silencing (GO: 0015026), including *Dnmt3l*, *Morc1* and *Piwil2* (Fig. 8d). Genes involved in antiviral innate immune responses are also upregulated, including *Ifit1* and *Rigi* (Fig. 8d); these genes are induced by TE activity[66,67] and can drive

apoptosis[68]. Indeed, TE expression persists in QKO EpiLCs, with 298 TE families upregulated at least twofold (Extended Data Fig. 10b–e and Supplementary Table 8). TE expression is more moderate in QKO EpiLCs compared with mES cells (Fig. 5 and Extended Data Fig. 10h), suggesting that distinct repressive mechanisms may be induced in primed pluripotency or that only EpiLCs expressing moderate levels of TEs survive. Inspecting L1MdA_1 LINE1s confirms that derepressed copies reside in WT LADs and are highly H3K9me2 modified (Extended Data Fig. 10j,k). We conclude that some TEs are repressed in constitutive LADs in both mES cells and EpiLCs, and that H3K9me2 displacement selectively derepresses these TEs.

We evaluated the distribution of dysregulated genes across domains identified in WT EpiLCs (Fig. 7). This analysis revealed that both up- and downregulated genes are significantly enriched within EpiLC-specific KODs (Fig. 8e). H3K9me2 levels on dysregulated KOD genes are similar in both WT and QKOs (Fig. 8f), implying that heterochromatin displacement influences the function, but not the deposition, of H3K9me2. If positioning influences the repressive function of H3K9me2, then the effect of displacement should resemble the effect of H3K9me2 loss. To explore this possibility, we compared our dataset with a published dataset of genes dysregulated by G9a KO in E6.25 mouse epiblasts[29]. Genes upregulated in QKO EpiLCs are also upregulated in G9a KO epiblasts, indicating the existence of a functional relationship between the lamins, LBR and H3K9me2 in primed pluripotency (Fig. 8g).

G9a and H3K9me2 are essential for embryonic development[24,29]. The relationships we uncovered between chromatin positioning, H3K9me2 and G9a activity motivated us to investigate the effect of heterochromatin displacement on key cell fate genes (Fig. 8h). Mutant EpiLCs succeed in silencing naive pluripotency genes and activating primed pluripotency genes. However, QKO EpiLCs also abnormally express alternative cell fate genes including markers of the primitive endoderm (*Fgfr2*, *Gata6* and *Pdgfra*), an extra-embryonic lineage that should be mutually exclusive with the epiblast fate[69,70] (Fig. 8h). QKOs also abnormally express other lineage-specific genes, such as *Wnt5a*, *Wnt6*, *Wnt8a*, *Tgfb1* and *Eomes* (Fig. 8h). Many of these dysregulated cell fate genes reside within H3K9me2 domains, such as the primitive endoderm-specific transcription factor *Gata6* (Fig. 8i) and the morphogen *Wnt5a* (Fig. 8j). This outcome suggests that while recruitment of heterochromatin to the nuclear periphery is not required for the transition from naive to primed pluripotency per se, it enables the effective specification of a lineage-specific gene expression programme. We conclude that the lamins and LBR together control both the spatial positioning and the repressive capacity of H3K9me2 to shape cell fate decisions during early mammalian development.

## Discussion

### The lamins and LBR exert broad influence on heterochromatin organization and function

The parallel roles of lamin A/C and LBR in heterochromatin organization were first described over 10 years ago[34]. Here, we have shown that the lamins and LBR together exert broad control on heterochromatin organization and function in pluripotent cells. Strikingly, the dense layer of compacted, electron-dense heterochromatin that is a hallmark of most eukaryotic cells disperses when the lamins and LBR are ablated (Fig. 1l). CUT&RUN with the INM-localized protein LAP2β confirms the global scope of heterochromatin detachment in the absence of the lamins and LBR (Fig. 2e–g). These data demonstrate that the unique enrichment of heterochromatin underneath the nuclear periphery is maintained by the redundant function of the lamin and LBR proteins.

In the absence of the lamins and LBR, H3K9me2-marked loci are displaced from the nuclear periphery and into nucleoplasmic and/or nucleolus-associated foci (Fig. 1d,f). H3K9me2 is also found in heterochromatic nucleolar-associated domains (NADs), and some loci shuffle between LADs and NADs[19], raising the possibility that LAD-resident loci

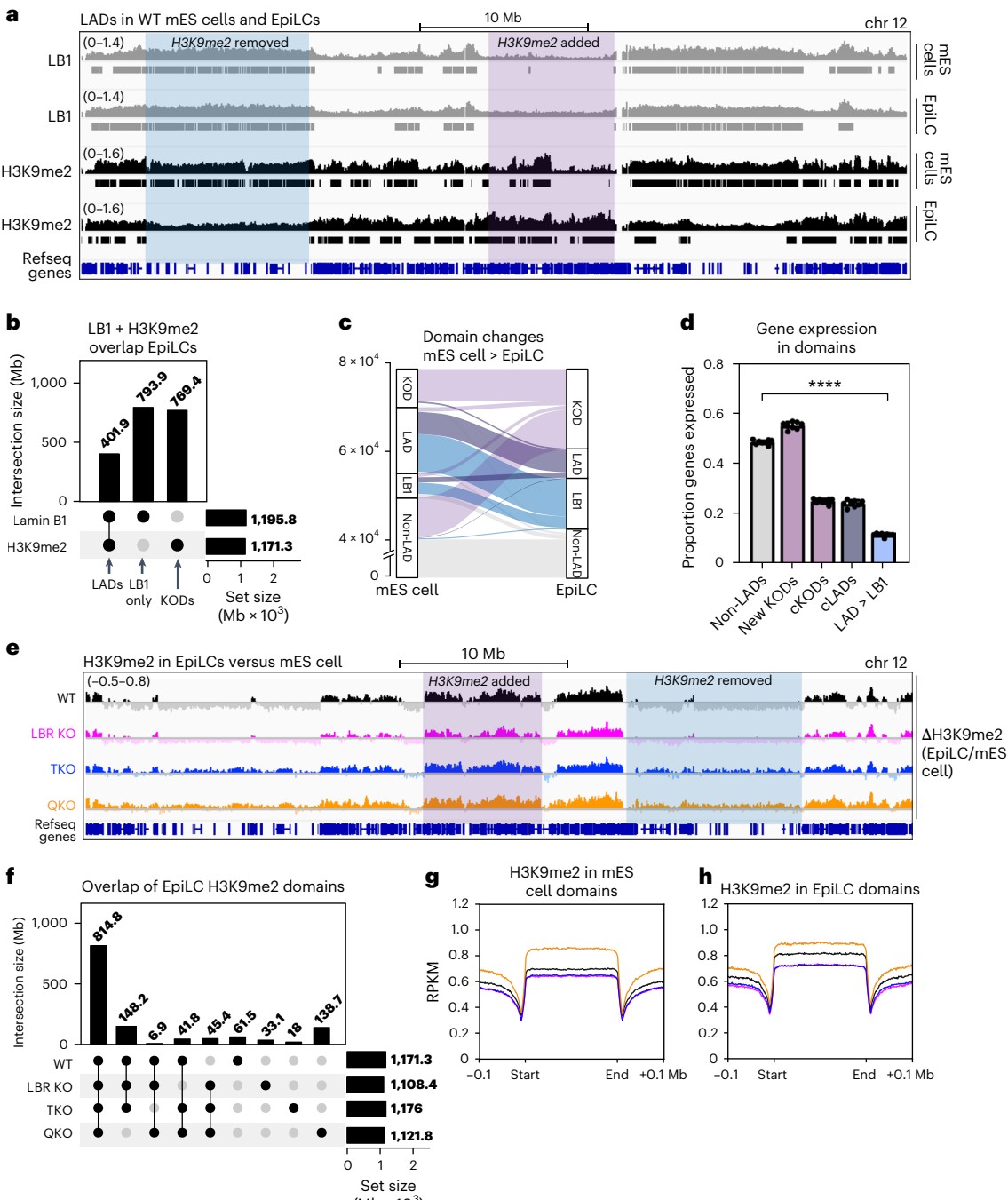

**Fig. 7 | Abnormal remodelling of H3K9me2 in lamin + LBR KO EpiLCs. a**, The average genome tracks and domain calls of spike-in controlled lamin B1 (LB1; top, grey) and H3K9me2 (bottom, black) CUT&RUN in WT mES cells and EpiLCs (n = 3 replicates per condition). Lamin B1 and H3K9me2 domains are marked in grey and black, respectively. The purple highlight indicates a H3K9me2 domain that appears in EpiLCs and the blue highlight indicates a LAD that is remodelled to remove H3K9me2 in EpiLCs. **b**, The overlap between lamin B1 and H3K9me2 domains in WT EpiLCs. **c**, An alluvial plot showing the movement of genes into and out of lamin B1 and H3K9me2 domains as WT mES cells differentiate into EpiLCs: 7,237 genes remain in constitutive KODs in both mES cells and EpiLCs (purple); 8,985 genes move from non-LADs into KODs in EpiLCs (purple); 4,990 genes remain in constitutive lamin B1-associated and H3K9me2-modified LADs in mES cells and EpiLCs (dark purple); 8,508 genes move from LADs into lamin B1-

only domains in EpiLCs (blue); 40,122 genes remain in non-LADs in both mES cells and EpiLCs (grey). **d**, The proportion of genes expressed (minimum of five TPMs) within non-LADs, new KODs, constitutive KODs (cKODs), constitutive LADs (cLADs) and LADs that lose H3K9me2 (LAD > LB1). N = 8 replicates. ****P < 0.0001 by one-way ANOVA. The columns indicate the mean and error bars indicate s.d. **e**, Difference maps of H3K9me2 signal in EpiLCs versus mES cells (average of three replicates per condition). The y-axis range indicated at the top left is the same for all tracks shown. The purple highlight indicates a H3K9me2 domain that appears in WT EpiLCs and blue highlight indicates a LAD that is remodelled to remove H3K9me2 in WT EpiLCs. **f**, The overlap of H3K9me2 domains across genotypes in EpiLCs. **g,h**, The averaged density of H3K9me2 in WT mES cell H3K9me2 domains (**g**) and in WT EpiLC H3K9me2 domains (**h**).

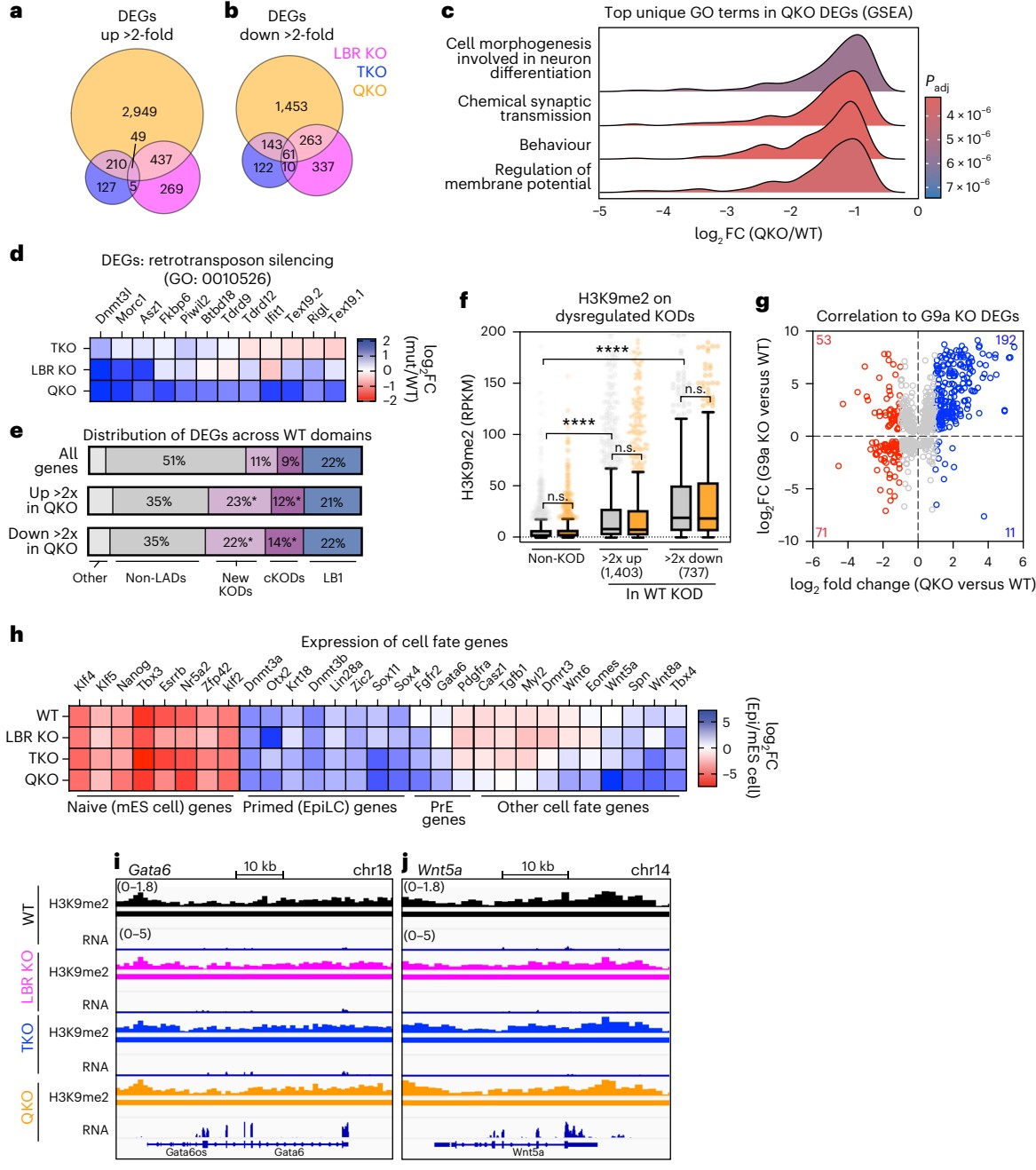

**Fig. 8 | Removal of the lamins and LBR derepresses genes and transposons and impairs lineage restriction in EpiLCs. a,b,** The number of genes significantly upregulated (**a**) and downregulated (**b**) at least twofold in LBR KO, TKO and QKO EpiLCs (n = 4 replicates per condition). **c,** Identification of the top unique GO terms in genes differentially expressed between QKO and WT EpiLCs (GO terms identified by GSEA and reduced by ReVIGO). See also Supplementary Table 9 for the full list of GO terms. **d,** The differential expression of genes involved in retrotransposon silencing (GO: 0010526), a GO term significantly enriched in QKO EpiLC upregulated genes, and transposon sensing (Ifit1 and Rig-I). **e,** The distribution of QKO DEGs across WT domain categories (non-LADs, new KODs, constitutive KODs (cKODs) and lamin B1 domains; refer also to Fig. 7c). *Significant enrichment of DEGs in both new and cKODs (P < 0.0001, two-sided Fisher's exact test). **f,** Analysis of H3K9me2 modification levels (averaged spike-in-controlled RPKM) on KOD genes upregulated >2-fold in QKO EpiLCs (n = 1,403), KOD genes downregulated >2-fold in QKO EpiLCs (n = 737) and a

random sample of non-LAD genes (n = 1,403). ****Comparisons indicated are significantly different (P < 0.0001) by Kruskal–Wallis multiple comparisons test with Dunn's correction; n.s., P > 0.999. The box plot (Tukey) centre line indicates the median, box limits indicate the 25th to 75th percentiles, whiskers indicate 1.5× the interquartile range and points indicate outlier values. **g,** The correlation between QKO EpiLC DEGs and a published dataset of G9a KO E6.25 epiblast DEGs (PMID 26551560). Blue indicates genes upregulated >2-fold in QKO EpiLCs and red indicates genes downregulated >2-fold in QKO EpiLCs. **h,** A heat map showing the expression of selected regulators of naive pluripotency, primed pluripotency, primitive endoderm or other cell fates in each genotype. The heat map values indicate log$_2$(FC) in expression between EpiLCs and mES cells. **i,j,** H3K9me2 and RNA levels of H3K9me2-modified cell fate regulatory genes Gata6 (**i**) and Wnt5a (**j**) across genotypes. The y-axis ranges indicated at the top left are the same for all the lower tracks within the panel.

move into NADs in the absence of the lamins and LBR. In neurons that downregulate lamin A/C and LBR as they differentiate, heterochromatin gradually coalesces into a central nucleoplasmic focus after cell cycle exit[34,71,72]. We surmise that the high proliferation rate of mES cells prevents the complete intranuclear coalescence of heterochromatin when the lamins and LBR are deleted. Alternatively, additional chromatin modifications and/or distinct chromatin-binding proteins may promote heterochromatin coalescence in neurons but not in mES cells. While neuronal chromatin inversion occurs without any disruption to heterochromatin-mediated repression[71,72], we find that heterochromatin displacement in pluripotent cells causes dramatic transcriptional dysregulation (Figs. 4 and 5).

Our data indicate that the lamins can sustain chromatin organization in the absence of LBR and vice versa. While LBR is much more highly expressed than lamin A/C in pluripotent cells, low levels of lamin A/C may influence gene expression in naive pluripotency[73]. Perhaps lowly expressed lamin A/C is sufficient to maintain chromatin organization in the absence of LBR; alternatively, the much more highly expressed B-type lamins may play a role in mES cells. LBR alone can drive peripheral heterochromatin positioning when ectopically expressed in various cell types, while lamin A/C cannot[34]. This implies that the heterochromatin tethering function of lamin A/C may be mediated by additional factors with variable expression levels across tissues, while LBR can either directly tether heterochromatin or alternatively work through ubiquitously expressed intermediary protein(s). LAP2β, HDAC3, PRR14 and others could promote lamin-mediated heterochromatin tethering[9,16,35,36], while LBR binds to the H3K9me2/3-binding protein HP1 (ref. 40) and can also interact with histone tails via its Tudor domain[41,42].

A recent study indicates that LAP2β and LBR together play a major role in heterochromatin organization in mammalian cells[37]. In contrast, we observe that LAP2β is unable to maintain heterochromatin positioning in the absence of the lamins and LBR (Figs. 1 and 2). As LAP2β binds selectively to the B-type lamins[74], one potential explanation that could unify these observations is that LAP2β requires the presence of the B-type lamins to enable its tethering function. Future dissections of these interactions in sensitized lamin-null or LBR-null backgrounds will provide further insight into the mechanisms of heterochromatin positioning.

### The nuclear periphery influences the repressive capacity of H3K9me2

We show that disruption of the lamins and LBR weakens the repression of H3K9me2-modified loci (Figs. 4e–g, 5d–f and 8f–h). While H3K9me2 has a well-established capability to repress transcription[24,25], our data demonstrate that nuclear spatial position influences H3K9me2 function and is consistent with findings that KODs are more permissive to transcription than are H3K9me2-modified LADs[32]. By acutely removing LBR in lamin-null mES cells, we determined that detachment of H3K9me2 precedes derepression (Extended Data Fig. 6f–k). Our data indicate that displacement of LADs modified with low versus high levels of H3K9me2 has distinct outcomes, as sparsely modified loci become permissive to transcription when displaced from the nuclear periphery, while densely modified loci remain potently repressed after displacement (Figs. 4e–g and 8f). It is possible that this latter group can relocate to NADs and remain repressed[75] as NADs are more densely modified by H3K9me2 than LADs[19].

Heterochromatin tethering could enhance repression by several potential mechanisms. Repression may be induced by steric occlusion if binding of H3K9me2 and oligomerization of tethering proteins together promote compaction of chromatin domains. Tethering to the nuclear periphery may limit the turnover of nucleosomes to enable long-term memory of chromatin state, as has been demonstrated in *S. pombe*[7]. Finally, the addition or removal of other chromatin marks may be enhanced at the periphery to consolidate repression.

For instance, tethering of a locus to the nuclear periphery is often accompanied by histone deacetylation[9,51].

### The nuclear periphery maintains repression of transposons

TEs are enriched in LADs in both pluripotent (Fig. 5a) and differentiated cells[23,56], and we find that lamin disruption allows the expression of some TEs in naive pluripotency (Extended Data Fig. 7d,g). Intriguingly, ageing- and senescence-linked alterations to the lamina have been linked to displacement and/or activation of TEs[76–79], suggesting that the lamina may also promote TE repression in differentiated cells.

H3K9me2 plays a major role in TE repression, and displacement of H3K9me2 from the nuclear periphery by ablation of the lamins and LBR potently activates TEs. By analysing the retrotransposition-active L1MdA_I LINE1 family, we determined that lamina-associated LINE1s are specifically activated by displacement despite retaining H3K9me2 modification (Fig. 5e,f). The magnitude of TE activation we observe is surprisingly comparable to the effect of removing H3K9me2 altogether. For instance, ablating G9a/GLP alone or in combination with SETDB1 and SETDB2 similarly derepresses many ERV LTRs as well as LINE1s[26–30,60]. We conclude that lamina tethering is required for the effective repression of H3K9me2-modified TEs in pluripotent cells.

TE activation has wide-ranging effects on genome function and stability (reviewed in ref. 80). If mobile TE families are activated by transcription, they will perturb other loci by transposition. Mobile TEs including L1MdA_I LINE1s[58] and ERV LTRs[81] are derepressed in QKOs. Further, the L1-encoded Orf1p protein is expressed in TKO and QKO mES cells (Fig. 5g–k), suggesting that heterochromatin displacement may unleash retrotransposition. The mobilization of autonomous TEs (such as LINE1s) can in turn induce non-autonomous TEs such as SINEs[80]. The cascading effects of TE activation may underlie the observed changes in both gene and TE expression that include, but are not limited to, H3K9me2-modified loci.

Pluripotent and differentiated cells differ in their sensitivity to TE expression: differentiated cells express innate immune proteins that sense TE-derived RNA and induce apoptosis[66–68] while pluripotent cells do not[82]. We noted pervasive expression of TEs in both QKO mES cells (Fig. 5) and EpiLCs (Extended Data Fig. 10b,e). QKO EpiLCs upregulate the RNA sensors *Rig-I* and *Ifit1* (Fig. 8d), leading us to speculate that TE expression may induce apoptosis in primed pluripotency.

### The nuclear periphery regulates H3K9me2 remodelling and function during differentiation

We show that H3K9me2 domains are remodelled as cells enter primed pluripotency (Fig. 7). While H3K9me2 is strongly enriched in naive LADs, it is instead found on many non-LADs in the primed state. Interestingly, LADs remain strongly repressive despite this remodelling, suggesting that other chromatin modifications and/or cofactors maintain repression of primed LADs. In the absence of the lamins and LBR, removal of H3K9me2 from LADs is blunted (Fig. 7e–g). In addition, H3K9me2 accumulates on new KODs to higher and more variable levels in QKO EpiLCs (Fig. 7e,h and Extended Data Fig. 9h). The levels of the G9a/GLP methyltransferase increase during EpiLC differentiation[29], but we do not observe further upregulation of G9a or GLP in QKOs that could explain the abnormally high levels of H3K9me2 observed. We conclude that heterochromatin tethering influences the deposition of H3K9me2 and/or its perdurance on modified loci in primed pluripotency.

Primed pluripotency is accompanied by H3K9me2 deposition onto loci that remain highly transcribed (Fig. 7d), including enhancers and promoters that retain chromatin marks associated with active transcription, such as H3K27ac[29,63]. This transitory co-occupation of *cis*-regulatory elements with H3K27ac and H3K9me2 has been proposed to prime these loci for future silencing[29,63]. We infer that this G9a-mediated process may be disrupted in QKO EpiLCs, as dysregulated genes overlap with G9a target genes (Fig. 8g) and are enriched

in EpiLC-specific KODs (Fig. 8e). Further, lineage-specific genes are discordantly co-expressed (Fig. 8h–j). We conclude that heterochromatin scaffolding at the nuclear periphery enables the repression of lineage-irrelevant genes and is required for the normal orchestration of development.

## Online content

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

## Methods

### Generation of knockout and miR-E-expressing mES cells

Lamin TKO (vial 'TKO97', passage 12) and littermate WT (vial '28538', passage 9) mES cells were from the laboratory of Dr. Yixian Zheng[43]. Lamin TKO was validated by endpoint PCR using exon-flanking primers published by the Zheng lab. Cell lines tested negative for mycoplasma and were karyotyped by WiCell. TKO mES cells had an abnormal karyotype with two populations: one having trisomy 8 and the other trisomy 8 and loss of the Y-chromosome. WT cells had 75% normal ploidy with two populations: one with loss of the Y-chromosome and the other with trisomy 6 and 8 and loss of the Y-chromosome.

Lbr knockout was performed by Lipofectamine 2000 (Invitrogen, 11668019) transfection of the PX458 CRISPR–Cas9 plasmid (Addgene, 48138) with one guide targeting exon 2 of Lbr (5′-TCATAATAA AGGGAGCTCCC-3′). After 48 h of recovery, cells were sorted by GFP fluorescence and seeded on a 10 cm dish at a density of 3,000 cells per dish. After 1 week, visible colonies were picked by pipetting into 96-well plates.

The Lbr knockout guide was designed to overlap a restriction enzyme site SacI near the Cas9 cut site such that indels would destroy the site. Genotyping was performed by PCR amplification of genomic DNA with primers flanking the SacI site followed by SacI digestion of the PCR product and running the digest on a 2% agarose gel. Homozygous indels showed one band at 1,388 bp. Heterozygous indels showed three bands at 1,388 bp, 938 bp and 450 bp, while unedited clones showed only 938 bp and 450 bp bands. This strategy yielded 36 homozygous Lbr indel clones from 78 screened in WT mES cells, and 10 homozygous Lbr indel clones from 85 screened in TKO mES cells. The presence of indels was verified by Sanger sequencing and analysis with CRISP-ID[83]. LBR depletion was validated by western blotting (Abcam, ab232731). While we initially isolated ten QKO clones, only one clone survived thaw after initial freeze. This single QKO clone is the major clone used for all experiments. A second attempt at isolating QKO mES cells yielded 8 homozygous Lbr indel clones out of 96 screened in TKO mES cells, but all died within 2 weeks.

To make mES cells expressing tetracycline-inducible miR-Es targeting Lbr or Luc, puromycin, hygromycin and neomycin resistance markers were first excised from lamin TKO mES cells by transient expression of Cre recombinase and clonal selection. Cre recombinase was expressed from a plasmid derived from pPGK-Cre-bpA (Addgene, 11543) with an SV40 polyA sequence instead of the bpA. Then, 97-mer oligos were ordered containing miR-E targeting LBR ('LBR634' 5′-TGCT GTTGACAGTGAGCGCCAGATATATAGTTACACAGTATAGTGAAGCCACA GATGTATACTGTGTAACTATATATCTGTTGCCTACTGCCTCGGA-3′) and Renilla Luciferase (5′-TGCTGTTGACAGTGAGCGCAGGAATTATAATGCT TATCTATAGTGAAGCCACAGATGTA TAGATAAGCATTATAATTCCTATGC CTACTGCCTCGGA-3′). These were amplified by two miR-E universal primers (fwd 5′-CTTAACCCAACAGAAGGCTCGAGAAGGTATATTGCT GTTGA CAGTGAGCG-3′) (rev 5′-ACAAGATAATTGCTCGAATTCTAGCC CCTTGAAGTCCGAGGCAGT AGGCA-3′) and cloned into the XhoI and EcoRI double-digested LT3GEPIR lentiviral vector (Addgene, 111177). HEK293T cells were transfected with these vectors to produce a lentivirus that was then used to transduce lamin TKO mES cells. Cells were selected with puromycin (1 μg ml⁻¹) for 4 days, then individual clones were selected. Using flow cytometry, GFP signal after doxycycline treatment (1 μg ml⁻¹) was measured to assess the expression of the RNAi system. Clones with a GFP-positive population greater than 88% were selected for experiments.

### Cloning of Halo-LBR expression constructs and generation of Halo-LBR QKO mES cells

The LBR coding sequence was cloned from WT mES cell cDNA by PCR using the forward primer 5′-GGCGGTAGATCTATGC CAAGTAGGAAGTTTGT-3′ and reverse primer 5′-ACTGCCGGATC CGTAAATGTAGGGGAATATGC-3′. The Halotag sequence plus a 10 serine-glycine linker was PCR amplified from a published plasmid (Addgene, 119907) using the forward primer 5′-GCAAAGAATTC CTCGAGGCCGCCACCATGGAA ATCGGTAC-3′ and the reverse primer 5′-ACAAACTTCCTACTTGGCATGACCGGTGGC CCTCCGCTAC CGCCAGA-3′. The LBR and Halo sequences were joined via Gibson cloning (NEBuilder HiFi DNA Assembly Master Mix, E2621L). This full-length Halo-LBR plasmid was used as a template to PCR amplify the nucleoplasmic (fwd 5′-CGGTAGCGGAGGGCCACCGGTCATGC CAAGTAGGAAGTTTGT-3′, rev 5′-GCCCTCTCCACTGCCGGATCCTC CAAACTCCAAGTCCTTCC-3′) and the transmembrane (fwd 5′-CGG TAGCGGAGGGCCACCGGTCGGAGTACCTGGTGCGGTC-3′, rev 5′-GCC CTCTC CACTGCCGGATCCGTAAATGTAGGGGAATATGCGG-3′) fragments of LBR to clone a nucleoplasmic-only (Halo-LBR-NP) and a transmembrane-only (Halo-LBR-TMD) plasmid. In addition, Halo-only with a multiple cloning site (fwd 5′-CCGGTCACGCGTAGATCTGG TACC G-3′, rev 5′-GATCCGGTACCAGATCTACGCGTGA-3′) and Halo-NLS from SV40 (fwd 5′- CCGGTCCCCAAAAAGAAACGGAAAGTAGACCC CAAGAAGAAACGCAAGGTGGACCCGAAAAAGAAGCGGAAAGTCG -3′, rev 5′- GATCCGACTTTCCGCTTCTTTTTCGGGTCCACCTTGCG TTTCTTCTTGGGGTCTACTTTCCGTTTCTTTTTGGGGA -3′) plasmids were cloned by annealing the respective forward and reverse oligos, then using these fragments to swap out the LBR insert by restriction enzyme digest and ligation.

Constructs were stably integrated into QKO mES cells by co-electroporating $1 × 10^6$ cells with 2 μg of Halo-LBR or Halo-NLS PiggyBac donor plasmid and 2 μg of PiggyBac transposase plasmid (Systembio, PB200A-1) using the Neon Transfection System (Invitrogen). After recovering for 1 day, cells were selected for stable integration of the construct with 10 μg ml⁻¹ blasticidin for 10 days.

### mES cell culture and EpiLC differentiation

mES cells were cultured at 37 °C in 5% $CO_2$ under normoxic conditions in serum-free 2i + LIF medium (N2B27 basal medium, 3 μM CHIR-99021 (Selleckchem, S1263), 1 μM PD0325901 (Selleckchem, S1036), $10^3$ U ml⁻¹ LIF (Millipore Sigma, ESG1107), 55 μM β-mercaptoethanol (Gibco, 21985023) and 1× penicillin–streptomycin (GenClone, 25-512)). N2B27 medium was made by mixing equal parts of Dulbecco's modified Eagle medium:F12 (GenClone, 25-503) and Neurobasal medium (Gibco, 21103049) then adding 1× N-2 (Gibco, 17502001), 1× B-27 (Gibco, 17504001) and 2 mM Glutamax (Gibco, 35050061). Dishes were coated with 0.1% gelatin (Millipore Sigma, ES-006-B) for 30 min at 37 °C before seeding cells. The medium was replenished every day and cells were passaged every other day, seeding $4 × 10^5$ cells per 6-well plate (approximately $4.21 × 10^4$ cells cm⁻²).

Differentiation of mES cells into epiblast-like cells was adapted from a published protocol[84]. In brief, a 6-well plate was coated with 1 ml of a 1:200 solution of either Geltrex (Gibco, A1413202) or Cultrex (Bio-Techne, 3445-005-01) in N2B27 at 37 °C. After 1 h, that solution was aspirated and $2× 10^5$ mES cells were seeded (approximately $2.11 × 10^4$ cells cm⁻²) with N2B27 media containing a final concentration of 20 ng ml⁻¹ FGF (Peprotech, 100-18B) and 12 ng ml⁻¹ activin A (Peprotech, 120-14E). Medium was replenished the next day.

For miR-E timepoints followed by RNA-seq, mES cells were seeded at $4.21 × 10^4$ cells cm⁻² on dishes precoated with 0.1% gelatin. mES cells were replenished with fresh 2i + LIF medium every day. Untreated controls were collected after 2 days. Treated cells received doxycycline at 1 μg ml⁻¹. For day 4 doxycycline timepoints, mES cells were split at day 2, keeping the same seeding density.

### RNA isolation and library preparation for RNA-seq

Four replicates, each from a single 6 well of cells, were prepared per condition. Total RNA was isolated using the RNeasy Plus Mini Kit (Qiagen, 74136). RNA was treated with DNAse using the TURBO DNA-free kit (Invitrogen, AM1907), and RNA integrity was confirmed by presence of intact 28S and 18S rRNA bands following electrophoresis and

ethidium bromide staining (0.6 µg ml⁻¹) on a 1% agarose gel. Libraries were prepped using the Illumina Stranded Total RNA Prep with Ribo-Zero Plus kit (Illumina, 20040525) following the manufacturer's protocol with 500 ng input. Libraries were amplified for 11 cycles. Each library was uniquely indexed with Integrated DNA Technologies for Illumina–DNA/RNA UD Indexes (Illumina, 20026121), then pooled together in equimolar amounts. Library concentration was measured using the Qubit dsDNA HS assay kit (Invitrogen, Q32851) and Qubit 4 fluorometer. Library size was assayed using the Agilent Bioanalyzer 2100 with the high-sensitivity DNA kit (Agilent, 5067-4626). Paired-end sequencing was performed on pooled libraries using the Illumina NextSeq2000 (read length of 150 bp, ~50 million reads per library).

For miR-E timepoints, total RNA was isolated using the RNeasy Plus Mini kit. The Tecan Genomics' Universal Plus mRNA prep kit (Tecan Genomics, 0520B-A01) was modified for rRNA depletion by replacing the mRNA isolation in the first segment of the protocol with ribo-depletion using FastSelect rRNA (Qiagen, 334387) as follows: 200 ng of total RNA in 10 µl of water was mixed with 10 µl of Universal Plus 2× Fragmentation Buffer and 1 µl FastSelect. The solutions were fragmented and ribo-depleted simultaneously per the FastSelect protocol. Then, 20 µl of the resulting fragmented/ribo-depleted RNA was prepped per the Universal Plus protocol starting with first strand synthesis. The Universal Plus Unique Dual Index Set B was used, and the samples were quality checked using a MiniSeq benchtop sequencer (Illumina). Normalized pools were made using the corrected protein-coding read counts of each. All concentrations and library/pool qualities were measured via a Fragment Analyzer (Agilent Technologies). Paired-end sequencing was performed on pooled libraries using the Illumina NovaSeq6000 platform at the UCSF CAT core (read length of 100 bp, ~ 30 million reads per library).

## RNA-seq analysis

Raw sequencing reads were trimmed of the first T overhang using cutadapt[85] (version 2.5). Trimmed reads were mapped to the GENCODE primary assembly M30 release of the mouse genome (GRCm39) using STAR[86] (version 2.7.10a): --runMode alignReads --readFilesCommand zcat --clip3pAdapterSeq CTGTCTCTTATA CTGTCTCTTATA --clip3pAdapterMMp 0.1 0.1 --outFilterMultimapNmax 1 --outSAMtype BAM SortedByCoordinate --twopassMode Basic. Bam files were indexed with SAMtools[87] (version 1.10). The deepTools2 (ref. [88]) (version 3.4.3) function bamCoverage was used to generate reads per kilobase per million reads (RPKM) normalized coverage track files. The subread featureCounts[89] package (version 1.6.4) and GENCODE M30 primary annotation file was used to generate a read counts table for all conditions, which was used as an input for DESeq2 (ref. [90]) (version 1.40.2) to obtain normalized counts and perform differential gene expression analysis following a published pipeline[91]. The subread featureCounts package was used again for obtaining read counts on uniquely mapped TEs using a custom annotation file made by the Hammell lab[92]. For miR-E experiments, reads were not trimmed of the first T overhang, but downstream analysis remained the same.

For multimapping of RNA-seq reads for TEtranscripts[57] analysis, STAR parameters were changed from –outFilterMultimapNmax 1 to the following: –outFilterMultimapNmax 100 --winAnchorMultimapNmax 100. A read counts table was generated using the TEcount command from the TEtranscripts package (version 2.2.3) with the following parameters: --sortByPos --format BAM --mode multi. The GENCODE M30 primary annotation file was used for gene annotations and a custom annotation file from the Hammell lab for TEs were used to generate a read counts table for use in DESeq2 analysis.

GO term enrichment analysis was performed with the enrichGO function from the clusterProfiler R package using the 'Biological Processes' subontology. The Benjamini–Hochberg method was used to adjust *P* values and results were filtered by having an adjusted *P* value less than 0.01. GSEA was performed with the gseGO function from the

clusterProfiler R package using the 'Biological Processes' subontology and default settings and minimum and maximum gene set sizes of 25 and 500, respectively.

Transcript isoforms were quantified from fastq files with the quant function from Salmon (version 1.10.3) using the default settings and the mouse GRCm39 transcriptome.

## CUT&RUN and library preparation

CUT&RUN was performed as described in ref. [93]. Live cells were collected with accutase and washed with PBS. Three replicates containing 200,000 cells were used per condition, except for H3K9me3 CUT&RUN (two replicates only). H3K9me2 (abcam, 1220, 1:100 dilution), LAP2 (Invitrogen, PA5-52519, 1:100), H3K9me3 (abcam, 8898, 1:1,000 dilution) or lamin B1 (Santa Cruz, sc56144, 1:500 dilution) antibodies were incubated at 4 °C overnight. Rabbit anti-mouse secondary antibody (abcam, ab6709, 1:100 dilution, 4 °C 1 h incubation) was used for samples incubated with H3K9me2 primary antibody and as an IGG negative control in conjunction with the LAP2 CUT&RUN. pA-MNase (batch #6 143 µg ml⁻¹) generously gifted from the Henikoff lab was used for the cleavage reaction. The 2× STOP buffer included genomic DNA from *S. cerevisiae* such that each sample received 0.3 pg of spike-in DNA as a spike-in control. The chromatin release step was carried out at 37 °C; H3K9me2 samples were incubated for 30 min whereas LAP2, H3K9me3, lamin B1 and IGG negative control samples were incubated for 1 h.

DNA was isolated by phenol–chloroform extraction, then resuspended with 40 µl of 1 mM Tris–HCl pH 8.0 and 0.1 mM EDTA. DNA concentration was measured using the Qubit dsDNA HS assay kit (Invitrogen, Q32851) and Qubit 4 fluorometer. DNA quality was assayed using the Agilent Bioanalyzer 2100 with the high-sensitivity DNA kit. Libraries were prepared using the NEBNext Ultra II DNA Library Prep with Sample Purification Beads (NEB, E7103S) following the manufacturer's protocol with 5 ng input. Libraries were amplified for eight cycles. Each library was uniquely indexed with NEBNext Multiplex Oligos for Illumina (NEB, E6440S), then pooled together in equimolar amounts. Library size was analysed with the Agilent Bioanalyzer 2100 using the high-sensitivity DNA kit. Paired-end sequencing was performed on the pooled library. H3K9me2 libraries were sequenced with the Illumina NextSeq2000 (read length of 35 bp, ~16 million reads per library). The Illumina NovaSeq X Plus was used to sequence H3K9me3 (read length of 50 bp, ~78 million reads per library) and LAP2 (read length of 50 bp, ~27 million reads per library) libraries.

## CUT&RUN analysis

Raw sequencing reads were mapped to the 'soft masked' mm39 mouse genome using Bowtie2 (ref. [94]) (version 2.3.5.1): -X 2000 -N 1 --local --dovetail. SAMtools was used to keep properly paired, primary alignments and remove unassembled contigs, duplicates and mitochondrial reads.

RPKM signal coverage in 1 kb and 10 kb bins were calculated using the deepTools2 function bamCoverage using the GENCODE M30 primary annotation file and output to bigwig files. Blacklist regions in the mm39 genome were obtained from a published dataset and excluded from the RPKM calculation[95]. The pomegranate (version 0.15.0) package in Python (version 3.11.10) was used to train HMMs using the Baum–Welch algorithm for WT naive mES cells and WT EpiLCs from three replicates of 10 kb-binned bigwig files. Four models (two state, three state, four state and five state) were built and evaluated for fitness to the training data using the Akaike information criterion and Bayesian information criterion[23].

A two-state model was chosen to identify two regions ((1) background and (2) high signal) of H3K9me2, H3K9me3 and LAP2 signal on replicate bigwig files from all available conditions using the Viberti algorithm with the output saved as BED files. Replicate BED files were merged to contain only the regions shared by all replicates using the bedtools (version 2.30.0) intersect and merge functions. The per cent

overlap between gene bodies and regions was calculated using the R packages GenomicRanges, rtracklayer and biomaRt. Genes in domains were called by having a 90% or greater overlap between the gene body and HMM domains.

Scale factors were calculated from the *S. cerevisiae* spike-in control to quantitatively compare RPKM signal between samples. Raw sequencing reads were mapped to the sacCer3 genome assembly and Bowtie2 and SAMtools were used for alignment and post-alignment processing. The SAMtools function flagstat was used to find the number of properly paired reads in sacCer3 alignments. These were used to calculate a scale factor by dividing an arbitrary constant number (3,000 for H3K9me2 CUT&RUN; 7,000 for H3K9me3 CUT&RUN; 8,000 for LAP2 CUT&RUN) by the number of properly paired reads[96]. Using the deepTools function bamCoverage, scale factors were used to generate 1 kb- and 10 kb-binned bigWig files containing spike-in normalized RPKM signal coverage tracks. For each condition, replicate 1 kb-binned and 10 kb-binned bigWig files were averaged using deepTools function bigwigAverage. H3K9me2 and LAP2 density in domains were plotted using the deepTools functions computeMatrix and plotProfile. Kernel density plots were made in R with the packages rtracklayer, GenomicRanges and ggplot2.

For TEs, spike-in controlled RPKM signal coverage in 10 kb bins were calculated with the deepTools function bamCoverage using a custom annotation file from the Hammell lab and including Blacklist regions where many TEs reside. For calculating modified $z$ scores of RPKM signal, RPKM values for each TE were first averaged across replicates. Then, the median RPKM and median absolute deviation were calculated for all TEs. A modified $z$ score for each TE was calculated using the following formula: $0.6745 \times$ (individual TE RPKM − median RPKM score)/(median absolute deviation RPKM).

## Immunoblotting

Cell pellets were lysed in buffer containing 8 M urea, 75 mM NaCl, 50 mM Tris pH 8.0 and cOmplete, Mini, EDTA-free Protease Inhibitor Cocktail (Roche, 11836170001). Lysates were run on 4–15% Mini-PROTEAN TGX precast gels (Bio-Rad, 4561083) and blots were probed with anti-LBR (1:1,000, Proteintech, 12398-1-AP), anti-SOX2 (1:1,000, Abcam, ab97959) and anti-LAP2 (1:1,000, Invitrogen, PA5-52519) primary antibodies in 5% milk TBST. Anti-mouse HRP-conjugated (Rockland, 610-1302) and anti-rabbit HRP-conjugated (Rockland, 611-1302) secondary antibodies were used at 1:5,000. Blots were visualized using Pierce ECL western blotting substrate (Thermo Scientific, 32209).

For quantitative immunoblotting of histone marks, purified histone extracts were isolated. Cell pellets were washed twice with ice-cold PBS with 5 mM sodium butyrate then lysed with Triton extraction buffer (PBS, 0.5% Triton X-100, 2 mM phenylmethylsulfonyl fluoride and 0.02% sodium azide) on ice for 10 min. Nuclei were isolated by centrifugation at 650$g$ for 10 min at 4 °C then washed in half the volume of TEB. Nuclear pellets were resuspended in 0.2 N HCl for overnight acid extraction of histones at 4 °C. The supernatant was collected and neutralized with 1/10th the volume of 2 M NaOH. Histone extracts from WT EpiLCs (0.5 µl, 1 µl, 2 µl, 5 µl and 10 µl) were used for a standard curve. For all samples, a 5 µl volume of histone extract was loaded. The primary antibodies used were H3K9me2 (Abcam, ab1220, 1:1,000) and H4 (Active Motif, 39269, 1:1,000). Secondary antibodies were anti-rabbit IgG (H + L) (DyLight 680 conjugate) (Cell Signaling, 5366P, 1:10,000) and IRDye 800CW goat anti-mouse IgG1-specific secondary antibody (Licor, 926-32350, 1:5,000). Blots were imaged with a LiCor Odyssey CLx imager.

## Cholesterol quantification assay

mES cells ($2 \times 10^6$ cells per sample) were lysed in 250 µl buffer containing 25 mM HEPES pH 7.4, 150 mM NaCl, 1% Trixon-X 100, 0.1% SDS and cOmplete, Mini, EDTA-free Protease Inhibitor Cocktail (Roche, 11836170001). Lysate was clarified by centrifugation at 10,000 rpm for 10 min at 4 °C. To determine cholesterol levels, 10 µl of each clarified lysate was analysed with the Amplex Red Cholesterol Assay kit (Invitrogen, A12216) according to the manufacturer's instructions.

## Immunofluorescence and Halo tag staining

IBIDI chambers (Ibidi USA, 80826) were coated with 150 µl of a 1:200 solution of Geltrex or Cultrex in N2B27 for 1 h at 37 °C. To obtain well-isolated cells for optimal imaging, $3 \times 10^4$ cells in 250 µl media were seeded into each well. After ~7 h of incubation, mES cells were washed with PBS then fixed in fresh 4% formaldehyde (Thermo Scientific, 28908) in PBS for 5 min. Cells were washed then stored at 4 °C until staining.

For visualizing Halo-tagged constructs, cells were stained before fixation by incubation in 2i + LIF medium containing 200 nM of Janelia Fluor 549 HaloTag ligand (Promega, GA1110) for 30 min, followed by a 5 min washout in N2B27 medium. To further reduce background, N2B27 was replaced for 2i + LIF medium and incubated 1 h before fixation.

Fixed cells were permeabilized in immunofluorescence buffer containing 0.1% Triton X-100, 0.02% SDS, 10 mg ml$^{-1}$ BSA in PBS. Primary and secondary antibodies were diluted in immunofluorescence buffer and incubated at room temperature for 2 h and 1 h, respectively, with immunofluorescence buffer washes in between antibodies. DNA was stained with Hoechst alongside the secondary antibody incubation. Primary antibodies used were H3K9me2 (1:300, Abcam, ab1220), H3K9me2 (1:1,000, ActiveMotif, 39041), LAP2 (1:400, Invitrogen, PA5-52519), LBR (1:500, Abcam, ab232731) and ORF1p (1:100, Abcam, ab216324). Secondary antibodies used were anti-mouse Alexa Fluor 488 (1:1,000, Invitrogen, A-11029), anti-rabbit Alexa Fluor 488 (1:1,000, Invitrogen, A-11008), anti-mouse Alexa Fluor 568 (1:1,000, Invitrogen, A-11004) and anti-rabbit Alexa Fluor 568 (1:1,000, Invitrogen, A-11011).

## Image acquisition and analysis

Confocal images were taken using a Nikon spinning disk confocal microscope with a 60× 1.4 numerical aperture oil objective. Images were acquired as 0.3 µm step zstacks. Unprocessed 16-bit images were saved as ND2 files using the Nikon Elements 5.02 build 1266 software. FIJI (ImageJ2 version 2.14.0) was used for cropping, producing max intensity projections, zslices and converting images to TIFF files. CellProfiler[97] was used to quantify signal intensity ('MeasureObjectIntensity' module) and radial intensity distribution ('MeasureObjectIntensityDistribution' module). LAP2β staining was used to segment nuclei. Signal intensity was quantified from max intensity projection images. Radial intensity distribution of H3K9me2 was quantified from a manually selected central nuclear $z$ slice. Output values were processed with RStudio and graphed with Prism.

## TEM and image analysis

Cells were seeded at $4.21 \times 10^4$ cells cm$^{-2}$ on 18 mm circular coverslips coated with Geltrex (Gibco, A1413202). Cells were fixed in Karnoversusky's fixative: 2% glutaraldehyde (EMS, 16000) and 4% formaldehyde (EMS, 15700) in 0.1 M sodium cacodylate (EMS, 12300) pH 7.4 for 1 h, then post-fixed in cold 1% osmium tetroxide (EMS, 19100) in water and allowed to warm for 2 h in a hood, washed 3× with ultrafiltered water, then stained for 2 h in 1% uranyl acetate at room temperature. Samples were dehydrated in a series of 10 min ethanol washes at room temperature: 30%, 50%, 70%, 95%, 100% and 100%, then propylene oxide for 10 min. Samples were infiltrated with EMbed-812 resin (EMS, 14120) mixed 1:1, and 2:1 with propylene oxide for 2 h each. The samples were then placed into EMbed-812 for 2 h, opened and then placed into flat moulds with labels and fresh resin and placed into 65 °C oven overnight.

Cells of interest were cut out with a gem saw and remounted on pre-labelled resin blocks with fresh resin then polymerized overnight. Once full, polymerized the glass coverslip was etched away using hydrofluoric acid for 20 min. Using the finder grid pattern left behind, the block faces were trimmed down, allowing for serial sectioning of the cells of interest.

Sections were taken around 90 nm, picked up on formvar/carbon-coated slot Cu grids, stained for 40 s in 3.5% uranyl acetate

# Article

in 50% acetone followed by staining in 0.2% lead citrate for 6 min. Observed in the JEOL JEM-1400 120 kV and photos were taken using a Gatan Orius 2k X 2k digital camera.

TEM images were quantified using FIJI. For each cell, 40 coordinates beneath the nuclear envelope and 40 coordinates inside the nucleoplasm were manually selected. Using a FIJI macro, a 0.2 μm × 0.2 μm square was drawn centred at each coordinate point, and the integrated signal density was measured. The average signal in 40 squares per region was used to calculate a nuclear envelope to nucleoplasm signal ratio per cell.

## Statistics and reproducibility
No statistical methods were used to predetermine sample size; however, no data were excluded from any analyses. Data were not randomized or blinded to the investigators before outcome assessment. Appropriate statistical tests were chosen based on the normal or non-normal distribution of the data.

## Reporting summary
Further information on research design is available in the Nature Portfolio Reporting Summary linked to this article.

## Data availability
Genomic and transcriptomic data generated during the course of this study were uploaded to the GEO database under the following accession numbers: mES cell and EpiLC RNA-seq data, GSE264599; LBR miR-E RNA-seq data, GSE264602; H3K9me2 CUT&RUN data, GSE264603; and LAP2β and LB1 CUT&RUN data, GSE288122. Source data are provided with this paper.

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

## Acknowledgements
We thank J. Tran and Y. Zheng for sharing lamin TKO and WT littermate mES cells, S. Henikoff for sharing protein-A-MNase for CUT&RUN, K. Meyer and E. Nora for guidance culturing and CRISPR-editing mES cells, K. Hansen for guidance with genomic library preparations and analyses, V. Virrueta for experimental assistance, the UCSF Genomics CoLab for RNA-seq library preparation and the UCSF CAT core for sequencing, the UCSF Wynton HPC for the computing power that enabled our bioinformatics analysis, and J. Perrino and the Stanford CSIF for TEM sample preparation and imaging. We thank G. Narlikar and O. Weiner as well as members of the Buchwalter and Nora labs for their helpful discussions since the origins of this project. A.B. was supported by grants from the National Institute of General Medical Sciences (R35GM142897) and the Chan Zuckerberg Biohub. B.A.-S. was supported by grants from the National Institute of General Medical Sciences (grant no. R35GM141888) and National Science Foundation (BIO directorate, 2113319). The project described was supported, in part, by an NIH S10 award from the Office of Research Infrastructure Programs (grant no. 1S10OD028536-01), although the work described here is solely the responsibility of the authors and does not represent the official views of the NCRR or the National Institutes of Health. Data for this study were acquired at the Center for Advanced Light Microscopy–CVRI Microscopy core on microscopes purchased though the UCSF Research Evaluation and Allocation Committee, the Gross Fund and the Heart Anonymous Fund.

## Author contributions
H.C.M. and A.B. conceived the project. H.C.M., E.S. and A.B. designed the experiments. H.C.M., E.S., C.A. and A.B. performed experiments. B.P., B.A.-S. and A.B. contributed reagents. H.C.M. generated all cell lines used in this study. H.C.M. performed immunofluorescence and image analysis. H.C.M. and E.S. performed CUT&RUN and analysis. H.C.M. performed RNA-seq and analysis. H.C.M. and C.A. performed EpiLC differentiation and cell count analysis. E.W.M. advised on the bioinformatics analysis. H.C.M. and A.B. analysed data and wrote the manuscript. B.P., B.A.-S. and A.B. supervised the project. All authors discussed results, reviewed and edited the manuscript.

## Competing interests
The authors declare no competing interests.

## Additional information
**Extended data** is available for this paper at https://doi.org/10.1038/s41556-025-01703-z.

**Correspondence and requests for materials** should be addressed to Abigail Buchwalter.

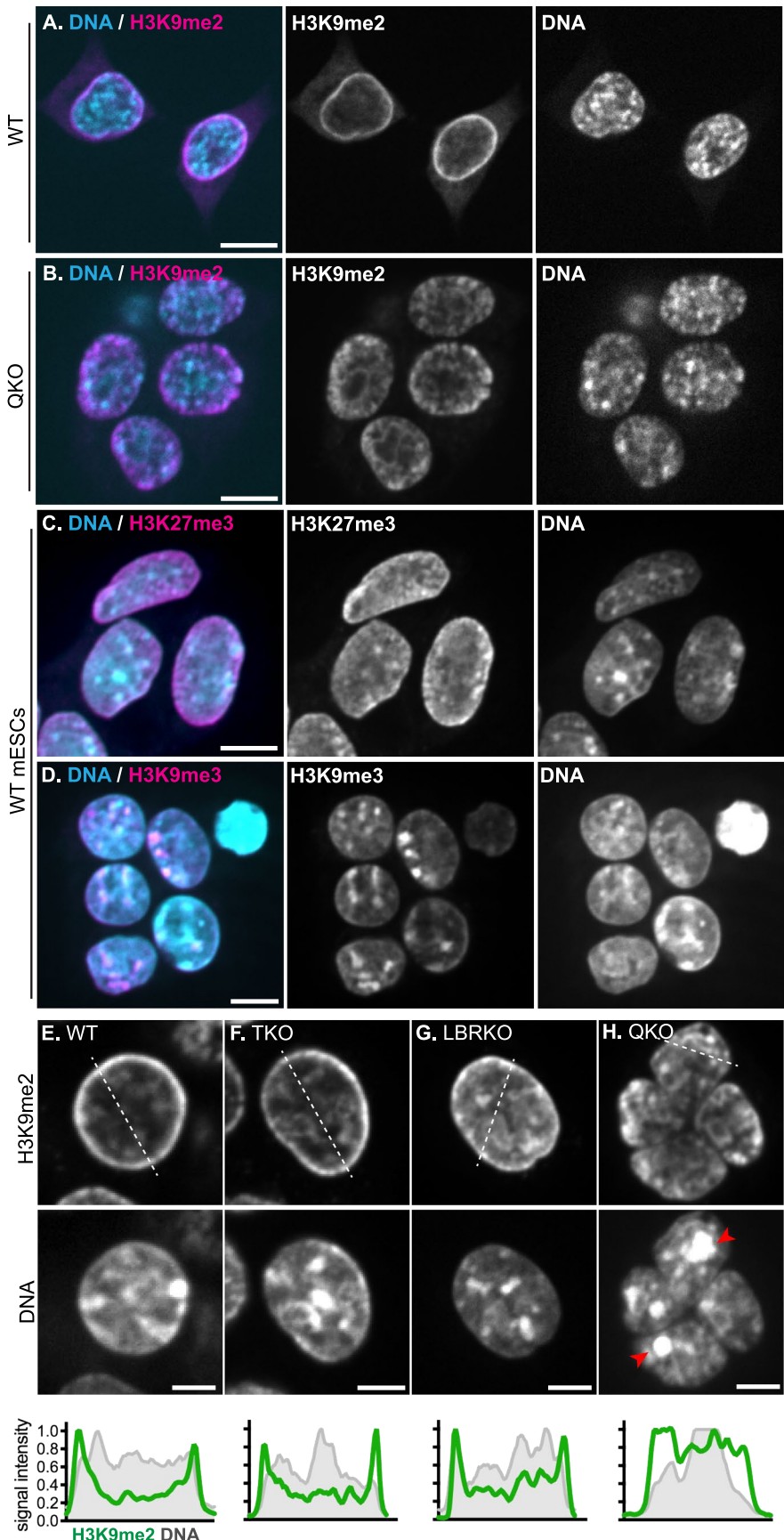

**Extended Data Fig. 1 | Immunostaining of histone modifications in mESCs.**
(**a-b**) Analysis of spatial distribution of H3K9me2 using rabbit polyclonal
antibody (Active Motif, ab39041) in WT (**a**) and QKO (**b**) mESCs. (**c-d**) Analysis of
spatial distribution of H3K27me3 (**c**) and H3K9me3 (**d**) by immunostaining in WT
mESCs. Representative images from 3 replicate experiments shown. **e-h**) Analysis
of H3K9me2 and DNA spatial distribution in WT mESCs (**e**), lamin TKO mESCs (**f**),
acute LBR CRISPR KO (**g**), and acute lamin + LBR QKO (**h**). Representative images
from a single experiment shown. Scale bar, 5 μm.

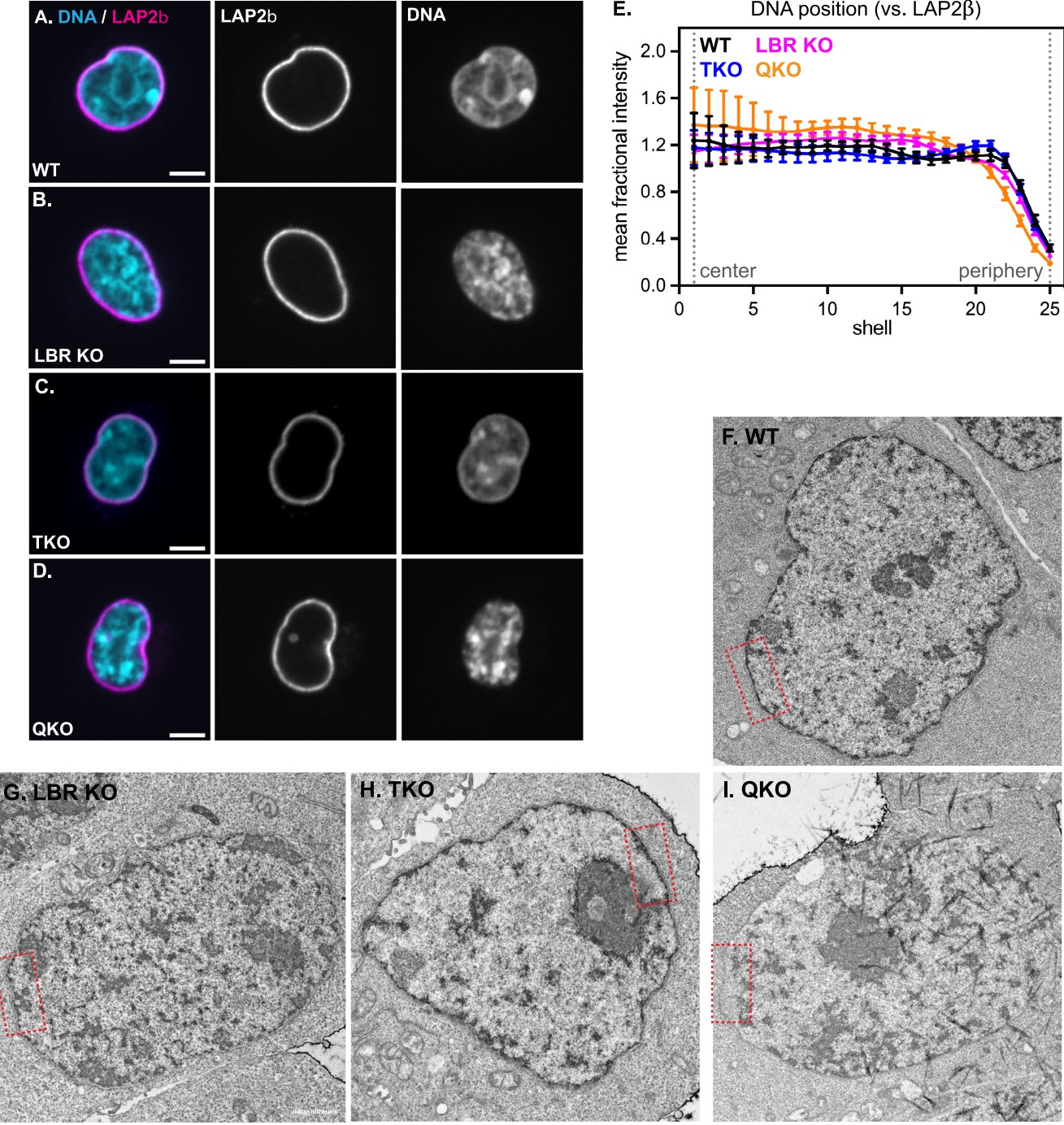

**Extended Data Fig. 2 | Microscopy of knockout mESCs.** Immunofluorescence of DNA localization (Hoechst stain) compared to the INM protein (LAP2β) in WT (**a**), LBR KO (**b**), lamin TKO (**c**), and lamin + LBR QKO (**d**) mESCs. Central z-slices (XY) are shown. Representative images from 1 replicate experiment shown; 3 replicates performed. Scale bar, 5 µm. (**e**) Radial intensity analysis of DNA position (Hoechst stain) in WT (n = 17), TKO (n = 25), LBR KO (n = 26), and QKO (n = 18) mESCs. ** p < 0.01, WT versus QKO shells 8–19, **** p < 0.0001,

WT versus QKO shells 21–25; * p < 0.05, WT versus LBR KO shells 14–18, ** p < 0.01, WT versus LBR KO shells 21–25; * p <0.05, WT versus TKO shells 18–21 by unpaired two-sided t-test. Points indicate mean and error bars indicate 95% confidence intervals. Transmission electron microscopy images showing full fields of view corresponding to Fig. 1 for WT (**f**), LBR KO (**g**), lamin TKO (**h**), and lamin + LBR QKO (**i**) mESCs. Inset positions that appear in Fig. 1 are shown in red dashed boxes (dimensions: 1.3 µm × 2.6 µm).

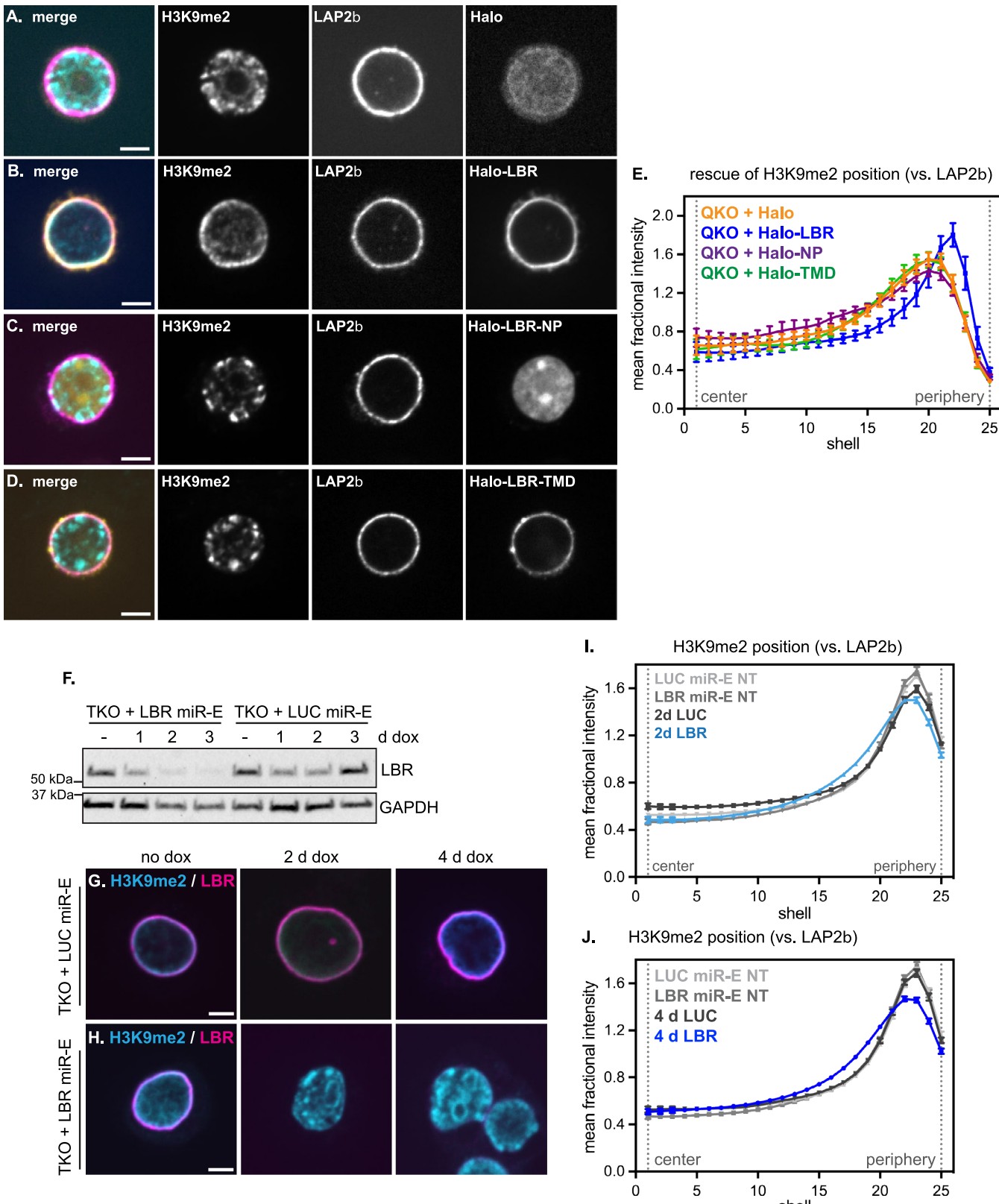

**Extended Data Fig. 3 | See next page for caption.**

**Extended Data Fig. 3 | Displacement of H3K9me2 from the nuclear periphery is reversible.** Immunofluorescence of H3K9me2 localization compared to the INM protein (LAP2β) in lamin + LBR QKO mESCS expressing Halo-NLS (**a**), Halo-LBR (**b**), Halo-LBR nucleoplasmic domain (NP) (**c**), and Halo-LBR transmembrane domain (TMD) (**d**). Central z-slices (XY) shown. Scale bar, 5 μm. (**e**) Radial intensity analysis of H3K9me2 position in QKO + Halo-NLS (n=27), QKO + Halo-LBR (n=16), QKO + Halo-TMD (n=17), and QKO + Halo-NP (n=26) 2 days after electroporation of plasmids. ** p < 0.05, Halo versus. Halo-LBR, shells 12–19; **** p < 0.0001, Halo versus. Halo-LBR, shells 22–25, unpaired two-sided t-test. ns, p > 0.05, Halo versus. Halo TMD and Halo versus. Halo NP in all shells. Points indicate mean and error bars indicate 95% confidence intervals. (**f**) Western blot showing LBR knockdown in lamin TKO mESCs expressing doxycycline-inducible LBR miR-E. Immunofluorescence of H3K9me2 (cyan) and LBR (magenta) in lamin TKO mESCs expressing LUC miR-E (**g**) or LBR miR-E (**h**). Scale bar, 5 μm. (**i**) Radial intensity analysis of H3K9me2 position in TKO + LUC miR-E untreated (NT) (n=107), TKO + LUC miR-E + dox 2d (n=140), TKO + LBR miR-E NT (n=102), TKO + LBR miR-E + dox 2d (n=83), unpaired t-test. **** p < 0.0001, 2d LBR versus 2d LUC miR-E, shells 1–13, 16–21, 23–25. **** p < 0.0001, LBR NT versus 2d LBR miR-E, shells 11–20, 22–24, unpaired two-sided t-test. Points indicate mean and error bars indicate 95% confidence intervals. (**j**) Radial intensity analysis of H3K9me2 position in TKO + LUC miR-E NT (n=107), TKO + LUC miR-E + dox 4d (n=97), TKO + LBR miR-E NT (n=102), TKO + LBR miR-E + dox 4d (n=95), unpaired two-sided t-test. **** p < 0.0001, 4d LBR versus 4d LUC miR-E, shells 14–20, 22–25. **** p < 0.0001, LBR NT versus 4d LBR miR-E, shells 4–20, 22–24, unpaired two-sided t-test. Points indicate mean and error bars indicate 95% confidence intervals.

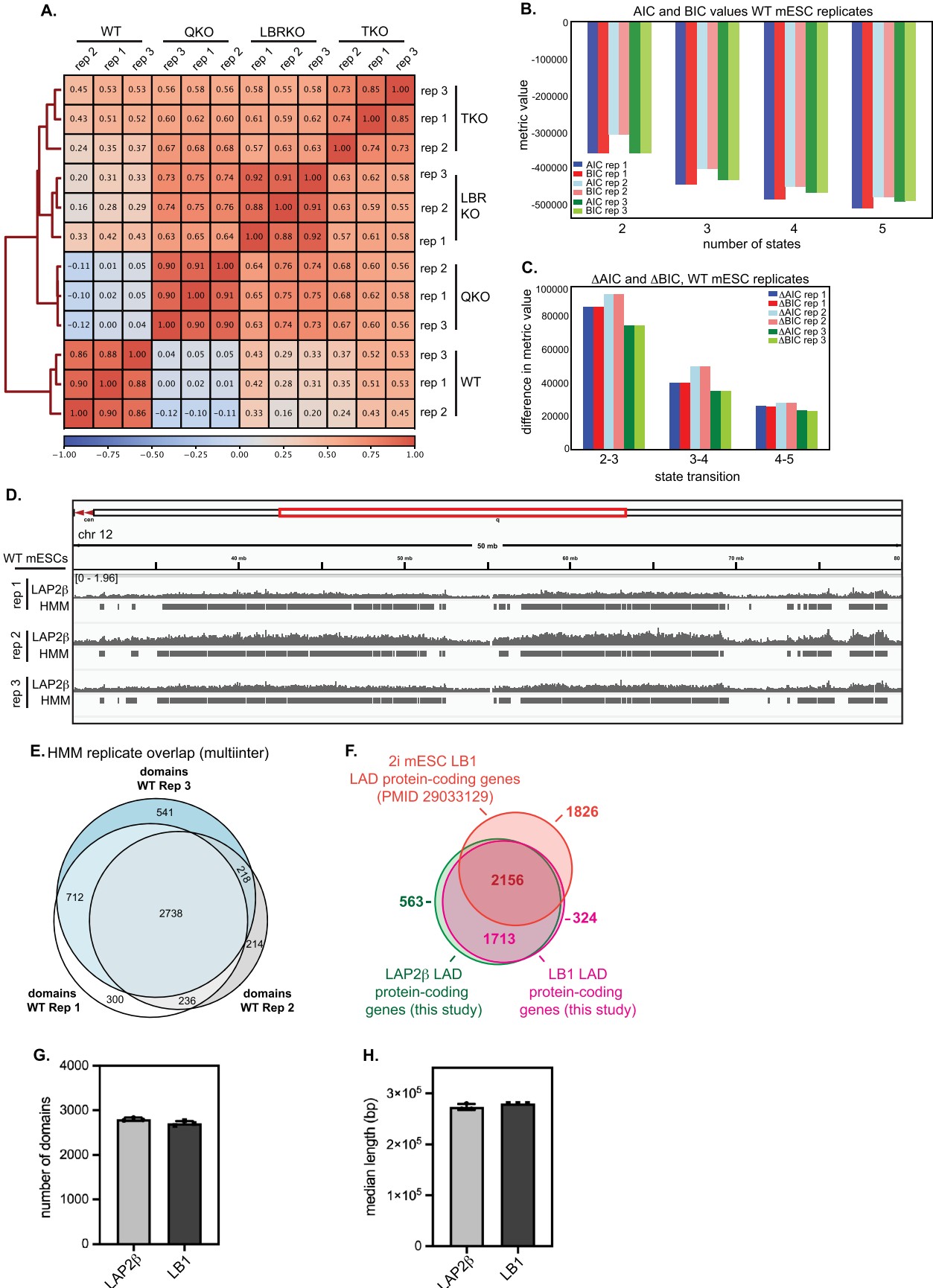

**Extended Data Fig. 4 | See next page for caption.**

**Extended Data Fig. 4 | Replicate clustering and analysis of LAP2β CUT & RUN in mESCs.** (**a**) Dendrogram and heatmap of individual LAP2β CUT & RUN replicates (3 per condition) showing similarity of replicates for each genotype. (**b**) Akaike Information Criterion (AIC) and Bayesian Information Criterion (BIC) values (**b**) and differences (**c**) between numbers of LAP2β domain states used in HMM domain calling. (**d**) Spike-in-controlled LAP2β tracks across a 50 Mb segment of chromosome 12 and corresponding 2-state HMM domain calls for 3 replicates of LAP2β CUT & RUN in WT mESCs. (**e**) Overlap between individual replicate LAP2β domains called by 2-state HMM in WT mESCs determined by multiinter package (Bedtools). (**f**) Comparison of LAP2β and LB1 domain protein-coding genes defined in this study versus previously defined LAD-resident protein-coding genes (determined by LB1 ChIP-seq) in mESCs in 2i + LIF culture conditions (PMID 29033129). (**g**) Number of LAP2β and LB1 domains identified in WT mESCs. (**h**) Genomic length of LAP2β and LB1 domains in WT mESCs.

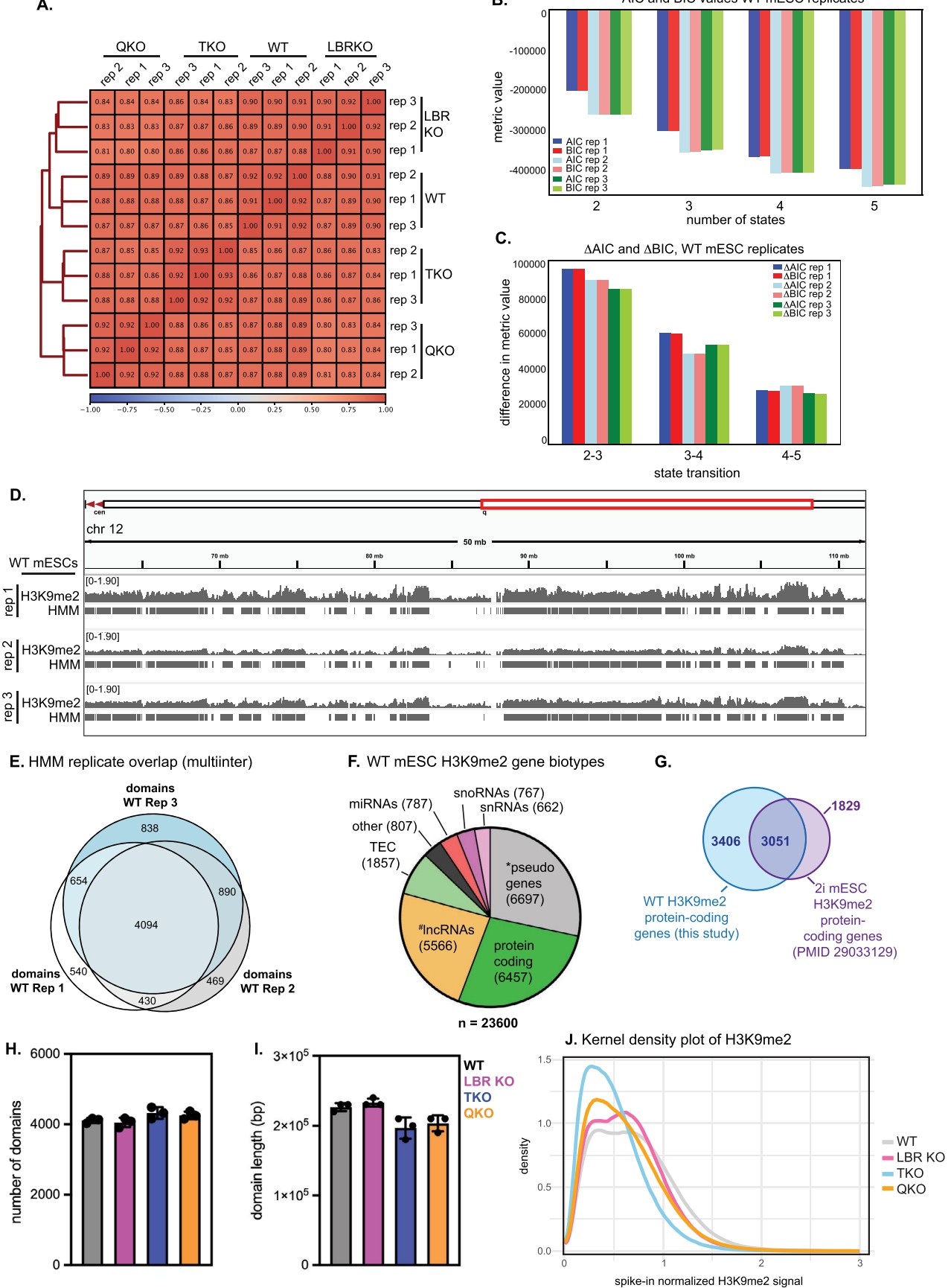

**Extended Data Fig. 5 | See next page for caption.**

**Extended Data Fig. 5 | Replicate clustering and analysis of H3K9me2 CUT & RUN in mESCs.** (**a**) Dendrogram and heatmap of individual H3K9me2 CUT & RUN replicates (3 per condition) showing similarity of replicates for each genotype. (**b**) Akaike Information Criterion (AIC) and Bayesian Information Criterion (BIC) values (**b**) and differences (**c**) between numbers of H3K9me2 domain states used in HMM domain calling. (**d**) Spike-in-controlled H3K9me2 tracks across a 50 Mb segment of chromosome 12 and corresponding 2-state HMM domain calls for 3 replicates of H3K9me2 CUT & RUN in WT mESCs. (**e**) Overlap between individual replicate H3K9me2 domains called by 2-state HMM in WT mESCs determined by multiinter package (Bedtools). (**f**) Breakdown of WT mESC H3K9me2 domain genes, defined as those genes at least 90% within an H3K9me2 domain, by gene class. # indicates that lncRNAs are significantly depleted compared to the mouse genome (two-sided $\chi^2$ test, p < 0.0001) while * indicates that pseudogenes are significantly enriched in H3K9me2 domains compared to the mouse genome (two-sided $\chi^2$ test, p < 0.0001). (**g**) Overlap of H3K9me2 domain protein-coding genes defined in this study versus in a previous list of protein-coding genes within H3K9me2 domains identified by ChiP-seq in mESCs in 2i + LIF culture conditions (PMID 29033129). (**h**) Number of H3K9me2 domains identified across genotypes. (**i**) Genomic length of H3K9me2 domains across genotypes. (**j**) Kernel density plot showing distribution of spike-in-controlled H3K8me2 signal across genotypes.

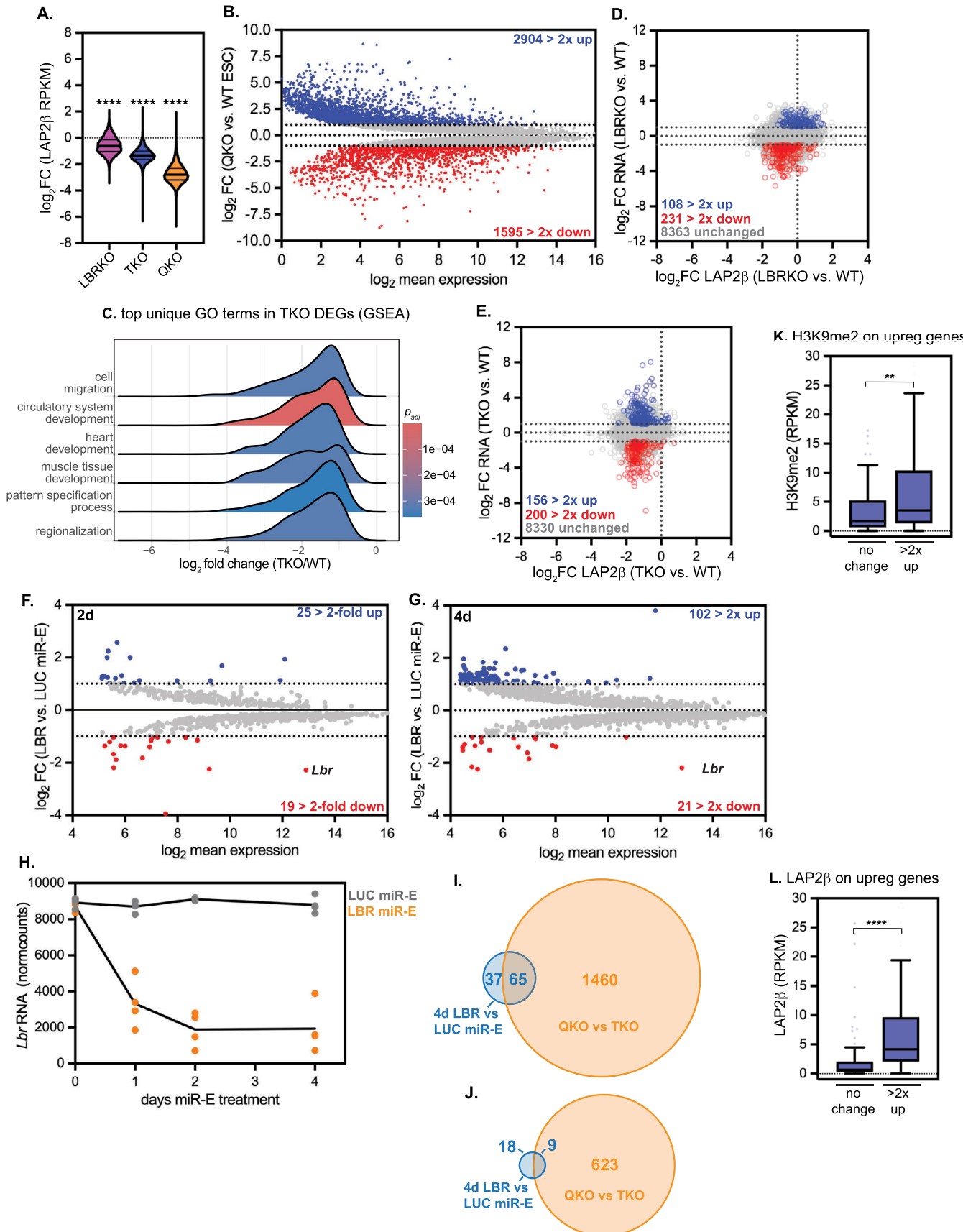

**Extended Data Fig. 6 | See next page for caption.**

**Extended Data Fig. 6 | Analysis of transcription after depletion of lamins and LBR in mESCs.** (**a**) Fold change of LAP2β contact frequency for WT LAD genes in LBRKO, TKO, and QKO mESCs. **** indicates significant deviation from 0 (p < 0.0001) by one-sample t-test. (**b**) MA plot comparing gene expression in QKO versus WT mESCs; 2904 genes are significantly upregulated by at least 2-fold ($p_{adj}$ < 0.05), while 1595 genes are significantly downregulated by at least 2-fold ($p_{adj}$ < 0.05). (**c**) Identification of top unique GO terms in genes differentially expressed between TKO and WT mESCs (GSEA). (**d-e**) Analysis of gene expression changes within WT LADs in LBRKO (**d**) and TKO (**e**) mESCs: comparison of fold change in gene expression to fold change in lamina association (LAP2β) in each indicated genotype. (**d**) in LBRKOs, 108 WT LAD genes are significantly upregulated >2-fold ($p_{adj}$ < 0.05), 231 genes are significantly downregulated >2-fold ($p_{adj}$ < 0.05), and 8363 genes exhibit no significant change in expression. (**e**) in TKOs, 156 WT LAD genes are significantly upregulated >2 fold ($p_{adj}$ < 0.05), 200 genes are significantly downregulated >2-fold ($p_{adj}$ < 0.05), and 8330 genes exhibit no significant change in expression. (**f-g**) MA plots comparing gene expression in lamin TKO mESCs expressing LBR or LUC miR-E for 2 days (**f**) or 4 days (**g**); (**f**) After 2 days treatment, 25 genes are upregulated at least 2-fold, while 19 genes are downregulated at least 2-fold. (**g**) After 4 days treatment, 102 genes are upregulated at least 2-fold, while 21 genes are downregulated at least 2-fold. (**h**) Expression of *Lbr* RNA over days of treatment with LUC or LBR miR-E. (**i-j**) Venn diagrams comparing overlap in significantly differentially expressed genes between indicated conditions. (**i**) Genes upregulated at least 2-fold both in QKO mESCs (versus TKO) and LBR miR-E (versus LUC miR-E) after 4 days treatment; (**j**) genes downregulated at least 2-fold both in QKO mESCs (versus TKO) and LBR miR-E (versus LUC miR-E) after 4 days treatment. (**k-l**) LAP2β levels (spike-in-controlled RPKM) (**k**) and H3K9me2 levels (spike-in-controlled RPKM) (**l**) in TKO mESCs on genes upregulated at least 2-fold by treatment with LBR miR-E for 4 days (n = 102) compared to a random sample of 102 genes with unchanged expression. **** indicates p <0.0001 and ** indicates p = 0.0079 by Mann-Whitney test. In box (Tukey) plots shown, center line indicates median; box limits indicate 25th to 75th percentiles; whiskers indicate 1.5x interquartile range; points indicate outlier values.

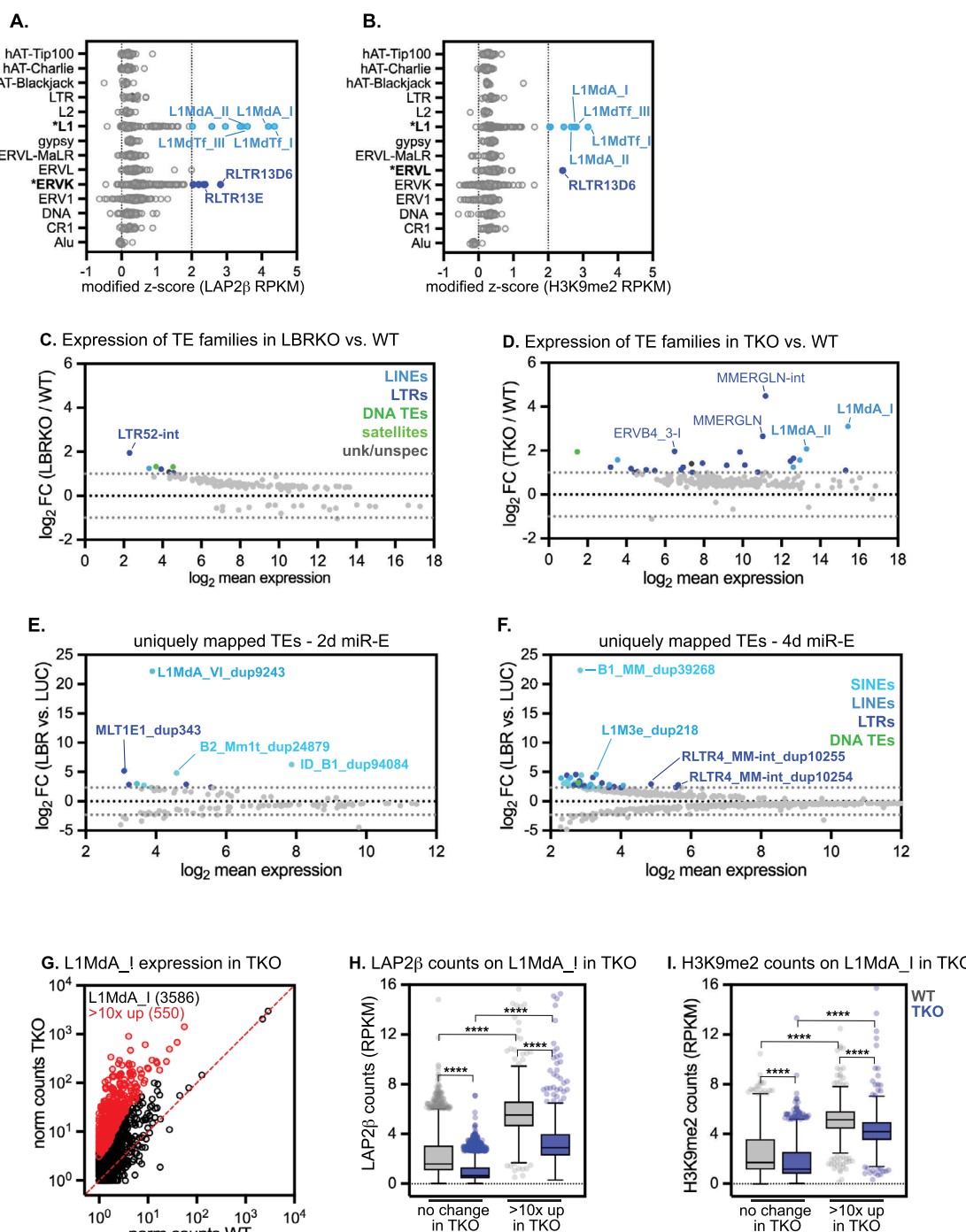

**Extended Data Fig. 7 | Analysis of TE expression. (a-b)** Identification of strongly LAD-associated (**a**) and/or H3K9me2-modified (**b**) TE families in WT mESCs. Modified z-score of spike-in-controlled LAP2β (**a**) and H3K9me2 (**b**) shown for ~1200 TE families. TE families with modified z-score >2 for H3K9me2 and/or LAP2β signal are color-coded according to class; all others shown in gray. TE classes with strong enrichment for LAP2β or H3K9me2 are marked in bold with *. See also Supplementary Table 5. (**c, d**) MA plot comparing expression of ~1200 TEs detected by TEtranscripts in (**c**) LBRKO versus WT mESCs and (**d**) TKO versus WT mESCs. All repeats without a significant change between genotypes are gray; significantly differentially expressed TEs (minimum 2-fold change) are colored correspondingly to TE family. (**e-f**) MA plots comparing expression of unique TE copies in LBR miR-E versus LUC miR-E after 2 days (**e**) or 4 days (**f**). (**e**) 116 TEs with $p_{adj}$ < 0.05 shown; 13 TEs were upregulated at least 5-fold, while 12 TEs were downregulated at least 5 fold. (**f**) 760 TEs with $p_{adj}$ < 0.05 shown;

42 TEs were upregulated at least 5-fold, while 20 TEs were downregulated at least 5-fold. All repeats without a significant change between conditions are gray; significantly differentially expressed TEs (minimum 5-fold change) are colored correspondingly to TE family. (**g**) Normalized RNA counts for 3586 uniquely mapped L1MdA_I LINE element genomic copies in TKO versus WT mESCs, all plotted as $\log_{10}$(average + 1). L1MdA_I copies with >10-fold change and significant difference in expression ($p_{adj}$ < 0.05) in are colored in red. (**h, i**) Spike-in normalized RPKM from uniquely mapped reads of LAP2β (**h**) and H3K9me2 (**i**) on L1MdA_I LINE elements with no significant change to expression versus those upregulated >10-fold in TKO mESCs. **** indicates p < 0.0001 by one-way Kruskal-Wallis multiple comparisons test with Dunn's correction. In box (Tukey) plots shown, center line indicates median; box limits indicate 25th to 75th percentiles; whiskers indicate 1.5x interquartile range; points indicate outlier values.

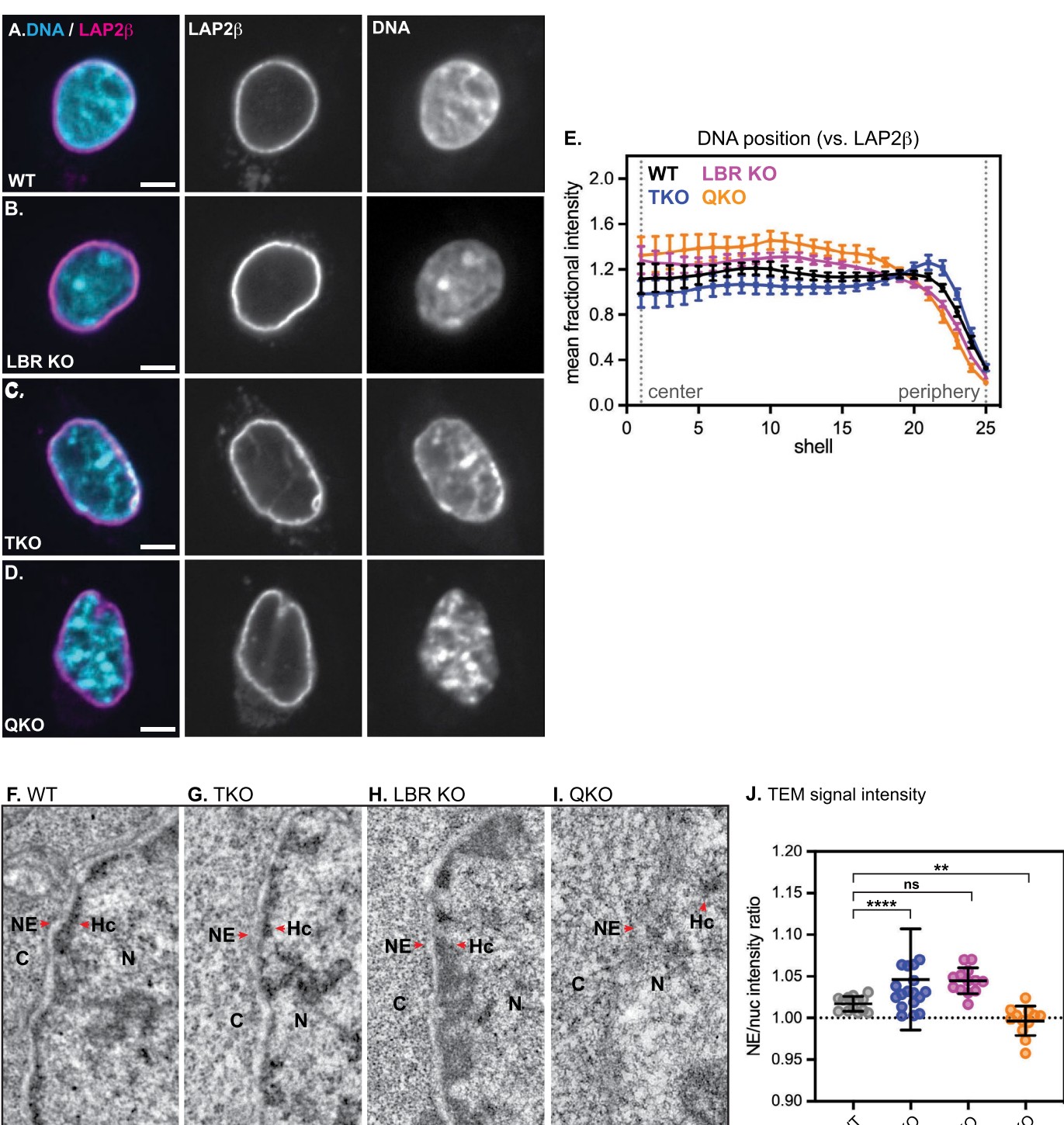

**Extended Data Fig. 8 | Fluorescence and electron microscopy of EpiLCs.**
(**a-d**) Immunofluorescence of DNA localization (Hoechst stain) compared to the
INM protein LAP2β in WT (**a**), LBR KO (**b**), lamin TKO (**c**), and lamin + LBR QKO
(**d**) EpiLCs. Scale bar, 5 μM. (**e**) Radial intensity analysis of DNA position (versus.
LAP2β) in WT (n = 26), TKO (n = 19), LBR KO (n = 38), and QKO (n = 20) EpiLCs.
** p < 0.01, WT versus QKO shells 5–18, **** p < 0.0001, WT versus QKO shells
21–25; * p < 0.05, WT versus LBR KO shells 8–10, ** p < 0.01, WT versus LBR KO
shells 11–17, *** p < 0.001, WT versus LBR KO shells 20–25; ** p < 0.01, WT versus
TKO shells 7–16, *** p < 0.001, WT versus TKO shells 21–23 by unpaired two-sided

t-test. Points indicate mean and error bars indicate 95% confidence intervals.
Transmission electron microscopy showing 1.3 μm by 2.6 μm section of the
nuclear periphery in WT (**f**), lamin TKO (**g**), LBR KO (**h**), and lamin + LBR QKO
(**i**) EpiLCs. C, cytoplasm; N, nucleus; NE, nuclear envelope; Hc, heterochromatin.
(**j**) Quantification of relative TEM signal intensity at the NE versus the
nucleoplasm for WT (n = 13), LBRKO (n = 18), TKO (n = 13), and QKO (n = 12) EpiLCs.
**** indicates $p_{adj}$ < 0.0001 for WT versus TKO; ** indicates $p_{adj}$ = 0.0066 for WT
versus QKO by one-way ANOVA followed by Dunnett's T3 multiple comparisons
test. Bars indicate mean and standard deviation.

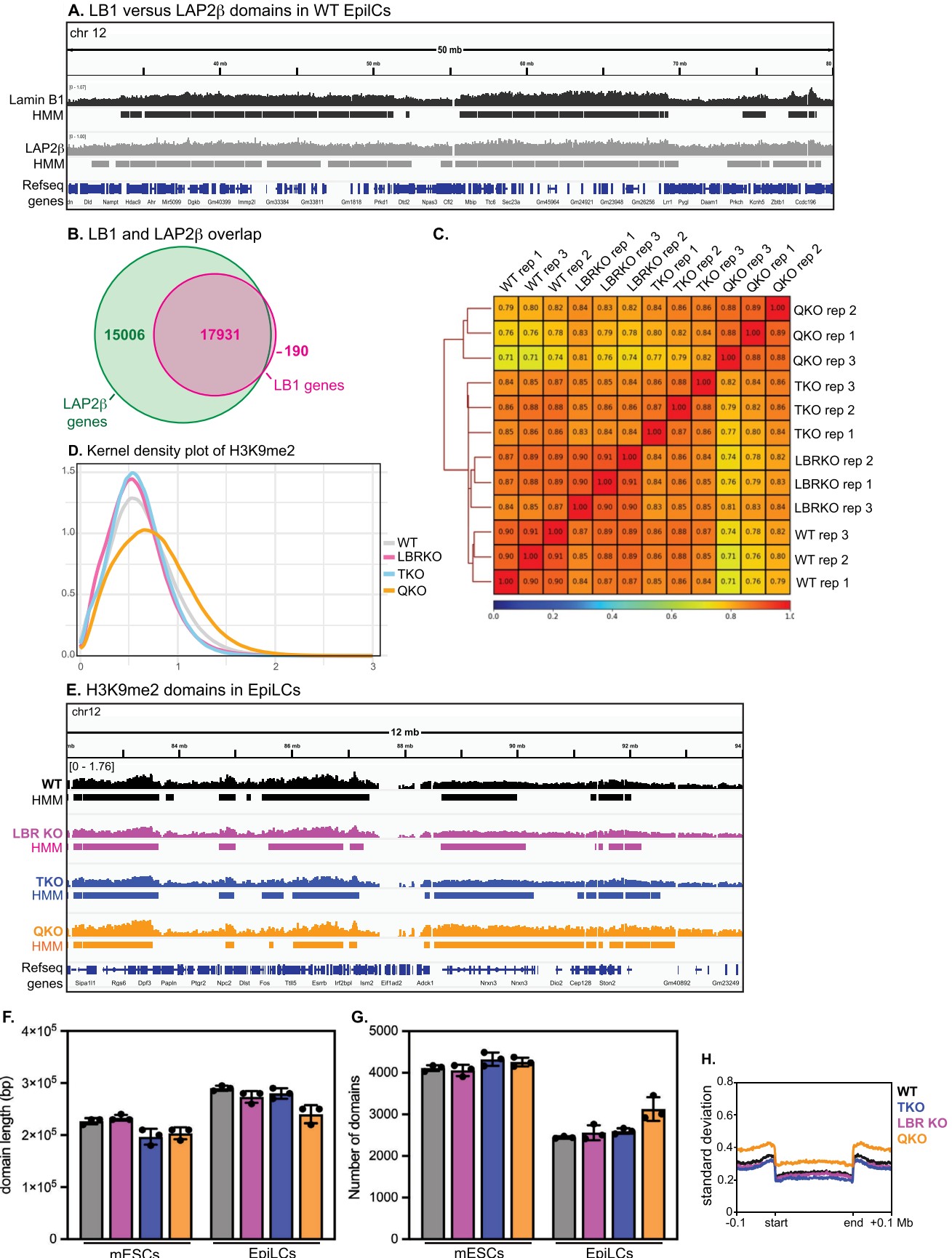

**A.** LB1 versus LAP2β domains in WT EpiLCs

**B.** LB1 and LAP2β overlap

**C.**

**D.** Kernel density plot of H3K9me2

**E.** H3K9me2 domains in EpiLCs

**F.**

**G.**

**H.**

Extended Data Fig. 9 | See next page for caption.

**Extended Data Fig. 9 | Replicate clustering and analysis of H3K9me2 Cut & Run in EpiLCs.** (**a**) Average genome tracks of Lamin B1 and LAP2β CUT & RUN in WT EpiLCs. (**b**) Overlap between LB1 and LAP2β domain loci in WT EpiLCs. (**c**) Dendrogram and heatmap of individual H3K9me2 Cut & Run replicates (3 per condition) showing similarity of replicates for each genotype. (**d**) Kernel density plot of spike-in normalized H3K9me2 signal intensity (RPKM) across genotypes.

(**e**) Average genome tracks (n=3 replicates) and domain calls for H3K9me2 in WT, LBR KO, TKO, and QKO ESCs on a 12 Mb section of chromosome 12. Y-axis range indicated at top left is the same for all tracks shown. (**f**) Size of H3K9me2 domains in each genotype. (**g**) Number of H3K9me2 domains called in each genotype. (**h**) Standard deviation of H3K9me2 signal in WT EpiLC domains.

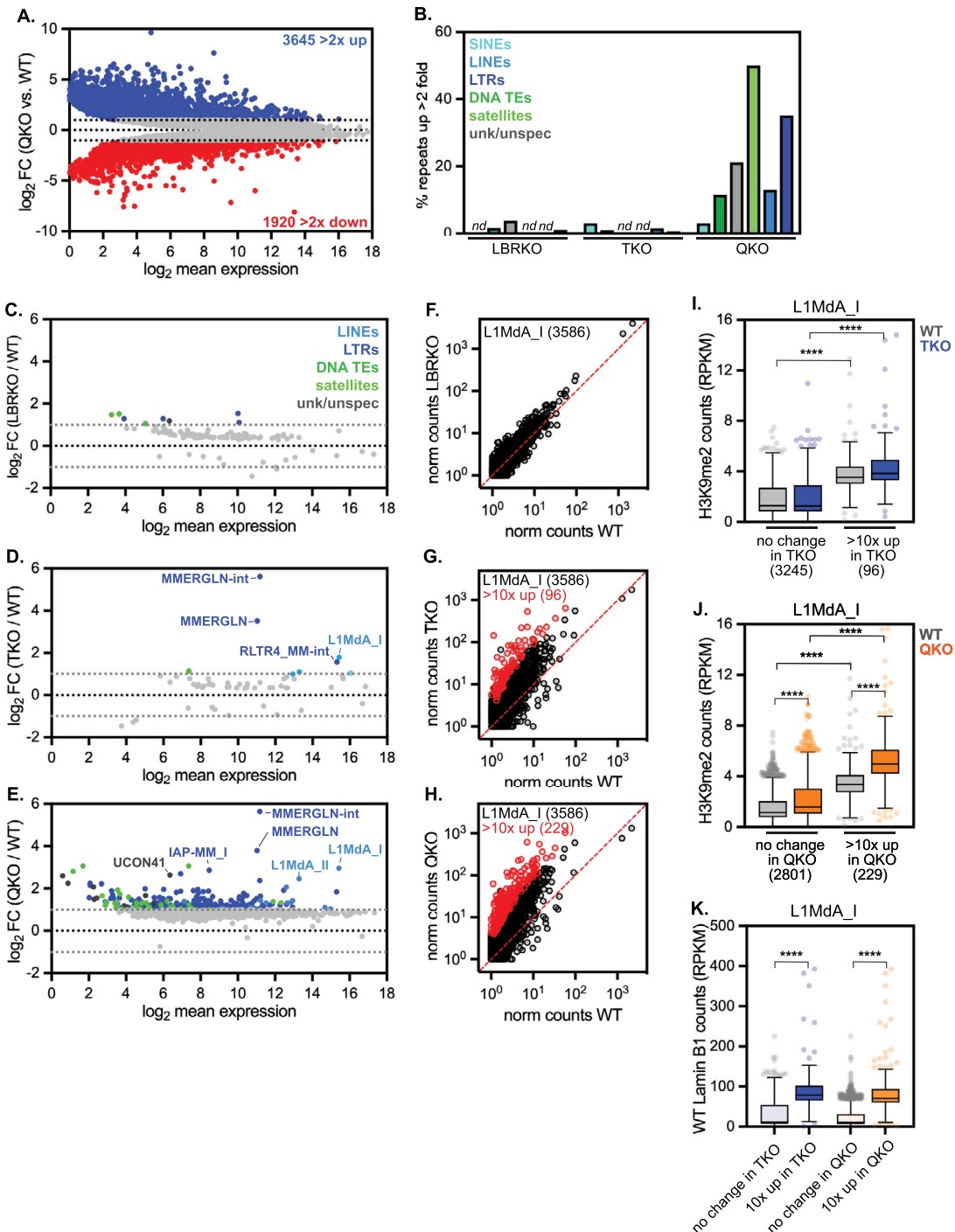

**Extended Data Fig. 10 | Analysis of transcription after depletion of lamins and LBR in EpiLCs.** (**a**) MA plot comparing gene expression in QKO *versus* TKO EpiLCs. 13188 genes with a minimum $p_{adj}$ of 0.05 shown; 3645 genes are upregulated at least 2-fold, while 1920 genes are downregulated at least 2-fold. (**b**) Summary of TEs upregulated >2-fold in LBRKO, TKO, and QKO EpiLCs compared to WT EpiLCs. (**c-e**) MA plots of TE expression in mutant EpiLCs. (**c**) 119 TEs in LBRKO EpiLCs, (**d**) 56 TEs in TKO EpiLCs, and (**e**) 950 TEs in QKO EpiLCs with p adj < 0.05 shown; TEs upregulated at least 2-fold are colored according to TE family. (**f-h**) Normalized counts for 3586 uniquely mapped L1MdA_I LINE element genomic copies in (**f**) LBRKO versus WT mESCs, (**g**) TKO versus WT mESCs, and (**h**) QKO versus WT mESCs (plotted as $\log_{10}$(average + 1)). L1MdA_I copies with

>10-fold change and significant difference in expression ($p_{adj}$ < 0.05) in are colored in red. (**i-j**) Spike-in-controlled RPKM from uniquely mapped reads of H3K9me2 on L1MdA_I LINE elements with unchanged expression versus those upregulated >10-fold in TKO mESCs (**i**, n = 96) and in QKO mESCs (**j**, n = 229). (**k**) Spike-in-controlled RPKM from uniquely mapped reads of Lamin B1 in WT EpiLCs on L1MdA_I LINE elements with unchanged expression versus those upregulated >10-fold in TKO mESCs (n = 96) and in QKO mESCs (n = 229). **** indicates p < 0.0001 by Kruskal-Wallis multiple comparisons test with Dunn's correction. Box (Tukey) plot center line indicates median; box limits indicate 25th to 75th percentiles; whiskers indicate 1.5x interquartile range; points indicate outlier values.

# Reporting Summary

## Statistics

For all statistical analyses, confirm that the following items are present in the figure legend, table legend, main text, or Methods section.

| n/a | Confirmed | |
|---|---|---|
| ☐ | ☒ | The exact sample size (*n*) for each experimental group/condition, given as a discrete number and unit of measurement |
| ☒ | ☐ | A statement on whether measurements were taken from distinct samples or whether the same sample was measured repeatedly |
| ☐ | ☒ | The statistical test(s) used AND whether they are one- or two-sided *Only common tests should be described solely by name; describe more complex techniques in the Methods section.* |
| ☒ | ☐ | A description of all covariates tested |
| ☐ | ☒ | A description of any assumptions or corrections, such as tests of normality and adjustment for multiple comparisons |
| ☐ | ☒ | A full description of the statistical parameters including central tendency (e.g. means) or other basic estimates (e.g. regression coefficient) AND variation (e.g. standard deviation) or associated estimates of uncertainty (e.g. confidence intervals) |
| ☐ | ☒ | For null hypothesis testing, the test statistic (e.g. *F*, *t*, *r*) with confidence intervals, effect sizes, degrees of freedom and *P* value noted *Give P values as exact values whenever suitable.* |
| ☒ | ☐ | For Bayesian analysis, information on the choice of priors and Markov chain Monte Carlo settings |
| ☒ | ☐ | For hierarchical and complex designs, identification of the appropriate level for tests and full reporting of outcomes |
| ☒ | ☐ | Estimates of effect sizes (e.g. Cohen's *d*, Pearson's *r*), indicating how they were calculated |

*Our web collection on statistics for biologists contains articles on many of the points above.*

## Software and code

Policy information about availability of computer code

| Data collection | The Nikon Elements software suite was used to acquire confocal microscopy data. |
|---|---|
| Data analysis | CellProfiler and Fiji open-source software were used for microscopy image analysis, and R and Graphpad Prism were used for further analysis and presentation of analyzed data. Standard R and Python packages and pipelines were used for analysis of transcriptomic and genomic data. Where previously published bioinformatic packages were used, citations are provided. |

For manuscripts utilizing custom algorithms or software that are central to the research but not yet described in published literature, software must be made available to editors and reviewers. We strongly encourage code deposition in a community repository (e.g. GitHub). See the Nature Portfolio guidelines for submitting code & software for further information.

## Data

Policy information about availability of data

All manuscripts must include a data availability statement. This statement should provide the following information, where applicable:
- Accession codes, unique identifiers, or web links for publicly available datasets
- A description of any restrictions on data availability
- For clinical datasets or third party data, please ensure that the statement adheres to our policy

Genomic and transcriptomic data generated during the course of this study were uploaded to the GEO database under the following accession numbers: mESC and EpiLC RNAseq data, GSE264599; LBR miR-E RNAseq data, GSE264602;  H3K9me2 Cut & Run data, GSE264603; and LAP2b and LB1 Cut & R un data, GSE288122.

# Research involving human participants, their data, or biological material

Policy information about studies with human participants or human data. See also policy information about sex, gender (identity/presentation), and sexual orientation and race, ethnicity and racism.

| | |
|---|---|
| Reporting on sex and gender | n/a |
| Reporting on race, ethnicity, or other socially relevant groupings | n/a |
| Population characteristics | n/a |
| Recruitment | n/a |
| Ethics oversight | n/a |

Note that full information on the approval of the study protocol must also be provided in the manuscript.

# Field-specific reporting

Please select the one below that is the best fit for your research. If you are not sure, read the appropriate sections before making your selection.

☒ Life sciences          ☐ Behavioural & social sciences          ☐ Ecological, evolutionary & environmental sciences

For a reference copy of the document with all sections, see nature.com/documents/nr-reporting-summary-flat.pdf

# Life sciences study design

All studies must disclose on these points even when the disclosure is negative.

| | |
|---|---|
| Sample size | No power analyses were performed to predetermine sample size. For microscopy analyses, experiments were performed in at least 3 independent biological replicates with two technical replicates per experiment. For each biological replicate, numerous individual cells were analyzed and each cell is reported as an independent data point. For confocal microscopy experiments at least 10 cells were analyzed per biological replicate.<br>For differentiation experiments, at least 3 biological replicates and 2 technical replicates were performed.<br>For bulk RNAseq analyses, 4 replicates were performed. For Cut & Run analyses, 3 or 4 replicates were performed.<br>These sample sizes align with field standards. |
| Data exclusions | No data were excluded from any experiments included in this study. |
| Replication | All replicates of experiments produced consistent outcomes, and we had no issues with reproducing any findings shown in the manuscript in multiple independent experiments. A minimum of 3 replicates were performed. |
| Randomization | N/A. We did not use categorical measurements or scoring that would benefit from randomization. |
| Blinding | N/A. We did not use categorical measurements or subjective scoring that would benefit from blinding. Instead, we report quantitative measurements. |

# Reporting for specific materials, systems and methods

We require information from authors about some types of materials, experimental systems and methods used in many studies. Here, indicate whether each material, system or method listed is relevant to your study. If you are not sure if a list item applies to your research, read the appropriate section before selecting a response.

## Materials & experimental systems

| n/a | Involved in the study |
|---|---|
| ☐ | ☒ Antibodies |
| ☐ | ☒ Eukaryotic cell lines |
| ☒ | ☐ Palaeontology and archaeology |
| ☒ | ☐ Animals and other organisms |
| ☒ | ☐ Clinical data |
| ☒ | ☐ Dual use research of concern |
| ☒ | ☐ Plants |

## Methods

| n/a | Involved in the study |
|---|---|
| ☐ | ☒ ChIP-seq |
| ☒ | ☐ Flow cytometry |
| ☒ | ☐ MRI-based neuroimaging |

# Antibodies

| | |
|---|---|
| Antibodies used | H3K9me2 antibody (Abcam ab1220) was used for immunostaining and for Cut & Run.<br>H3K9me2 antibody (Active Motif 39041) was also used for immunostaining.<br>LAP2 antibody (Invitrogen PA5-52519) and LBR antibody (Abcam ab232731) were used for immunostaining.<br>SOX2 antibody (Rockland 610-1302) was used for Western blotting. Orf1p antibody (Abcam ab216324) was used for immunostaining. |
| Validation | The specificity of the H3K9me2 ab1220 antibody for H3K9me2 by ChIP-seq assays and of the H3K9me2 Active Motif antibody for immunostaining assays has been previously demonstrated (using competitor peptides) by Andrey Poleshko, Raj Jain, and colleagues (see Poleshko et al., Cell 2017 and eLife 2019). In addition, we verified that ab1220 and Active Motif H3K9me2 immunostaining signal disappears when cells are treated with the G9a/GLP inhibitor UNC0638.<br>The LAP2 antibody recognizes both the nucleoplasmic alpha and nuclear membrane-inserted beta isoforms, although the latter is much more readily detected by this antibody in mESCs. We have verified that LAP2 immunostaining at the nuclear periphery diminishes when LAP2b expression is inhibited by RNAi in human cells.<br>We verified the specificity of the LBR antibody by noting its lack of signal in LBR-null and LBR-RNAi conditions. |

# Eukaryotic cell lines

Policy information about cell lines and Sex and Gender in Research

| | |
|---|---|
| Cell line source(s) | lamin TKO mESCs and wildtype littermate mESCs were the generous gift of Yixian Zheng and are described in their publication (Zheng et al., Mol Cell 2018). These mESCs are XY. |
| Authentication | wildtype and lamin TKO mESC cell lines were originally authenticated by the Zheng laboratory. We independently authenticated these lines by performing qPCR to validate lack of LMNA, LMNB1, and LMNB2 expression in lamin TKO mESCs. LBR-null and lamin + LBR quadruple null mESCs were genotyped by PCR and Sanger sequencing, and lack of expression of detectable LBR was confirmed by immunostaining with multiple LBR antibodies.<br>Wild type, lamin TKO, and lamin + LBR QKO mESCs were karyotyped. These lines exhibited some karyotypic abnormalities that are frequently observed in stem cells which are reported in the Methods. |
| Mycoplasma contamination | wild type and lamin TKO mESCs were tested and confirmed to be mycoplasma negative. |
| Commonly misidentified lines (See ICLAC register) | none used. |

# Plants

| | |
|---|---|
| Seed stocks | n/a |
| Novel plant genotypes | n/a |
| Authentication | n/a |

# ChIP-seq

## Data deposition

☒ Confirm that both raw and final processed data have been deposited in a public database such as GEO.

☒ Confirm that you have deposited or provided access to graph files (e.g. BED files) for the called peaks.

| | |
|---|---|
| Data access links<br>*May remain private before publication.* | Here are three reviewer tokens for the three GEO entries:<br>https://www.ncbi.nlm.nih.gov/geo/query/acc.cgi?acc=GSE264599 token: ijuhaqkqlvadvij<br>https://www.ncbi.nlm.nih.gov/geo/query/acc.cgi?acc=GSE264602 token: ebofuiisxvqtvqx<br>https://www.ncbi.nlm.nih.gov/geo/query/acc.cgi?acc=GSE264603 token: mzgbkyworpkdjkn |
| Files in database submission | raw files:<br>LBRKO_epi_1_R1_001.fastq.gz<br>LBRKO_epi_1_R2_001.fastq.gz<br>LBRKO_epi_2_R1_001.fastq.gz<br>LBRKO_epi_2_R2_001.fastq.gz<br>LBRKO_epi_3_R1_001.fastq.gz<br>LBRKO_epi_3_R2_001.fastq.gz<br>LBRKO_naive_1_R1_001.fastq.gz |

LBRKO_naive_1_R2_001.fastq.gz
LBRKO_naive_2_R1_001.fastq.gz
LBRKO_naive_2_R2_001.fastq.gz
LBRKO_naive_3_R1_001.fastq.gz
LBRKO_naive_3_R2_001.fastq.gz
QKO_epi_1_R1_001.fastq.gz
QKO_epi_1_R2_001.fastq.gz
QKO_epi_2_R1_001.fastq.gz
QKO_epi_2_R2_001.fastq.gz
QKO_epi_3_R1_001.fastq.gz
QKO_epi_3_R2_001.fastq.gz
QKO_naive_1_R1_001.fastq.gz
QKO_naive_1_R2_001.fastq.gz
QKO_naive_2_R1_001.fastq.gz
QKO_naive_2_R2_001.fastq.gz
QKO_naive_3_R1_001.fastq.gz
QKO_naive_3_R2_001.fastq.gz
TKO_epi_1_R1_001.fastq.gz
TKO_epi_1_R2_001.fastq.gz
TKO_epi_2_R1_001.fastq.gz
TKO_epi_2_R2_001.fastq.gz
TKO_epi_3_R1_001.fastq.gz
TKO_epi_3_R2_001.fastq.gz
TKO_naive_1_R1_001.fastq.gz
TKO_naive_1_R2_001.fastq.gz
TKO_naive_2_R1_001.fastq.gz
TKO_naive_2_R2_001.fastq.gz
TKO_naive_3_R1_001.fastq.gz
TKO_naive_3_R2_001.fastq.gz
WT_epi_1_R1_001.fastq.gz
WT_epi_1_R2_001.fastq.gz
WT_epi_2_R1_001.fastq.gz
WT_epi_2_R2_001.fastq.gz
WT_epi_3_R1_001.fastq.gz
WT_epi_3_R2_001.fastq.gz
WT_naive_1_R1_001.fastq.gz
WT_naive_1_R2_001.fastq.gz
WT_naive_2_R1_001.fastq.gz
WT_naive_2_R2_001.fastq.gz
WT_naive_3_R1_001.fastq.gz
WT_naive_3_R2_001.fastq.gz

processed files:
LBRKO_epi_1_RPKM_10kb.bw
LBRKO_epi_1_RPKM_1kb.bw
LBRKO_epi_2_RPKM_10kb.bw
LBRKO_epi_2_RPKM_1kb.bw
LBRKO_epi_3_RPKM_10kb.bw
LBRKO_epi_3_RPKM_1kb.bw
LBRKO_naive_1_RPKM_10kb.bw
LBRKO_naive_1_RPKM_1kb.bw
LBRKO_naive_2_RPKM_10kb.bw
LBRKO_naive_2_RPKM_1kb.bw
LBRKO_naive_3_RPKM_10kb.bw
LBRKO_naive_3_RPKM_1kb.bw
QKO_epi_1_RPKM_10kb.bw
QKO_epi_1_RPKM_1kb.bw
QKO_epi_2_RPKM_10kb.bw
QKO_epi_2_RPKM_1kb.bw
QKO_epi_3_RPKM_10kb.bw
QKO_epi_3_RPKM_1kb.bw
QKO_naive_1_RPKM_10kb.bw
QKO_naive_1_RPKM_1kb.bw
QKO_naive_2_RPKM_10kb.bw
QKO_naive_2_RPKM_1kb.bw
QKO_naive_3_RPKM_10kb.bw
QKO_naive_3_RPKM_1kb.bw
TKO_epi_1_RPKM_10kb.bw
TKO_epi_1_RPKM_1kb.bw
TKO_epi_2_RPKM_10kb.bw
TKO_epi_2_RPKM_1kb.bw
TKO_epi_3_RPKM_10kb.bw
TKO_epi_3_RPKM_1kb.bw
TKO_naive_1_RPKM_10kb.bw
TKO_naive_1_RPKM_1kb.bw
TKO_naive_2_RPKM_10kb.bw

TKO_naive_2_RPKM_1kb.bw
TKO_naive_3_RPKM_10kb.bw
TKO_naive_3_RPKM_1kb.bw
WT_epi_1_RPKM_10kb.bw
WT_epi_1_RPKM_1kb.bw
WT_epi_2_RPKM_10kb.bw
WT_epi_2_RPKM_1kb.bw
WT_epi_3_RPKM_10kb.bw
WT_epi_3_RPKM_1kb.bw
WT_naive_1_RPKM_10kb.bw
WT_naive_1_RPKM_1kb.bw
WT_naive_2_RPKM_10kb.bw
WT_naive_2_RPKM_1kb.bw
WT_naive_3_RPKM_10kb.bw
WT_naive_3_RPKM_1kb.bw
cutandrun_counts_TE_unique.txt
cutandrun_counts_genes.txt

Genome browser session
(e.g. UCSC)

n/a

## Methodology

**Replicates**

Three replicates of each genotype were performed for Cut & Run analyses. All replicates clustered together in unsupervised hierarchical clustering, and these data are reported in Supplementary Figures 4 and 9.

**Sequencing depth**

For Cut & Run, approximately 16 million reads were sequenced per library (paired-end sequencing, 35bp read length)
For RNAseq, approximately 30 million reads were sequenced per library (paired-end sequencing, 100 bp read length).

**Antibodies**

H3K9me2 antibody ab1220 from Abcam and rabbit anti-mouse secondary antibody (Abcam ab6709) wer used for Cut & Run experiments.

**Peak calling parameters**

Cut & Run reads were mapped using Bowtie2 and filtered with SAMtools. H3K9me2 domains were called using a Hidden Markov Model approach via the Pomegranate Python package.

**Data quality**

Cut & Run data has a significantly higher dynamic range and lower background than ChIP-seq data, and fold-enrichment cutoffs or FDR cutoffs are less frequently needed for this type of data. (see work from Henikoff & colleagues in eLife 2017 and Nat Comms 2019.)

**Software**

Cut & Run data were mapped and analyzed using published bioinformatic pipelines and packages in R and Python; previously published packages used are cited in the Methods section.

