## [Peer Review File · Nature Cell Biology]

The nuclear periphery confers repression on H3K9me2-marked genes and transposons to shape cell fate

Corresponding Author: Dr Abigail Buchwalter

Version 0:

Decision Letter:

Revise extended OD

*Please delete the link to your author homepage if you wish to forward this email to co-authors.

Dear Dr Buchwalter,

Your manuscript, "The nuclear periphery confers repression on H3K9me2-marked genes and transposons to shape cell fate", has now been seen by 3 referees, who are experts in nuclear architecture (referee 1); spatial organisation of the genome (referee 2); and gene regulation (referee 3). As you will see from their comments (attached below) they find this work of potential interest, but have raised substantial concerns, which in our view would need to be addressed with considerable revisions before we can consider publication in Nature Cell Biology.

Nature Cell Biology editors discuss the referee reports in detail within the editorial team, including the chief editor, to identify key referee points that should be addressed with priority, and requests that are overruled as being beyond the scope of the current study. To guide the scope of the revisions, I have listed these points below. We are committed to providing a fair and constructive peer-review process, so please feel free to contact me if you would like to discuss any of the referee comments further.

In particular, it would be essential to:

- A- Perform further experiments to distinguishing between two possible models of chromatin repositioning at the nuclear periphery (Reviewer#1 pt 1, Reviewer#3 major pt 7)
- B- Clarify the rationale behind using the distribution of H3K9me2 as the major readout for the experiments and validate the choice of experimental readout (Reviewer#2 pt 1), as well as the choice of the 4-state HMM model (Reviewer#3 major pts 1,2)
- C- Further experimentally investigate the release of H3K9me2 from the nuclear envelope (Reviewer#2 pt 2, reviewer#3 pt 1)
- D- Perform further analysis of the gene expression analysis as per the recommendations (Reviewer#1 pt 3, Reviewer#3 major pt4) .

- All other referee concerns pertaining to strengthening existing data, providing controls (Reviewer#1 pt 2, Reviewer#3), methodological details, clarifications and textual changes, should also be addressed.

- Finally please pay close attention to our guidelines on statistical and methodological reporting (listed below) as failure to do so may delay the reconsideration of the revised manuscript. In particular please provide:

We would be happy to consider a revised manuscript that would satisfactorily address these points, unless a similar paper is published elsewhere, or is accepted for publication in Nature Cell Biology in the meantime.

- ensure that it conforms to our format instructions and publication policies (see below and www.nature.com/nature/authors/).

- provide a point-by-point rebuttal to the full referee reports verbatim, as provided at the end of this letter.

- provide the completed Editorial Policy Checklist (found here <https://www.nature.com/authors/policies/Policy.pdf>), and Reporting

Summary (found here https://www.nature.com/authors/policies/ReportingSummary.pdf). This is essential for reconsideration of the manuscript and these documents will be available to editors and referees in the event of peer review. For more information see http://www.nature.com/authors/policies/availability.html or contact me.

Nature Cell Biology is committed to improving transparency in authorship. As part of our efforts in this direction, we are now requesting that all authors identified as 'corresponding author' on published papers create and link their Open Researcher and Contributor Identifier (ORCID) with their account on the Manuscript Tracking System (MTS), prior to acceptance. ORCID helps the scientific community achieve unambiguous attribution of all scholarly contributions. You can create and link your ORCID from the home page of the MTS by clicking on 'Modify my Springer Nature account'. For more information please visit please visit www.springernature.com/orcid.

Link Redacted

We would like to receive a revised submission within six months. We would be happy to consider a revision even after this timeframe, however if the resubmission deadline is missed and the paper is eventually published, the submission date will be the date when the revised manuscript was received.

We hope that you will find our referees' comments, and editorial guidance helpful. Please do not hesitate to contact me if there is anything you would like to discuss.

Best wishes,

Sabrya Carim

Sabrya Carim, PhD
(she/her/hers)
Associate Editor, Nature Cell Biology
Nature Portfolio

Springer Nature
The Campus, 4 Crinan Street, London N1 9XW, UK
sabrya.carim@springernature.com
<https://orcid.org/0000-0001-9485-1938>

Reviewers' Comments:

Reviewer #1:

Remarks to the Author:

Marin and colleagues study the role of lamin proteins and lamin B receptor (LBR) in positioning of chromatin at the nuclear periphery. They perform a series of genomic assays to map chromatin enriched for H3K9me2 in a variety of genetic manipulations. They also complement this with expression analysis using RNA-seq. H3K9me2 marked chromatin is found at the nuclear periphery in WT cells and genes at the nuclear periphery are known to be predominantly repressed. The authors find that knockout of LBR and three lamin proteins results in a reduction in the amount of chromatin at the nuclear periphery, suggesting that spatial positioning and H3K9me2 status can be unlinked. They also find that genes can be active and retain H3K9me2 in the lamin/LBR depleted conditions. Finally, they differentiate ESCs into epiblast-like cells to understand the effect of lamin and LBR depletion on differentiation.

The rules underlying epigenetic control of 3D genome organization and effects on genome function are of wide interest, especially with relation to cellular identity/cell fate. While the work is of interest, the rationale for comparisons can be hard to follow at times. In addition, multiple conceptual and technical points should be addressed before the manuscript can be published so that the impact of the work is a bit clearer. I have provided my major comments in four major themes:

1. The authors present a model where LBR and lamin proteins regulate positioning of chromatin at the nuclear periphery. A fraction of genes retain H3K9me2 and their expression is dysregulated. However, there still appears to be H3K9me2 marked chromatin at the nuclear periphery in the lamin + LBR KOs according to the imaging (Fig 1D, H). This raises the question whether H3K9me2-marked chromatin is repositioned from the periphery (and if so, to what extent). This can be addressed by measuring the distance of candidate loci (nominated from their datasets) from the nuclear envelope using DNA FISH. The authors would also be able to determine whether individual DNA FISH probes gain volume in relation to their distance to the periphery across conditions using this approach - allowing them to determine whether lamin and LBR mediate positioning and/or compaction status.

Relatedly, the authors should also determine measure the distance of bursting candidate loci (using intronic RNA FISH) to the nuclear envelope. While the authors suggest that movement away from the periphery precedes transcriptional activity, it is possible that lamin and LBR induce transcriptional silencing of loci at the periphery. If that is the case, KO of lamins and LBR will abrogate the silencing and the subsequent transcriptional activity may facilitate locus movement (and that of nearby regions/H3K9me2 domains) to the nuclear interior (consistent with PMID 25477464). Distinguishing between these two models would be very impactful and helpful for the field.

2. Additional controls are required to properly interpret the changes in H3K9me2 and related CUT and RUN data. Firstly, the conclusions would be more convincing if the authors showed how the differential areas behave across replicates to ensure the replicates are closely aligned and the domain caller is accurate. For example, while the majority of genes identified in H3K9me2 domains in PMID 2903319 are also identified as in a H3K9me2 domain by the authors (Fig S4F), 23,114 other genes are not identified in a H3K9me2 domain by the other publication (PMID 2903319).

Second, western blots of H3K9me2 across all the various conditions and cell types are necessary to support that H3K9me2 levels are not changed in ESCs across genetic KO and that H3K9me2 levels increase upon differentiation.

3. The expression analysis could benefit from a more focused approach and additional analysis, as there is likely a mix of primary and secondary effects observed in the data as they are presented currently. Firstly, are the same genes which are upregulated in the TKO versus WT also upregulated upon LBR KO or are they distinct? How does this compare to those identified as upregulated in the QKO? A similar analysis should be performed for down regulated genes. Secondly, the conclusions focusing on any role of H3K9me2 would be better supported if the authors focused on those genes in H3K9me2 domains in WT cells (or close to them on the genome) and their change in expression (and localization as suggested in point 1) in the various genetic and cellular conditions. Thirdly, how do the authors reconcile genes which are upregulated or downregulated in the QKO have elevated levels of H3K9me2 compared to those with no change. If H3K9me2 plays a repressive role, shouldn't there be a difference in H3K9me2 levels between up- and downregulated genes (Fig 3D)? Fourthly, to support the conclusion that H3K9me2 modification unable to support transcriptional repression, the authors should demonstrate that under the same H3K9me2-domain calling parameters the genes enriched for H3K9me2 have minimal or significantly lower transcriptional activity than H3K9me2-enriched genes in TKO, LBRKO, and QKO. Is the lack of H3K9me2-repressive function is observed in QKO only?

A PCA showing similarity of RNA-seq replicates across conditions should be provided.

4. The authors should complement the differentiation studies with additional experiments. They should include marker analysis to more definitively determine differentiation efficiency and the differentiation defect observed in the lamin + LBR KO. Relatedly, the authors argue that there is an expansion of H3K9me2 upon epiblast differentiation. They observe this in the lamin + LBR KO (Fig. 6F, S9J), but the vast majority of cells in these cultures are not epiblast cells based on the analysis provided?

Additional minor points:

1. It would be helpful if the authors annotated the figures a bit more.

2. In the first paragraph of the introduction, the authors state, "These observations have led the prevailing model that peripheral heterochromatin positioning promotes the establishment of cell fate by repressing alternative fate genes. However, this model has not yet been tested, and the mechanism by which the nuclear periphery confers repression on associated chromatin remains unknown." This statement should be edited. The role of peripheral chromatin positioning in mediating alternative fate has been tested to some degree - (PMID: 34737442, 33529599 and 26607792).

3. It would be helpful if the authors better defined the number of genes used in their analysis and consistent across the manuscript (protein coding? lncRNA?)

4. The data in S7J does not appear statistically significant - the text should be edited so this is more clear.

5. How do the authors reconcile the large effect in repeat elements expressed in QKO versus the WT given the relatively few number of repeat elements unregulated in the QKO versus TKO and TKO versus WT?

6. Line 189 - "These unregulated genes are significantly enriched for H3K9me2 compared to unaffected or down regulated genes (Fig. 3I)." This conclusion is not consistent with the data shown, as there does not appear to be difference between the unregulated and down regulated genes.

7. Lines 281-286. While the authors have made a good attempt to rule out additional functions of LBR and lamins to mediate the differentiation phenotype, it is still not definitive that "H3K9me2 positioning is required for EpiLC viability". This conclusion should be toned down.

8. It would be helpful for the readers if the authors be more specific describing "the nuclear periphery" throughout. For example, in the abstract lines 27-28 it says, "We conclude that the nuclear periphery controls the spatial position, dynamic remodeling, and repressive capacity of H3K9me2-marked heterochromatin to shape cell fate decisions." What exactly do the authors mean by the nuclear periphery? Nuclear periphery proteins? Localization at the nuclear periphery? A layer of the peripheral heterochromatin?

9. Line 164: "...Lamin TKO mESCs (where heterochromatin positioning is intact)". The data showed in Fig. 1M suggests otherwise. There is a significant difference between WT and TKO (as well as LBR KO). The authors should consider changing "intact" to "retain localization at the nuclear periphery".

10. The authors reported 59% (WT ESC), 65% (LBR KO), and 66% (WT EpiLC) genome coverage by the H3K9me2 histone modification. The transition from WT ESC to EpiLCs (59% to 66%) is described as expansion of H3K9me2 across the genome. At the same time LBR KO result is similar changes (59% to 65%). Why are these similar results interpreted differently?

Reviewer #2:

Remarks to the Author:

The MS by Marin et al describes an interesting phenomenon of nuclear architecture disruption by quadruple knockout (QKO) of all lamins and LBR in mESCs. Earlier, it has been shown that depletion of B-lamins does not affect chromatin distribution, whereas lamin A/C and LBR function as interchanging or synergetic tethers of peripheral heterochromatin. The authors demonstrate that QKO of all these proteins leads to dissociation of heterochromatin from the nuclear periphery, which ultimately causes dysregulation of large number of genes and de-repression of transposons. Furthermore, differentiation of QKO mESCs into epiblast-like cells (EpiLCs) is severely impaired. The QKO EpiLCs exhibit similar detachment of heterochromatin from the nuclear periphery, but nonetheless they silence naïve pluripotency genes and activate epiblast stage genes.

The manuscript is very clearly written and sufficiently illustrated (with some small exceptions we mentioned below). It is especially pleasing to see that authors supplemented RGB microscopy images with grey-scale images of single channels that greatly facilitate understanding of the phenomena. The methods used in the paper are diverse and appropriate, which allowed the authors to tackle their questions from different sides. We also have to praise the authors for the very clear Material and Methods section. All-in-all, We estimate this work as a potential great contribution to our understanding of genome organization and regulation. In particular, it marks the major step in distinguishing between the repressive effects of heterochromatin and its attachment to the periphery.

We have three major comments to the authors. The first is in regards to the readout of their experiments, i.e. the distribution of H3K9me2 PTM. It remains unclear why the authors consider it as is the major silencing mark of the peripheral heterochromatin. There are other not less important marks - such as H3K27me3, H3K9me3, H4K20me3, etc. For instance, the authors claim activation of TEs, which are known to be silenced by both H3K9me2 and H3K9me3. Looking at the references, we guess that the authors selected H3K9me2 mark based on highly debatable data of strictly peripheral distribution of this PTM, which originates from the same single lab [e.g., 10.7554/eLife.49278.001; 10.1126/sciadv.abj3035; etc]. The data from other groups, however, show more internal distribution of this mark [10.1242/dev.127308; 10.1104/pp.114.255737; 10.1158/1541-7786.MCR-14-0474; etc]. Moreover, the latter view is supported by living cell observations demonstrating that constitutive LADs, greatly enriched in H3K9me2, can be found not only at the nuclear but also at the nucleolar periphery, as well as in other internal loci [10.1016/j.cell.2013.02.028].

In this connection, we suggest the authors to present more balanced view on the distribution of H3K9me2 mark and tune down citation of the controversial results from that single group. In addition, to validate the choice of experimental readout and demonstrate, that the heterochromatin in general and not only H3K9me2-positive chromatin, is displaced or not displaced – we suggest to perform immunostaining of several other heterochromatin markers at the all three conditions - TKO, LBR-KO, QKO.

Our second point is that the authors show a clear release of H3K9me2 marked chromatin by microscopy. However, it is entirely untested which nuclear envelope-genome interactions have been altered or how widely this occurs through the genome. Are all nuclear envelope-heterochromatin broken? Conversely, does euchromatin replace heterochromatin at the periphery as in rod cells? If so, do euchromatic loci show repression when repositioned and does this explain why many loci are downregulated in the QKO mutant? Cut&Run or DamID from Lap2β in QKO cells should reveal this. This could also be complimented by staining for several euchromatin marks, like acetylated histones, H3K4me3 or H3K36me3. In short, these questions can be readily answered by the authors and are necessary to make the manuscript a true advance.

Finally, it is not clear how many clones are used as replicates in the analyses throughout the paper. Figure 5E states that at least two independent clones were tested by IF and the materials and methods indicate 10 homozygous KO clones were generated by CRISPR.

How many clones were tested as replicates for genomics assays, including RNA-seq and Cut&Run, in figures 2, 3, 4, 6 and 7?

Considering the significant impact of these findings, we feel it is essential that the trends are reproducible across at least 2 clones. This is because the selection of specific ESC sub-clones can itself introduce significant differences in chromatin and gene expression.

Detailed comments:

Lines 46-47, 50-51: heterochromatin is positioned also at the nucleoli periphery and serves as a silencing compartment similarly to the nuclear periphery; in mouse cells, it also surrounds chromocenters built by major satellite repeat.

Line 56: it should be mentioned, that the paper Smith et al 2021, postulates a controversial view on distribution of histone modifications in retinal cells – compare with other publications [10.1007/s10577-013-9375-7 or 10.1016/j.neuron.2019.08.002].

Line 86: the reference 36 is not correctly cited - Zheng et al showed that upon depletion of all lamins LADs decondense or detach from the nuclear periphery.

Line 100: in Fig.1, all panels with images have to be supplemented by DNA staining of the corresponding cells – this will allow to appreciate distribution of heterochromatin, including chromocenters, and other nuclear features. This comment is valid for all other figures in the MS showing nuclear images - Fig.5, S3, S7.

Could the authors comment on the increased internal H3K9me2 staining in LBR-KO visible in the 1C?

Line 101: in Fig.S2B and Fig.S2G, increase of heterochromatin lamps inside the LBR-KO nuclei is obvious. E.g., we do not see a difference in DNA distribution between Fig.S2B and Fig.S2D.

We are aware that the authors undertook quantification of the DNA signal (Fig.S2E), however, in our view the correct analysis based on a single optical section is difficult for mESCs because of the AT-rich chromocenters and large nucleolus. Therefore, we suggest the authors to prepare a Supplementary Information file showing mid-confocal sections of DNA staining for all collected nuclei and for all conditions.

Line 118: Fig.S3F: in this figure, as examples for TKO+LBR miR-E the pairs of nuclei are shown (two panels in the right low corner). These pairs are apparently sister cells in early G1, the stage when chromatin is partially condensed and nuclear architecture is not yet established. They have to be replaced by examples of later cell cycle stages.

Lines 121-123: Could the authors comment whether the movement of H3K9me2-positive heterochromatin inside and outside occurs within one cell cycle or after cell division? In other words, do cells have to go through mitosis with following nuclear formation in order to rearrange heterochromatin?

Line 140-141: The distribution of H3K9me2 appears similar between all samples. However, absolute levels of signal cannot be compared in the absence of a spike-in control. The materials and methods states this is the case with all Cut&Run experiments, but the text implies Spike-in was only performed in Figure 7.

Can the authors demonstrate that there is no difference in overall H3K9me2 signal in mutant vs wildtype ESCs via Spike-in normalised Cut&Run? We feel this is essential to be able to make this claim.

Line 183-192: There is a strong discordance between number of mis-expressed genes in QKO knockout cells versus those with the inducible LBR knockdown. Naturally this could just be the result of incomplete loss of LBR or a time delay before H3K9me2-repressive fails. Could the authors confirm that the genes that are mis-expressed in knockdown cells are also mis-expressed in knockout cells (and in the same direction)?

Related to this, what is the overlap between gene mis-regulation in QKO cells and those where G9a/GLP is inactivated? A difference in the latter groups of genes may indicate that some H3K9me2-marked loci are more sensitive to LAD loss than others. Alternatively, if no overlap is observed, this would indicate that the connection between lost lamina-association and gene expression is not intrinsically linked to H3K9me2.

Line 284: Fig.S7J: are the values of EM density in the graph significant? In our view, such evaluation of chromatin density after TEM makes no sense because areas in vicinity to nuclear envelope and in nucleoplasm can unavoidably be selected arbitrarily.

There is no reference to the Fig.S8 through the entire MS.

Line 301: Plot 6C indicates that the vast majority of the genome (18729 genes) become incorporated into H3K9me2 regions. This far exceeds previous estimates, including cell-types that arise after EpiLCs. Do the authors have an explanation for this?

We feel it is necessary to rule out technical artefacts to make this claim. Could the authors also show the IgG control from both cell-types and confirm there is also not the same increase in non-specific signal in EpiLCs? For the reviewers, it would be key to see the IgG control alone, as well as a log2 ratio.

How are other chromatin marks affected, such as H3K9me3 and H3K27me3?

Line 425-428: these conclusions by Smith et al are highly debatable and in conflict with other existing literature about H3K9me2 as a repressive mark of heterochromatin positioned also inside the nucleus, e.g. around nucleoli. As for mouse rod nuclei, to which the authors refer themselves, other researchers have shown absence of H3K9me2 at the nuclear periphery and its presence in the central heterochromatin blob [10.1007/s10577-013-9375-7]. We believe, the authors have to refer to and discuss literature in a more balanced manner.

Reviewer #3:

Remarks to the Author:

In this manuscript, Marin et al., describe the effect of quadruple knockout of Lamin B1 / B2, Lamin A/C, and LBR on spatial DNA positioning, heterochromatic organization and gene expression in mouse embryonic stem cells. As a result of QKO, the authors find detachment of lamina-bound genomic regions. Intriguingly, the characteristic H3K9me2/3 histone modifications on these regions remain intact, yet become permissive to gene activation and TE expression. Furthermore, mESCs are unable to efficiently commit to the epiblast lineage during in-vitro differentiation towards epiblast-like cells.

This work is of high quality, and the observations will be of great interest to the broader (epi)-genomics field. The authors show that proximity to the nuclear periphery provides essential context for heterochromatin-mediated silencing of genomic regions. Therefore, we see a lot of value in this work being published in Nature cell Biology.

Nonetheless, we have comments that we would like the authors to address and that we think would improve the manuscript further.

Major points:

- The light- and EM-microscopy data show a striking repositioning of heterochromatin away from the nuclear lamina. This observation would benefit from complementary analysis through genomic means. For instance, through DamID-seq, pA-DamID or CUT&RUN against NL-components such as Emerin or Lap2B.
- The authors describe using a 4-state Hidden Markov Model. However, a justification for choosing this number of states is lacking. This is significant because the states are the basis for subsequent analyses on K9 decoration and gene expression. The authors should 1) explain their choice and methods in more detail, 2) show the other two states in the figure panels as well, and 3) compare the overlap between the states and blacklisted regions rigorously.
- Fig 6 is dependent on the 4-state HMM, which the authors rightfully mention that the HMM is found to be ineffective in describing the K9me2/3 domains in EpiLCs. This should be investigated in greater detail in light of our previous comments.
- The analysis of genes upregulated upon QKO is cursory and lacks depth. It would be interesting if the authors could look more into gene regulatory networks, regulation of major transcription factors and classes of genes that are affected upon manipulation in the context of genomic location. Now, too often, the narrative is dictated by cherry-picked examples and GO-term analyses are lacklustre.
- It would be interesting to cross reference the differentially expressed genes upon the knockout and the acute RNAi-mediated knockout condition.
- It has already been shown that LBR and Lamin A/C depletion alone can cause massive chromatin repositioning. It is unclear if the QKO-results are thus driven by absence of LBR and Lamin A/C or that there is a true (novel) effect of the quadruple depletion. It would be very interesting to see what the contribution of the B-type lamins is to the phenotype, or whether the results are caused primarily or exclusively by the absence of LBR and Lamin A
- Looking at the microscopy, detachment is still preferentially located to the NL. An interesting alternative hypothesis is that there is no real detachment but rather decompaction of heterochromatin. The authors should investigate the distinction between detachment and decompaction.

Minor points:

- The Venn-diagrams in the supplemental show that this manuscript's domains cover much more genomic loci than the data they are comparing to. The authors should show that their calls are not inflated by false positives (i.e., background-signal).
- The mechanism of LBR binding heterochromatin through HP1 is not mentioned until the discussion. This context would be valuable to mention in the introduction.
- Throughout the manuscript, the authors don't always mention the sample size for presented analyses. For example, Fig. S2E, 4A/B, 7A

and more.

-The authors find moderate effects on LBR KO, and the TKO conditions but don't discuss them much. Some reflections on these observations could be helpful to give context to their findings.

-Figure 3A is counterintuitive: why are the numbers shown for QKO-vs-TKO not the difference between QKO-vs-WT and TKO-vs-WT?

-Figures 3D and 4I mask gene-specific effects and clusters. It would be better to visualise this with enrichment-heatmaps (made with tools such as DeepTools2).

-Some figures could be merged (fig 5 & 6) and some panels are not interesting enough for main text figures. E.g. 2E/F, 3C, 3G (generally the GO terms are uninteresting filler). Fig 6 B/C/D are largely uninteresting.

-Line 166 onwards, the authors mention that upregulation is the main effect of QKO. Although we agree, there's a considerable fraction of genes downregulated as well. It would be interesting to give these genes more context.

-The authors decide to focus on the L1MdA_I LINE1 family. No justification is given why this family is particularly interesting.

-Is the morphology of 2i/LIF mESCs normal in QKO conditions? In S1G, they look less dome-shaped?

-Fig 5B is missing a scale bar.

-Fig 6G, H, the authors conclude that K9 is differentially enriched on domains. But it seems to me the enrichment is the same, rather the baseline is just higher?

-Fig 7H colours are super confusing and it is unclear if these are all the genes of interest or that these are cherry-picked.

-The authors hypothesize that division is preventing nuclear inversion. Could they test this?

Methods should be written concisely, but should contain all elements necessary to allow interpretation and replication of the results. As a

guideline, Methods sections typically do not exceed 3,000 words. The Methods should be divided into subsections listing reagents and techniques. When citing previous methods, accurate references should be provided and any alterations should be noted. Information must be provided about: antibody dilutions, company names, catalogue numbers and clone numbers for monoclonal antibodies; sequences of RNAi and cDNA probes/primers or company names and catalogue numbers if reagents are commercial; cell line names, sources and information on cell line identity and authentication. Animal studies and experiments involving human subjects must be reported in detail, identifying the committees approving the protocols. For studies involving human subjects/samples, a statement must be included confirming that informed consent was obtained. Statistical analyses and information on the reproducibility of experimental results should be provided in a section titled "Statistics and Reproducibility".

All Nature Cell Biology manuscripts submitted on or after March 21 2016 must include a Data availability statement at the end of the Methods section. For Springer Nature policies on data availability see <http://www.nature.com/authors/policies/availability.html>; for more information on this particular policy see <http://www.nature.com/authors/policies/data/data-availability-statements-data-citations.pdf>. The Data availability statement should include:

- Accession codes for primary datasets (generated during the study under consideration and designated as "primary accessions") and secondary datasets (published datasets reanalysed during the study under consideration, designated as "referenced accessions"). For primary accessions data should be made public to coincide with publication of the manuscript. A list of data types for which submission to community-endorsed public repositories is mandated (including sequence, structure, microarray, deep sequencing data) can be found here <http://www.nature.com/authors/policies/availability.html#data>.
- Unique identifiers (accession codes, DOIs or other unique persistent identifier) and hyperlinks for datasets deposited in an approved repository, but for which data deposition is not mandated (see here for details <http://www.nature.com/sdata/data-policies/repositories>).
- At a minimum, please include a statement confirming that all relevant data are available from the authors, and/or are included with the manuscript (e.g. as source data or supplementary information), listing which data are included (e.g. by figure panels and data types) and mentioning any restrictions on availability.
- If a dataset has a Digital Object Identifier (DOI) as its unique identifier, we strongly encourage including this in the Reference list and citing the dataset in the Methods.

We recommend that you upload the step-by-step protocols used in this manuscript to [protocols.io](https://www.protocols.io). More details can be found at <https://www.protocols.io/help/publish-articles>.

All imaging data should be accompanied by scale bars, which should be defined in the legend. Cropped images of gels/blots are acceptable, but need to be accompanied by size markers, and to retain visible background signal within the linear range (i.e. should not be saturated). The boundaries of panels with low background have to be demarked with black lines. Splicing of panels should only be considered if unavoidable, and must be clearly marked on the figure, and noted in the legend with a statement on whether the samples were obtained and processed simultaneously. Quantitative comparisons between samples on different gels/blots are discouraged; if this is unavoidable, it should only be performed for samples derived from the same experiment with gels/blots were processed in parallel, which needs to be stated in the legend.

Regardless of format, all figures must be vector graphic compatible files, not supplied in a flattened raster/bitmap graphics format, but

should be fully editable, allowing us to highlight/copy/paste all text and move individual parts of the figures (i.e. arrows, lines, x and y axes, graphs, tick marks, scale bars etc.). The only parts of the figure that should be in pixel raster/bitmap format are photographic images or 3D rendered graphics/complex technical illustrations.

The total number of Supplementary Figures (not including the “unprocessed scans” Supplementary Figure) should not exceed the number of main display items (figures and/or tables (see our Guide to Authors and March 2012 editorial <http://www.nature.com/ncb/authors/submit/index.html#suppinfo>; <http://www.nature.com/ncb/journal/v14/n3/index.html#ed>). No restrictions apply to Supplementary Tables or Videos, but we advise authors to be selective in including supplemental data.

GUIDELINES FOR EXPERIMENTAL AND STATISTICAL REPORTING

REPORTING REQUIREMENTS – To improve the quality of methods and statistics reporting in our papers we have recently revised the reporting checklist we introduced in 2013. We are now asking all life sciences authors to complete two items: an Editorial Policy Checklist (found here <https://www.nature.com/authors/policies/Policy.pdf>) that verifies compliance with all required editorial policies and a reporting summary (found here <https://www.nature.com/authors/policies/ReportingSummary.pdf>) that collects information on experimental design and reagents. These documents are available to referees to aid the evaluation of the manuscript. Please note that these forms are dynamic ‘smart pdfs’ and must therefore be downloaded and completed in Adobe Reader. We will then flatten them for ease of use by the reviewers. If you would like to reference the guidance text as you complete the template, please access these flattened versions at <http://www.nature.com/authors/policies/availability.html>.

STATISTICS – Wherever statistics have been derived the legend needs to provide the n number (i.e. the sample size used to derive statistics) as a precise value (not a range), and define what this value represents. Error bars need to be defined in the legends (e.g. SD, SEM) together with a measure of centre (e.g. mean, median). Box plots need to be defined in terms of minima, maxima, centre, and percentiles. Ranges are more appropriate than standard errors for small data sets. Wherever statistical significance has been derived, precise p values need to be provided and the statistical test used needs to be stated in the legend. Statistics such as error bars must not be derived from n<3. For sample sizes of n<5 please plot the individual data points rather than providing bar graphs. Deriving statistics from technical replicate samples, rather than biological replicates is strongly discouraged. Wherever statistical significance has been derived, precise p values need to be provided and the statistical test stated in the legend.

We strongly recommend the presentation of source data for graphical and statistical analyses as a separate Supplementary Table, and request that source data for all independent repeats are provided when representative experiments of multiple independent repeats, or averages of two independent experiments are presented. This supplementary table should be in Excel format, with data for different figures provided as different sheets within a single Excel file. It should be labelled and numbered as one of the supplementary tables,

titled "Statistics Source Data", and mentioned in all relevant figure legends.

Version 1:

Decision Letter:

Our ref: NCB-A54557A

18th March 2025

Dear Dr. Buchwalter,

Thank you for submitting your revised manuscript "The nuclear periphery confers repression on H3K9me2-marked genes and transposons to shape cell fate" (NCB-A54557A) and for your patience with the review process. It has now been seen by the original referees and their comments are below. The reviewers find that the paper has improved in revision, and therefore we'll be happy in principle to publish it in Nature Cell Biology, pending minor revisions to satisfy the referees' final requests and to comply with our editorial and formatting guidelines.

Please note that our articles must have 6 to 8 main figures and they can have up to 10 ED figures. If there are more figures than that, they become supplementary figures. Also supp figures, and supp notes and other supplementary materials, are included in the supplementary information PDF on the website. These are less accessed than our main and ED figures so we try to limit the use of supplementary figures as much as we can. Please ensure that all figures fit into a single page (not multiple pages) and adhere to a maximum page size of roughly 180mm wide x 200mm high and use a font size of no smaller than 6pt throughout the figures, to ensure legibility of the figures once resized for publication.

We are now performing detailed checks on your paper and will send you a checklist detailing our editorial and formatting requirements in about 10 days. **Please do not upload the final materials and make any revisions until you receive this additional information from us.

Thank you again for your interest in Nature Cell Biology Please do not hesitate to contact me if you have any questions.

Sincerely,

Sabrya Carim, PhD
(she/her/hers)
Senior Editor, Nature Cell Biology
Nature Portfolio

Springer Nature
The Campus, 4 Crinan Street, London N1 9XW, UK
sabrya.carim@springernature.com
<https://orcid.org/0000-0001-9485-1938>

Reviewer #1 (Remarks to the Author):

I am satisfied with the authors response and feel the manuscript should be accepted. Congratulations to the authors on a fantastic study!

Reviewer #2 (Remarks to the Author):

The authors have made significant effects to address our concerns and we believe, as we already stated in our first review, that this is a highly timely and exciting paper for the field. In particular, the authors have (a) edited the text to clarify their focus on H3K9me2 and controversies regarding its peripheral localization, (b) profiled breaking genome-nuclear envelope interactions via Lap2 β Cut&Run, (c) re-focused their gene expression analysis.

We have only two remaining concerns that should be addressed before publication.

Lines 102-106: The authors clarify that all QKO data derives from a single clone because others have not survived after they were thawed for expansion. This is naturally a concern, as the single successfully isolated clone may be an aberration. The authors make significant effort to address this by demonstrating similar effects through other strategies (e.g., inducible KOs and rescue experiments). As such, we feel their conclusions are valid. Nevertheless, we feel they should make the fact that only a single clone could be analyzed clearly articulated in the main text, as it is an important caveat that should not be overlooked.

Fig. 2: The authors did excellent and careful work validating Lap2 β as a target for peripheral DNA mapping by Cut&Run. Nevertheless, we see clear indications of a very large scale increasing in non-LAD contacts in the KO cells. Specifically, LBR-KO cells show a strong redistribution of signal into previously non-LAD regions (left hand side of Fig.2F). This indicates that ectopic signal is diminished and more localized to specific loci in the QKO cells. To us, this strongly suggests that there is at least some form of chromatin inversion and so is an important point to clarify before publication.

Specifically, how extensive is this signal redistribution in each successive mutant? Do the gained regions correspond with a specific chromatin signature (e.g. H3K27ac, H3K27me3, H3K9me3)? Quantifying this globally as well as showing extended figure views of broader chromosome scale plots would be important.

This going along with our previous minor point 9. We do appreciate all the efforts the authors made for image quantification. However, following saying "seeing is believing", we still think it would be useful for readers and followers of this exciting study to see a broad collection of mid-sections for WT, TKO, LBR-KO and QKO. Relatively low number of analyzed nuclei – WT (n = 17), TKO (n = 25), LBR KO (n = 26), and QKO (n = 18) – allows to show them all in a Supplementary Figure without new experiments.

Additional Comments in response to Reviewer #3's remaining points:

1. The distinction between chromatin decompaction and NL detachment is underemphasized. The newly added Lap2β data suggest substantial detachment, yet Ext. Data Fig. 1D/H indicates that H3K9me2-decorated chromatin remains near the NL: while the rim-staining is lost, the chromatin still appears in close proximity to the NL. This distinction is critical, as it affects the interpretation of gene expression changes, H3K9me2 alterations, and other downstream effects.

- In my view, the comment is not valid, because the reviewer refers to H3K9me2 staining, which might be not always successful, however, accompanying DAPI images clearly show that chromatin does dissociate from the lamina in QKO.

2. Due to technical limitations, Lap2β CUT&RUN data in EpiLCs is missing. As a result, the authors rely on H3K9me2 as a proxy for DNA-NL tethering, which muddles the interpretation of the data and makes it challenging to infer an underlying mechanism.

- It would indeed be more ideal if the authors could successfully show Lap2β CUT&RUN data in KO EpiLCs. However, there are 3 points that support the author's conclusions. (1) the authors have already profiled LADs in WT EpiLCs, and so have determined what loci are normally at the nuclear periphery. (2) They also performed IF in EpiLCs, confirming the H3K9me2 is significantly released from the periphery in the QKO condition. (3) They are careful in their wording when describing the CUT&RUN QKO data to avoid describing.

E.g., "We show that H3K9me2 domains are extensively remodeled during the transition from naïve to primed pluripotency (Fig. 6). While LADs are overwhelmingly modified by H3K9me2 in the naïve state (Fig. 2B), H3K9me2 is removed from many LADs in the primed state (Fig. 6A-C). Interestingly, LADs remain strongly repressive despite this remodeling (Fig. 6D), suggesting that other chromatin modifications and/or cofactors maintain repression of primed LADs."

Thus, though a limitation, it does not alter the paper's findings.

3. The manuscript lacks mechanistic insight. The analyses remain relatively superficial, missing an opportunity to explore the biological implications of the interesting QKO phenotype in greater depth.

- We disagree with this comment. If the mechanistic insight would be possible, the paper will be published in Nature. This is an important advance in describing the factors that mediate lamina-attachment and so should be made available to the community immediately.

Specific remaining points:

1. The IGV figures display only a few megabases of the genome. Since LADs are large, multi-megabase structures, it would be beneficial to present data across an entire chromosome to better assess the effects of QKO and the overall data quality.

- I agree that showing multiple scales would provide important context as to the local and large-scale effects on LADs.

2. The CUT&RUN data for H3K9me3 shows unexpectedly poor overlap with H3K9me2 and LADs. This apparent anti-correlation is puzzling to us, as previous studies have reported significant overlap between H3K9me3 and LADs. The authors need to further clarify and/or discuss this discrepancy.

- H3K9me3 has indeed been reported to correlate with LADs. However, our unpublished data shows that the extent of this correlation strongly varies with cell-type. For example, ESCs show limited "peaky" H3K9me3 while multiple cancer cell lines and differentiated tissues possess broader H3K9me3 that often, but not always, overlaps with LADs. This can be seen in available encode H3K9me3 ChIP-seq data.

However, we agree that this point is not obvious in the field and so the authors should discuss it to prevent confusion in the field.

3. In Fig. 2F, some regions gain contacts with the NL upon QKO. These regions could provide insight into the underlying mechanism— are they actively transcribed or repressed? This is an example of a missed opportunity to really dive into the mechanistic underpinnings of their observation.

- We do agree, with this point and highlighted in our own review. The authors stated that these gained peaks were minimal and not significant. However, they appear highly significant to us and so we believe the authors should perform a list a general analysis of their distribution in the genome and the properties of the genes/loci they contain.

4. The role of LmnA and LBR in driving the phenotype remains unclear. The authors attempted to re-express LmnA in the TKO cell line, but this experiment was unsuccessful. We would appreciate seeing these data and understanding why this approach was chosen instead of knocking out LmnA in the existing LBR KO cell line, which would more directly address this question.

- We have not found experiments, successful or unsuccessful, with re-expression of LmnA in TKO in the paper. However, ED.Fig.4F shows depletion of LBR in TKO, and results indicate the major role of LBR in chromatin tethering. Hence, generation of QKO. The exact mechanisms of chromatin tethering by LMNA and LBR, indeed, remain to be elucidated, but the phenomenon is clearly described.

5. The hypothesis that cell cycle dynamics prevent complete chromatin inversion in QKO is intriguing, but no supporting data are provided. If this hypothesis is retained, additional supporting evidence would be valuable; otherwise, it may be best to remove it.

- The hypothesis was proposed not by these authors but was first discussed in Solovei et al (2013) and Falk et al (2019), where authors claimed that in difference to postmitotic cells, quickly cycling cells cannot complete nuclear inversion.

6. Our previous concern regarding the underexploration of gene expression changes in TKO and QKO remains (see Major Point 3).

- In our opinion, the transcriptome analysis of mutants is sufficient.

7. Our previous comment on boxplots also still holds: it is difficult to determine whether effects are normally distributed or exhibit a different pattern (e.g., bimodal). A heatmap would be ideal, but a violin plot would also improve clarity.

- We cannot comment on this.

8. Similarly, the lack of analysis into upregulated genes remains an issue (see Major Point 3).

- As mentioned above, we do not think more deep analysis is needed.

9. Previous minor point 19 still holds true in our opinion: the kernel plot seems to support our comment, showing higher counts of K9me2 across the genome. This would point to a higher background. We'd like the authors to respond to this.

- If the reviewer means Fig.5J, we do not see controversy here: the plot shows shift of H3K9me2 chromatin from the periphery in the interior although not all heterochromatin shifted.

10. A recent bioRxiv preprint from the Kuntay lab (December 2024) reports a similar phenotype in mESCs but highlights the essential role of Lap2 β and LBR in DNA-NL tethering. It would be valuable for the authors to discuss these findings and address any discrepancies between the two studies.

- This is a very good suggestion: the authors should cite Levis et al manuscript published recently on BioRxiv

Reviewer #3 (Remarks to the Author):

We have reviewed the revised manuscript by Marin et al., which investigates the effects of quadruple knockout (QKO) of Lamin B1, Lamin B2, LmnA/C, and LBR in mouse embryonic stem cells. The authors report that QKO influences DNA-NL interactions and further examine its impact on H3K9me2 distribution, gene expression, and differentiation potential. We appreciate the addition of Lap2 β CUT&RUN data and the refined implementation of Hidden Markov Modeling in this revision.

While we find the core observation of the study intriguing, we believe that several aspects of the analysis remain unclear and lack mechanistic insights, particularly given the standards of Nature Cell Biology. Below, we outline our key concerns:

Major general comments:

1. The distinction between chromatin decompaction and NL detachment is underemphasized. The newly added Lap2 β data suggest substantial detachment, yet Ext. Data Fig. 1D/H indicates that H3K9me2-decorated chromatin remains near the NL: while the rim-staining is lost, the chromatin still appears in close proximity to the NL. This distinction is critical, as it affects the interpretation of gene expression changes, H3K9me2 alterations, and other downstream effects.
2. Due to technical limitations, Lap2 β CUT&RUN data in EpiLCs is missing. As a result, the authors rely on H3K9me2 as a proxy for DNA-NL tethering, which muddles the interpretation of the data and makes it challenging to infer an underlying mechanism.
3. The manuscript lacks mechanistic insight. The analyses remain relatively superficial, missing an opportunity to explore the biological implications of the interesting QKO phenotype in greater depth.

Specific remaining points:

1. The IGV figures display only a few megabases of the genome. Since LADs are large, multi-megabase structures, it would be beneficial to present data across an entire chromosome to better assess the effects of QKO and the overall data quality.
2. The CUT&RUN data for H3K9me3 shows unexpectedly poor overlap with H3K9me2 and LADs. This apparent anti-correlation is puzzling to us, as previous studies have reported significant overlap between H3K9me3 and LADs. The authors need to further clarify and/or discuss this discrepancy.
3. In Fig. 2F, some regions gain contacts with the NL upon QKO. These regions could provide insight into the underlying mechanism— are they actively transcribed or repressed? This is an example of a missed opportunity to really dive into the mechanistic underpinnings of their observation.
4. The role of LmnA and LBR in driving the phenotype remains unclear. The authors attempted to re-express LmnA in the TKO cell line, but this experiment was unsuccessful. We would appreciate seeing these data and understanding why this approach was chosen instead of knocking out LmnA in the existing LBR KO cell line, which would more directly address this question.
5. The hypothesis that cell cycle dynamics prevent complete chromatin inversion in QKO is intriguing, but no supporting data are provided. If this hypothesis is retained, additional supporting evidence would be valuable; otherwise, it may be best to remove it.
6. Our previous concern regarding the underexploration of gene expression changes in TKO and QKO remains (see Major Point 3).
7. Our previous comment on boxplots also still holds: it is difficult to determine whether effects are normally distributed or exhibit a different pattern (e.g., bimodal). A heatmap would be ideal, but a violin plot would also improve clarity.
8. Similarly, the lack of analysis into upregulated genes remains an issue (see Major Point 3).
9. Previous minor point 19 still holds true in our opinion: the kernel plot seems to support our comment, showing higher counts of K9me2 across the genome. This would point to a higher background. We'd like the authors to respond to this.
10. A recent bioRxiv preprint from the Kuntay lab (December 2024) reports a similar phenotype in mESCs but highlights the essential role

of Lap2 β and LBR in DNA-NL tethering. It would be valuable for the authors to discuss these findings and address any discrepancies between the two studies.

We appreciate the efforts made in this revision and hope these comments will help strengthen the manuscript.

Version 2:

Decision Letter:

Dear Dr Buchwalter,

I am pleased to inform you that your manuscript, "The nuclear periphery confers repression on H3K9me2-marked genes and transposons to shape cell fate", has now been accepted for publication in Nature Cell Biology. Congratulations!

Please note that *Nature Cell Biology* is a Transformative Journal (TJ). Authors may publish their research with us through the traditional subscription access route or make their paper immediately open access through payment of an article-processing charge (APC). Authors will not be required to make a final decision about access to their article until it has been accepted. [Find out more about Transformative Journals](https://www.springernature.com/gp/open-research/transformative-journals)

If you have not already done so, we strongly recommend that you upload the step-by-step protocols used in this manuscript to protocols.io (<https://protocols.io>), an open online resource that allows researchers to share their detailed experimental know-how. All uploaded protocols are made freely available and are assigned DOIs for ease of citation. Protocols and Nature Portfolio journal papers in which they are used can be linked to one another, and this link is clearly and prominently visible in the online versions of both. Authors who performed the specific experiments can act as primary authors for the Protocol as they will be best placed to share the methodology details, but the Corresponding Author of the present research paper should be included as one of the authors. By uploading your Protocols onto protocols.io, you are enabling researchers to more readily reproduce or adapt the methodology you use, as well as increasing the visibility of your protocols and papers. You can also establish a dedicated workspace to collect your lab Protocols. Further information can be found at <https://www.protocols.io/help/publish-articles>.

Nature Cell Biology encourages authors presenting evidence for cell, biological, molecular, and genetic interactions to consider communicating these findings using Biofactoid (<https://biofactoid.org/>). This tool helps users share a searchable representation of interactions (e.g. binding, gene expression, post-translational modification) between genes, gene products, or chemicals. Information added to Biofactoid, with author attribution, is shared on social media and public databases, such as Pathway Commons, where it can be discovered and analyzed in the context of a large and growing corpus of knowledge.

With kind regards,

Sabrya.

Sabrya Carim, PhD
(she/her/hers)
Senior Editor, Nature Cell Biology
Nature Portfolio

Springer Nature
The Campus, 4 Crinan Street, London N1 9XW, UK
sabrya.carim@springernature.com
<https://orcid.org/0000-0001-9485-1938>

** Visit the Springer Nature Editorial and Publishing website at http://editorial-jobs.springernature.com?utm_source=ejp_NCB_email&utm_medium=ejp_NCB_email&utm_campaign=ejp_NCB for more information about our career opportunities. If you have any questions please click [here](mailto:editorial.publishing.jobs@springernature.com).

Response to Editor

We sincerely thank the editor and the reviewers for their thoughtful critiques and experimental ideas in their initial evaluation of our manuscript. In the initial assessment of our manuscript in July 2024, there were four main concerns (**A-D**, concerns and updates summarized below) that required new experiments and analyses before our manuscript could be accepted for publication. We have responded to these four areas of concern in our revised manuscript and provide a detailed point-by-point response to all reviewer critiques below. Reviewers' original comments appear numbered in black text, and our responses appear in blue. We have made every effort to provide additional data, controls, methodological details, and improvements to the text where requested. We feel that the reviewers' perceptive critiques and our labors to address them have substantially improved our revised manuscript.

A- Perform further experiments to distinguish between two possible models of chromatin repositioning at the nuclear periphery (Reviewer#1 pt 1, Reviewer#3 major pt 7)

While Reviewer 1 proposed DNA-FISH of candidate loci to address this question, we reasoned that we could address this question genome-wide by mapping contacts between the inner nuclear membrane (INM) and the genome in normal and mutant cells. We validated a new INM target for this genome-binding analysis (LAP2 β). By performing LAP2 β CUT & RUN, we found that deletion of tethering proteins causes nearly global detachment of heterochromatin from the nuclear envelope. These data directly demonstrate that displacement, rather than local decompaction, is a major consequence of the loss of heterochromatin tethers.

B- Clarify the rationale behind using the distribution of H3K9me2 as the major readout for the experiments and validate the choice of experimental readout (Reviewer#2 pt 1), as well as the choice of the 4-state HMM model (Reviewer#3 major pts 1,2)

We include new microscopy of the spatial enrichment of other heterochromatin marks and also map the H3K9me3 modification across the genome by CUT & RUN. These data demonstrate that H3K9me2, more so than H3K9me3, is enriched on lamina-associated heterochromatin in naïve stem cells. We improved our HMM analysis method and provide additional validation data.

C- Further experimentally investigate the release of H3K9me2 from the nuclear envelope (Reviewer#2 pt 2, reviewer#3 pt 1)

As stated in A above, we now map LAP2 β :genome contacts in normal and mutant stem cells and demonstrate the major displacement of H3K9me2-marked chromatin from the nuclear periphery in mutant cells.

D- Perform further analysis of the gene expression analysis as per the recommendations (Reviewer#1 pt 3, Reviewer#3 major pt4)

We have overhauled our gene expression analyses and now focus more coherently on gene expression changes specifically within spatially displaced chromatin domains in mutant cells.

Response to Reviewer 1

Major Points:

1a. The authors present a model where LBR and lamin proteins regulate positioning of chromatin at the nuclear periphery. A fraction of genes retain H3K9me2 and their expression is dysregulated. However, there still appears to be H3K9me2 marked chromatin at the nuclear periphery in the lamin + LBR KOs according to the imaging (Fig 1D, H). This raises the question whether H3K9me2-marked chromatin is repositioned from the periphery (and if so, to what extent). This can be addressed by measuring the distance of candidate loci (nominated from their datasets) from the nuclear envelope using DNA FISH. The authors would also be able to determine whether individual DNA FISH probes gain volume in relation to their distance to the periphery across conditions using this approach - allowing them to determine whether lamin and LBR mediate positioning and/or compaction status.

We thank the reviewer for raising this important point. Rather than following the localization of select candidate loci by DNA-FISH, we took a genomic approach to map nuclear envelope-proximal chromatin in each genotype. To do this, we validated the use of the inner nuclear membrane protein LAP2 β as a fiducial marker of LADs (see Extended Data Fig. 5 and 7) and performed spike-in-controlled CUT & RUN. In WT mESCs, LB1 domains, LAP2 β domains, and H3K9me2 domains are highly correlated (Figure 2B). In mutant mESCs, LAP2 β :genome association progressively and globally decreases as indicated by spike-in-normalized CUT & RUN DNA yield (Figure 2E) and within WT LADs as indicated by spike-in-normalized profile plots (Figure 2G). Meanwhile, H3K9me2 remains present on the same loci and at similar levels in each genotype (Figure 2H-K). Therefore, we conclude that deletion of the lamins and LBR causes nearly global spatial displacement of H3K9me2-modified chromatin from the nuclear periphery. We cannot rule out the possibility that deletion of the lamins and LBR influences chromatin compaction.

1b. Relatedly, the authors should also determine measure the distance of bursting candidate loci (using intronic RNA FISH) to the nuclear envelope. While the authors suggest that movement away from the periphery precedes transcriptional activity, it is possible that lamin and LBR induce transcriptional silencing of loci at the periphery. If that is the case, KO of lamins and LBR will abrogate the silencing and the subsequent transcriptional activity may facilitate locus movement (and that of nearby regions/H3K9me2 domains) to the nuclear interior (consistent with PMID 25477464). Distinguishing between these two models would be very impactful and helpful for the field.

We agree with the reviewer that this is an intriguing model: that the lamins and LBR direct silencing at the nuclear periphery, and that locus displacement occurs as a consequence of transcriptional activation when the lamins and LBR are absent. If the loss of lamins and LBR relieved transcriptional silencing, we would expect to observe pervasive transcription within displaced loci in QKO mESCs. Our new LAP2 β Cut & Run data indicates essentially global displacement of WT LAD loci into the nuclear interior (Fig. 2E-G; Fig. 3D; Supplemental Figure 10A). Nearly twice as many WT LAD genes are upregulated as downregulated (Fig. 3D), indicating that repression of these domains is weakened. However, the majority of WT LAD genes (more than 90%) are unchanged in their expression in spite of their spatial displacement. Similarly, L1 LINE elements are strongly associated with the lamina in WT cells (Fig. 4A), but only a small proportion of the vast number of LAD-resident L1 elements become de-repressed in QKO mESCs, even though a large proportion of them lose contact with the nuclear periphery. It seems very unlikely that transcription of a relatively small number of genes and TEs could induce the massive genome reorganization that we observe. Instead, we favor the interpretation that spatial displacement into the nucleoplasm makes loci more permissive to transcription. This model is also supported by recent work that induced transcriptional activation within LAD loci followed by

genomic mapping of nearby lamina contacts; transcriptional activation induced only localized detachment of ~50-100 kb of a LAD from the lamina as a consequence (PMID 32080885).

2. Additional controls are required to properly interpret the changes in H3K9me2 and related CUT and RUN data. Firstly, the conclusions would be more convincing if the authors showed how the differential areas behave across replicates to ensure the replicates are closely aligned and the domain caller is accurate. For example, while the majority of genes identified in H3K9me2 domains in PMID 2903319 are also identified as in a H3K9me2 domain by the authors (Fig S4F), 23,114 other genes are not identified in a H3K9me2 domain by the other publication (PMID 2903319).

We thank the reviewer for raising this point. We now include CUT & RUN tracks and HMM domain call outcomes from individual replicates for CUT & RUN experiments in Extended Data Figures 6 (Lamin B1), 7 (LAP2 β), and 8 (H3K9me2); these show that CUT & RUN signal and HMM domain calling are highly reproducible across replicates.

We apologize for the lack of clarity in our comparisons to previously published data (PMID 2903319). When we initially compared to this previously published dataset, we used published supplemental data that reported only protein-coding genes, while our dataset includes both protein-coding and non-coding loci. To make a more appropriate comparison between these datasets, we now report the proportion of protein-coding genes, lncRNAs, pseudogenes, and other biotypes covered in our datasets in Figure 2C and Extended Data Figure 7F. We then compare the protein-coding genes in our datasets to those reported in mESCs in PMID 2903319 in Extended Data Figure 8G (H3K9me2 domains) and Extended Data Figure 7F (LADs). We do detect about 25% more H3K9me2-resident protein-coding genes than previously reported, which could be due to method used (CUT & RUN here, versus ChIP-seq previously), sequencing depth, or additional reference genome annotations since this previous study was completed in 2017. We have also increased the stringency of our CUT & RUN domain calling strategy. We realized that our gene calls were modestly inflated by calling any gene partially covered by a domain as “within” a domain. Now, only genes that are at least 90% within a domain are reported as resident genes. This more stringent filtering decreased gene calls by a modest amount (10-20%), probably by removing a small number of genes that straddle domain borders. With these adjustments, we detect comparable numbers of protein-coding LADs and comparable LAD genome coverage (1081 Mb, or approximately 40%, see Fig. 2B) to this previous study.

3. western blots of H3K9me2 across all the various conditions and cell types are necessary to support that H3K9me2 levels are not changed in ESCs across genetic KO and that H3K9me2 levels increase upon differentiation.

We now show that H3K9me2 levels remain consistent across mESC genotypes by two independent methods: (1) quantitative immunostaining of H3K9me2, shown in Figure 1E, and (2) by quantitation of total yield by spike-in-controlled H3K9me2 CUT & RUN, shown in Figure 2J. Here, direct comparison of the ratio between spike-in DNA and Cut & Run library DNA indicates consistent levels of H3K9me2 across genotypes.

We show that H3K9me2 levels increase during the transition from mESC to EpiLC by quantitative immunostaining (Figure 5K). We now include a Western blot of WT, TKO, and QKO mESCs and EpiLCs in Supplementary Figure 14E-F.

4a. The expression analysis could benefit from a more focused approach and additional analysis, as there is likely a mix of primary and secondary effects observed in the data as they are

presented currently. Firstly, are the same genes which are upregulated in the TKO versus WT also upregulated upon LBR KO or are they distinct? How does this compare to those identified as upregulated in the QKO? A similar analysis should be performed for down regulated genes.

We thank the reviewer for raising this point. We have overhauled our gene expression analyses. First, we no longer focus on QKO versus TKO expression comparisons, as we realized that this comparison magnified small differences between these genotypes while minimizing differences between each mutant genotype and WT cells. We now focus on mutant versus WT comparisons.

We now include a 3-way Venn plot showing the overlap in DEGs between LBRKOs, TKOs, and QKOs. Overall, the genes dysregulated in TKO and LBRKO mESCs do not overlap with each other, but do share more extensive overlap with QKO DEGs. Importantly, however, there is a major synergistic effect of depleting both the lamins and LBR; for instance, while 729 and 672 genes are upregulated at least 2-fold in LBRKO and TKO mESCs, respectively, > 2900 genes are upregulated at least 2-fold in QKO mESCs.

We now primarily use GSEA to more comprehensively identify co-regulated genes in pathways using the entire DESEQ2 output and focusing on mutant versus WT comparisons. In TKO and QKO mESCs, GSEA identifies morphogenesis, proliferation, and development-linked terms amongst downregulated genes, suggesting that these cells may have altered differentiation potential. Interestingly, GSEA identifies barely any GO terms that include upregulated genes, even though upregulated genes are numerous. We interpret this outcome to mean that upregulated genes are not unified by a particular gene regulatory network, and may instead relate to specific chromatin states that are more permissive to transcription upon displacement from the lamina. For this reason, we focus extensively on relationships between LADs, H3K9me2, and gene expression in our revised expression analyses in Figure 3, 4, and 7. We compare genes dysregulated upon acute LBR depletion to genes dysregulated in stable KO clones to infer primary *versus* secondary effects of heterochromatin displacement.

4b. Secondly, the conclusions focusing on any role of H3K9me2 would be better supported if the authors focused on those genes in H3K9me2 domains in WT cells (or close to them on the genome) and their change in expression (and localization as suggested in point 1) in the various genetic and cellular conditions. Thirdly, how do the authors reconcile genes which are upregulated or downregulated in the QKO have elevated levels of H3K9me2 compared to those with no change. If H3K9me2 plays a repressive role, shouldn't there be a difference in H3K9me2 levels between up- and downregulated genes (Fig 3D)? Fourthly, to support the conclusion that H3K9me2 modification unable to support transcriptional repression, the authors should demonstrate that under the same H3K9me2-domain calling parameters the genes enriched for H3K9me2 have minimal or significantly lower transcriptional activity than H3K9me2-enriched genes in TKO, LBRKO, and QKO. Is the lack of H3K9me2-repressive function is observed in QKO only?

We thank the reviewer for this clarifying suggestion, which we think improves the manuscript. We leverage our new LAD identification data to evaluate changes to gene and TE expression within WT LADs and within WT H3K9me2 domains. We find that LADs are overwhelmingly H3K9me2-modified in WT mESCs (Fig. 2B) and are progressively lost in LBRKO, TKO, and QKO mESCs (Fig. 2E-G; Extended Data Fig. 10A). While WT LADs lose association to the nuclear periphery in mutant mESCs, they remain H3K9me2-modified (Fig. 2K). As suggested by the reviewer, we now focus our gene expression analysis on WT LADs by plotting the fold change in gene expression versus fold change in LAP2 β :genome contacts in LBRKO (Extended Data Fig. 10D), TKO (Extended Data Fig. 10E), and QKO mESCs (Fig. 3D). This analysis indicates modest

numbers of dysregulated WT LAD genes in LBRKO and TKO mESCs that are equally distributed between up- and down-regulation. In contrast, WT LAD genes in QKO mESCs are nearly twice as likely to be upregulated as downregulated (Fig. 3D). Because these loci remain H3K9me2-modified, we interpret this result to mean that H3K9me2's repressive function on WT LAD genes is weakened only in QKO mESCs. Clearly, however, this effect is partial, as many WT LAD genes are unaffected by displacement from the nuclear periphery in QKOs (see response to major point 1b above). We infer that displacement from the nuclear periphery makes loci more *permissive* to transcription but is not sufficient to induce transcriptional activation.

As the reviewer notes, the data indicate that both upregulated and downregulated genes are H3K9me2-modified. In fact, we find that upregulated LAD genes have moderate levels of H3K9me2 (Fig. 3G) and an intermediate frequency of LAP2 β contact in WT mESCs (Fig. 3H), while downregulated genes have high levels of H3K9me2 (Fig. 3G) and high levels of LAP2 β contact in WT mESCs (Fig. 3H). Both groups of genes remain H3K9me2-modified but lose LAP2 β contact in QKO mESCs (Fig. 3G). We note that downregulated genes are very lowly transcribed in WT LADs, but their transcription further decreases in QKOs; we now show H3K9me2, LAP2b, and RNAseq tracks for an example gene, *Mgst1*, in Figure 3F. We speculate that strongly H3K9me2-modified genes are able to maintain (or even modestly enhance) repression in the absence of tethering to the lamina, perhaps by self-association and coalescence into heterochromatic foci. One intriguing possibility is that these strongly H3K9me2-modified genes relocate to heterochromatic nucleolar-associated domains (NADs), as recent work has shown that some highly H3K9me2-modified loci shuffle between LADs and NADs, while moderately H3K9me2-modified loci are more restricted to LADs (PMID 35304483).

5. A PCA showing similarity of RNA-seq replicates across conditions should be provided.

PCA plots for all RNAseq experiments are now included in Extended Data Figure 11.

6. The authors should complement the differentiation studies with additional experiments. They should include marker analysis to more definitively determine differentiation efficiency and the differentiation defect observed in the lamin + LBR KO. Relatedly, the authors argue that there is an expansion of H3K9me2 upon epiblast differentiation. They observe this in the lamin + LBR KO (Fig. 6F, S9J), but the vast majority of cells in these cultures are not epiblast cells based on the analysis provided?

We now include FACS sorting for Nanog (high in mESCs) and Otx2 (high in EpiLCs) to compare differentiation efficiency across genotypes (Fig. 5C and Extended Data Fig. 13). These analyses indicate that most of the surviving cells in each genotype are Otx2+, indicating they have transitioned into the EpiLC state, and Nanog-low, indicating that they have transitioned out of the mESC state.

While QKO mESCs generate far fewer viable cells upon differentiation stimulus than do other genotypes (Fig. 5B), the surviving cells are Otx2-positive (Fig. 5C) and do transcriptionally resemble EpiLCs (Fig. 7I): they have shut off naive pluripotency genes and induced primed pluripotency genes. However, they do not resemble EpiLCs in one key feature: they also co-express markers and morphogens of other lineages, including various Wnt ligands and markers of primitive endoderm.

Minor Points:

7. It would be helpful if the authors annotated the figures a bit more.

We have made an effort to add descriptive subheadings to increase the readability of the figures.

8. In the first paragraph of the introduction, the authors state, “These observations have led the prevailing model that peripheral heterochromatin positioning promotes the establishment of cell fate by repressing alternative fate genes. However, this model has not yet been tested, and the mechanism by which the nuclear periphery confers repression on associated chromatin remains unknown.” This statement should be edited. The role of peripheral chromatin positioning in mediating alternative fate has been tested to some degree - (PMID: 34737442, 33529599 and 26607792).

We thank the reviewer for pointing out this oversimplification. Gasser and colleagues (PMID 34737442 and 26607792) showed in *C. elegans* that the INM protein Cec-4 tethers heterochromatin to the nuclear periphery in embryos, but puzzlingly, its removal displaced heterochromatin without altering gene expression (PMID 26607792). This work did show that Cec-4 facilitates the ectopic induction of an alternative cell fate, but did not explore whether this related to abnormal regulation of cell fate gene expression. In subsequent work, Gasser and colleagues explored the influence of H3K9me itself on repression of tissue-specific genes (PMID 34737442); this work is relevant to our interest in H3K9me’s repressive function but does not directly speak to the role of the nuclear periphery in mediating that function.

We agree with the reviewer that prior work on laminopathy mutations and cell fate from Raj Jain (PMID 33529599) as well as Joseph Wu (PMID 31316208) is relevant here. While these studies do not demonstrate a large-scale displacement of heterochromatin from the nuclear periphery in cells harboring laminopathy mutations, they do show that local detachment of some loci can allow lineage-irrelevant gene expression and we have modified the text to cite these studies as follows: “Disease-linked mutations to the lamin A/C protein weaken both peripheral positioning and repression of lineage-irrelevant genes (refs). These observations have led to the prevailing model that peripheral heterochromatin positioning promotes the establishment of cell fate by repressing alternative fate genes. However, the mechanism by which the nuclear periphery confers repression on associated chromatin remains unknown.”

9. It would be helpful if the authors better defined the number of genes used in their analysis and consistent across the manuscript (protein coding? lncRNA?)

We now report the gene biotypes identified within H3K9me2 domains (Extended Data Fig. 8F) and within LADs (Fig. 2C).

10. The data in S7J does not appear statistically significant - the text should be edited so this is more clear.

The lack of statistical analysis in Figure S7J (now Extended Data Fig. 15J) was an oversight. We have now included statistics, which indicate that NE/nuclear chromatin intensity ratios differ significantly between WT and TKO as well as between WT and QKO conditions (as judged by ANOVA followed by Dunnett’s T3 multiple comparisons test, which makes comparisons between control group (WT) and each other genotype without assuming equal variance for each sample). The figure and legend have been updated accordingly.

11. How do the authors reconcile the large effect in repeat elements expressed in QKO versus the WT given the relatively few number of repeat elements unregulated in the QKO versus TKO and TKO versus WT?

In the revised manuscript we focus our gene expression analysis on mutant versus WT comparisons and have omitted QKO versus TKO comparisons for clarity. The phenomenon that the reviewer noticed in our initial submission was most pronounced in our summary of TE classes upregulated more than 2-fold in Figure 4B. There are a number of TE classes that are upregulated to some sub-threshold extent in TKO mESCs and upregulated more highly in QKO mESCs. In fact, it is apparent upon inspection of MA plots in Figure 4C and Figure 4D that there is a broad trend toward derepression of TEs both above and below our 2-fold change cutoff. Thus, the fold change difference between QKO and WT may be larger than 2-fold while the fold change difference between either TKO and WT or QKO and TKO may each be less than 2-fold. We recognize that the QKO versus TKO comparison was confusing and have omitted it from the revision.

Altogether these analyses show that TKO mESCs permit expression of some TEs which was not previously detected in PMID 30201095. TE derepression is minor in TKO EpiLCs, however (Extended Data Fig. 18). The implication of these data overall is that QKO cells completely fail to repress TEs, while TKO cells can rein in their expression as they differentiate.

12. Line 189 - “These unregulated genes are significantly enriched for H3K9me2 compared to unaffected or down regulated genes (Fig. 3I).” This conclusion is not consistent with the data shown, as there does not appear to be difference between the unregulated and down regulated genes.

Thank you for pointing out this inconsistency. Upon reflection, we decided to refrain from discussing the level of H3K9me2 modification on downregulated genes in the LBR miR-E condition because of the very small sample size (only 21 genes downregulated). We have adjusted the text as follows: “These upregulated genes...are significantly enriched for H3K9me2 and LAP2 β association compared to unaffected genes.”

13. Lines 281-286. While the authors have made a good attempt to rule out additional functions of LBR and lamins to mediate the differentiation phenotype, it is still not definitive that “H3K9me2 positioning is required for EpiLC viability”. This conclusion should be toned down.

We attempted to test alternative functions of the lamins and LBR to understand the EpiLC survival defect. However, our interpretation of the EpiLC survival defect is also based on previously published data that indicates the deposition of H3K9me2 by G9a/GLP is essential for EpiLC viability (PMID 26551560, referenced in the manuscript). Because our data indicate that H3K9me2 is deposited, but displaced, in lamin + LBR null EpiLCs, we infer that the lamins and LBR influence the essential functions of H3K9me2 in this developmental transition. We agree with the reviewer’s point that we have not directly demonstrated that the lamins and LBR achieve this effect by positioning H3K9me2-modified chromatin to the nuclear periphery. In recognition of this point, we have adjusted the text to state that “H3K9me2 is deposited, but is spatially displaced, in EpiLCs lacking the lamins and LBR. As these mutant EpiLCs fail to survive, H3K9me2 displacement (by deletion of the lamins + LBR) phenocopies the effect of H3K9me2 loss (by deletion of the G9a methyltransferase) in EpiLCs. Therefore, we infer that the lamins and LBR promote the essential functions of H3K9me2 during this developmental transition.” (lines 385-389)

14. It would be helpful for the readers if the authors be more specific describing “the nuclear periphery” throughout. For example, in the abstract lines 27-28 it says, “We conclude that the nuclear periphery controls the spatial position, dynamic remodeling, and repressive capacity of

H3K9me2-marked heterochromatin to shape cell fate decisions.” What exactly do the authors mean by the nuclear periphery? Nuclear periphery proteins? Localization at the nuclear periphery? A layer of the peripheral heterochromatin?

We thank the reviewer for this thoughtful critique. We envision the nuclear periphery as a spatial location within the cell, and the nuclear lamins and other INM proteins (such as LBR) as abundant proteins found in this locale. We refer to the nuclear periphery rather than the nuclear lamina because it is clear that enrichment at this spatial location influences chromatin function even in organisms that lack a lamina (such as yeasts), while non-lamin INM proteins (such as LBR) also have important functions in chromatin positioning.

We have adjusted the sentence in the abstract flagged by the reviewer to be more specific, and now state “We conclude that the lamins and LBR control the spatial position, dynamic remodeling, and repressive capacity of H3K9me2-marked heterochromatin to shape cell fate decisions”. In the initial submission we were more vague with this statement as we did not have data profiling LADs in WT cells or LAD rearrangement/loss in mutant cells. Our new LAP2 β CUT & RUN data resolves that ambiguity and allows us to be more definitive about the major consequences of disrupting the lamins and LBR on heterochromatin organization.

15. Line 164: “...lamin TKO mESCs (where heterochromatin positioning is intact)”. The data showed in Fig. 1M suggests otherwise. There is a significant difference between WT and TKO (as well as LBR KO). The authors should consider changing “intact” to “retain localization at the nuclear periphery”.

This statement has been removed in the revised manuscript. Additionally, LAP2 β CUT & RUN indicates significant decreases in LAP2 β :genome contacts in both the lamin TKO and LBRKO mESCs (Fig. 2E-G), consistent with the reviewer’s astute point that chromatin organization is not completely normal in any mutant genotype.

16. The authors reported 59% (WT ESC), 65% (LBR KO), and 66% (WT EpiLC) genome coverage by the H3K9me2 histone modification. The transition from WT ESC to EpiLCs (59% to 66%) is described as expansion of H3K9me2 across the genome. At the same time LBR KO result is similar changes (59% to 65%). Why are these similar results interpreted differently?

Please see response to Reviewer 3 point 2 below. In the revision we use a simplified 2-state HMM to call H3K9me2 domains, which performs more robustly across cell states and genotypes. This model identifies similar numbers of H3K9me2 domains across genotypes and actually identifies moderately fewer domains in EpiLCs than ESCs. While an increase in bulk H3K9me2 abundance during this transition has been demonstrated by others (including PMID 26551560), and we also see evidence of a bulk increase in H3K9me2 levels by quantitative immunostaining and Western blotting (Fig. 1; Extended Data Fig. 14E-F), whether or not our genomic analyses support the claim that H3K9me2 domains *expand in genomic distance* in EpiLCs depends on the specifics of the domain calling parameters used. For this reason, we no longer focus on the expansion of H3K9me2 across the genome during the transition from naïve to primed pluripotency. Instead, we focus on the major remodeling of H3K9me2 both within and outside of LADs during this transition, and the enrichment of dysregulated genes within new H3K9me2-only domains (KODs) in mutant EpiLCs.

Response to Reviewer 2

1. ...In regards to the readout of their experiments, i.e. the distribution of H3K9me2 PTM. It

remains unclear why the authors consider as is the major silencing mark of the peripheral heterochromatin. There are other not less important marks - such as H3K27me3, H3K9me3, H4K20me3, etc. For instance, the authors claim activation of TEs, which are known to be silenced by both H3K9me2 and H3K9me3. Looking at the references, we guess that the authors selected H3K9me2 mark based on highly debatable data of strictly peripheral distribution of this PTM, which originates from the same single lab [e.g., 10.7554/eLife.49278.001; 10.1126/sciadv.abj3035; etc]. The data from other groups, however, show more internal distribution of this mark [10.1242/dev.127308; 10.1104/pp.114.255737; 10.1158/1541-7786.MCR-14-0474; etc]. Moreover, the latter view is supported by living cell observations demonstrating that constitutive LADs, greatly enriched in H3K9me2, can be found not only at the nuclear but also at the nucleolar periphery, as well as in other internal loci [10.1016/j.cell.2013.02.028].

In this connection, we suggest the authors to present more balanced view on the distribution of H3K9me2 mark and tune down citation of the controversial results from that single group.

There is broad consensus across eukaryotic systems that heterochromatin associated with the nuclear periphery is abundantly H3K9 methylated, although H3K9me2 and H3K9me3 have each been variably reported to associate with the nuclear periphery. We cite examples of each of these, from multiple groups, in yeast (PMID 31883795), in *C. elegans* (PMID 26607792), and in mammals (PMID 23523135 and 29033129). We appreciate the reviewer's point that H3K9me2-modified loci can also be enriched in nucleolar-associated domains (NADs), and now cite recent work indicating similarities between LADs and NADs, which are both H3K9me2-modified (PMID 35304483).

H3K9me2/3 and its depositing enzymes have been strongly implicated in the formation and/or function of LADs. For instance, deletion of the H3K9 methyltransferases MET-2 (mono- and dimethyltransferase) and SET-25 (tri-methyltransferase) impairs anchoring of loci to the nuclear lamina in *C. elegans* (PMID 22939621) while inhibition of G9a impairs the positioning of LAD loci in human cells (PMID 23523135). Conversely, induction of H3K9me2 by a dCas9-G9a fusion promotes association of a locus with the nuclear lamina (PMID 32712024). The transcriptional effects that we see when H3K9me2-modified chromatin is displaced in QKO cells also resemble the transcriptional consequences of a G9a knockout (Fig. 7H, see also response to point 14 below) further strengthening the functional relationship between H3K9me2 and the lamina.

We include several pieces of supplementary data to clarify our motivation for focusing on H3K9me2 in this study. First, we show the strong enrichment of H3K9me2 at the nuclear periphery and its displacement by ablation of the lamins and LBR using both a mouse monoclonal antibody (Abcam ab1220, shown in Figure 1) and with a rabbit polyclonal antibody (Active Motif 39041, shown in Extended Data Fig. 1C-E). Second, we show immunostaining of H3K9me3 and H3K27me3, which do not enrich as strongly at the nuclear periphery in mESCs (Extended Data Fig. 1A-B). Third, we perform H3K9me3 Cut & Run in WT mESCs and analyze its deposition on the genome in relation to LADs (Extended Data Fig. 9). This analysis indicates that H3K9me3 overlaps with a small proportion of LADs, but that many LADs lack H3K9me3. Instead, H3K9me3 covers a much smaller proportion of the genome in mESCs than either LADs or H3K9me2.

Finally, we recently made an observation that we think provides some additional context for the debate of which chromatin modifications are enriched at LADs. When we compared LADs in WT ESCs and EpiLCs, we noticed that H3K9me2 decreases on many LADs during the ESC to EpiLC transition (Fig. 6). This is a major change; LADs and H3K9me2 overlap nearly perfectly in naïve mESCs (Fig. 2B) while only a third of LADs overlap with H3K9me2 in EpiLCs (Fig. 6B). We speculate that the specific chromatin modifications enriched at LADs may be dynamically

remodeled through development and differentiation. Interestingly, however, we notice that EpiLC LADs remain strongly repressive even when H3K9me2 is removed (Fig. 6D), indicating that stable repression of LADs can be maintained by other marks / other means.

2. In addition, to validate the choice of experimental readout and demonstrate, that the heterochromatin in general and not only H3K9me2-positive chromatin, is displaced or not displaced – we suggest to perform immunostaining of several other heterochromatin markers at the all three conditions - TKO, LBR-KO, QKO.

We have now identified nuclear envelope-proximal chromatin by LAP2 β CUT & RUN in all genotypes, which reveals global displacement of LADs from the nuclear periphery in the absence of the lamins and LBR (Fig. 2E-G). While we cannot rule out the possibility that these displaced loci harbor other chromatin modifications, they are extensively H3K9me2-modified (Fig. 2B). TEM also indicates that heterochromatin is much less prominent underneath the nuclear periphery in QKO cells (Fig. 1L-M), consistent with a major effect on chromatin positioning. We do not claim that either of these pieces of data indicate that only H3K9me2-marked chromatin is displaced. However, we do identify strong relationships between LADs and H3K9me2 in mESCs (Fig. 2A-B), and show that LAD genes and LAD-resident TEs that are dysregulated are H3K9me2-modified (Fig. 3G, Fig. 4G, Fig. 4J), indicating that disrupting the lamins and LBR affects the repressive function of H3K9me2.

3. The authors show a clear release of H3K9me2 marked chromatin by microscopy. However, it is entirely untested which nuclear envelope-genome interactions have been altered or how widely this occurs through the genome. Are all nuclear envelope-heterochromatin broken? Conversely, does euchromatin replace heterochromatin at the periphery as in rod cells? If so, do euchromatic loci show repression when repositioned and does this explain why many loci are downregulated in the QKO mutant? Cut&Run or DamID from Lap2 β in QKO cells should reveal this. This could also be complimented by staining for several euchromatin marks, like acetylated histones, H3K4me3 or H3K36me3. In short, these questions can be readily answered by the authors and are necessary to make the manuscript a true advance.

We thank the reviewer for raising this point. We have validated the use of the inner nuclear membrane protein LAP2 β as a fiducial marker of LADs/INM-proximal chromatin (see Extended Data Fig. 5 and 7) and performed spike-in-controlled CUT & RUN. In WT mESCs, LB1 domains, LAP2 β domains, and H3K9me2 domains are highly correlated (Figure 2B). In mutant mESCs, LAP2 β :genome association progressively and globally decreases as indicated by spike-in-normalized CUT & RUN DNA yield (Figure 2E) and within WT LADs as indicated by spike-in-normalized profile plots (Figure 2G). Meanwhile, H3K9me2 remains present on the same loci and at similar levels in each genotype (Figure 2H-K). Therefore, we conclude that deletion of the lamins and LBR causes nearly global spatial displacement of H3K9me2-modified chromatin from the nuclear periphery.

While loss of heterochromatin association with the INM is the major outcome observed in these experiments, we do also note a slight increase in frequency of non-LAD, euchromatic regions with the INM in mutant EpiLCs (Fig. 2F, see left side of plots). However, the signal in these regions is very low - far lower than the signal observed within WT LADs. Downregulated genes are distributed across WT nonLADs, LADs, and KODs (Fig. 3I). Altogether, these data are not consistent with the possibility that *de novo* association of euchromatin with the nuclear periphery drives downregulation. Instead, we speculate that downregulated genes are secondary consequences of LAD / H3K9me2 dysfunction.

3. Finally, it is not clear how many clones are used as replicates in the analyses throughout the paper. Figure 5E states that at least two independent clones were tested by IF and the materials and methods indicate 10 homozygous KO clones were generated by CRISPR. How many clones were tested as replicates for genomics assays, including RNA-seq and Cut&Run, in figures 2, 3, 4, 6 and 7? Considering the significant impact of these findings, we feel it is essential that the trends are reproducible across at least 2 clones. This is because the selection of specific ESC sub-clones can itself introduce significant differences in chromatin and gene expression.

We apologize for the lack of clarity about the number of clones used. While we did initially isolate multiple QKO clones, only one survived thaw after initial freeze. This single clone is the major clone used for all experiments. A second attempt to isolate additional QKO clones yielded 8 additional homozygous QKO clones, but each clone failed to survive and expand during sub-culturing. We infer that disruption of the lamins + LBR severely impairs viability and/or that the surviving QKO clone acquired additional adaptations that enabled its survival.

We recognize the limitation of showing data from only one clone and have taken several steps to support our findings in QKO cells. First, we include supplemental data showing that acute (nonclonal) CRISPR knockout of LBR in lamin TKO mESCs induces chromatin reorganization (Extended Data Fig. 1E-H). Second, we use the miR-E system to inducibly deplete LBR in lamin TKO cells, which shows spatial displacement of H3K9me2-modified chromatin by microscopy (Extended Data Fig. 4), consistent with the QKO; shows a strong bias toward de-repression by RNAseq of a small number of genes that overlap with the much larger group of de-repressed genes in QKO cells (Figure 3J, Extended Data Fig. 10I); and shows an acute EpiLC differentiation defect (Figure 5E). Third, we have performed rescue experiments in the QKO background and show that re-introduction of Halo-LBR rescues H3K9me2 positioning (Figure 1F-H). We now include additional rescue data showing that Halo-LBR rescues EpiLC differentiation of QKO ESCs (Figure 5D).

In ongoing work, we are performing LBR mutagenesis and rescue experiments in QKO cells by generating stable cell lines overexpressing Halo-tagged LBR constructs. We have evaluated rescue by RNAseq in EpiLCs, where we are comparing fold change (QKO/TKO) transcriptomes to fold change in rescue. For rescue comparisons, we have evaluated (A) (QKO + Halo-LBR/QKO) as well as (B) (QKO + Halo-LBR/QKO + Halo-NLS). We share example correlation plots from these RNAseq experiments here (n = 4 replicates per condition), which show a strong anticorrelation. This means that re-introduction of Halo-LBR in QKOs significantly alters transcriptomes to more closely resemble the TKO state (indicating an effective rescue of transcriptional changes). This effect is significant both in comparisons (1) and (2) but is dampened in the latter, because overexpression of Halo-NLS itself alters transcription. Notably these lines were made by PiggyBac integration of rescue constructs and are nonclonal populations. We do not include these data in the manuscript because we are planning to repeat these experiments with single-copy integration of rescue constructs, and they will be included in a future structure/function follow-up study. Nevertheless, we share them here to further underscore the specificity of the effects we see in QKO cells to the loss of LBR.

Reviewer Figure 1

A. Halo-LBR rescue vs. QKO

B. Halo-LBR rescue vs. Halo-NLS control

Minor Points:

4. Lines 46-47, 50-51: heterochromatin is positioned also at the nucleoli periphery and serves as a silencing compartment similarly to the nuclear periphery; in mouse cells, it also surrounds chromocenters built by major satellite repeat.

We did not intend to imply that heterochromatin is exclusively found at the nuclear periphery, merely that it is enriched there. In the revised manuscript, we do now discuss the overlap between nucleolar-associated domains (NADs) and LADs in the discussion.

5. Line 56: it should be mentioned, that the paper Smith et al 2021, postulates a controversial view on distribution of histone modifications in retinal cells – compare with other publications [10.1007/s10577-013-9375-7 or 10.1016/j.neuron.2019.08.002].

We cited Smith et al Science Advances 2021 here to support the statement that H3K9me2 is not always repressive. We are aware of the controversial elements of this publication and in the revised text have limited our citations of it. Relevant to the specific point of H3K9me2's variable impact on repression, we now cite work showing that H3K9me2 domains are more permissive to transcription than H3K9me3-modified domains (PMID 28682306), and work showing that H3K9me2 modification and lamina association correlate with stronger repression than H3K9me2 modification alone (PMID 34023908).

6. Line 86: the reference 36 is not correctly cited - Zheng et al showed that upon depletion of all lamins LADs decondense or detach from the nuclear periphery.

We cite both an earlier study by Amendola and van Steensel (PMID 25784758) that reports generally preserved LAD organization (detected with emerin-DamID) in lamin B1/B2 DKO mESCs treated with Lamin A/C RNAi, and cite the Zheng 2018 study (PMID 30201095) that reanalyzed the same dataset and detected decreases in some LAD contacts as well as increases in other LAD contacts (see Fig. 4 of the Zheng study). The Zheng et al. study shows that four candidate LAD loci become decompacted by FISH, and that two LAD loci increase their distance from the nuclear periphery (also by FISH). This study does show that LADs are affected measurably by lamin ablation by these candidate locus FISH analyses and by an in-depth analysis of chromatin

compaction by Hi-C. However, overall, there is no evidence from these previous studies that depletion of the lamins causes a major global change to LAD enrichment at the nuclear periphery.

To more accurately report the nuanced effect of lamin depletion in mESCs, we have altered this sentence to state “mESCs lacking all lamin isoforms are viable and pluripotent and exhibit modest changes to peripheral heterochromatin positioning, genome folding, and gene expression.”

7. Line 100: in Fig.1, all panels with images have to be supplemented by DNA staining of the corresponding cells – this will allow to appreciate distribution of heterochromatin, including chromocenters, and other nuclear features. This comment is valid for all other figures in the MS showing nuclear images - Fig.5, S3, S7.

To make effective use of precious main figure space, we show and quantify DNA staining in mESCs and EpiLCs in separate figures (Extended Data Fig. 3 and 15, respectively). Additional examples of DNA staining in WT and QKO mESCs appear in Extended Fig. 1 as well. These images show the distribution of heterochromatin throughout the nucleus and how it is affected in mutant cells.

8. Could the authors comment on the increased internal H3K9me2 staining in LBR-KO visible in the 1C?

While this image does show more nucleoplasmic signal than the other genotypes, our quantification of H3K9me2 signal as a function of distance from the nuclear periphery indicated a significant difference ($p < 0.05$) between WT and LBRKO cells in only 3 of 25 “shells” (see Figure 1H and legend). TEM analysis indicates a significant decrease in the intensity ratio between peripheral and internal heterochromatin (Fig. 1I-M), which may reflect a partial decompaction effect of LBR KO.

9. Line 101: in Fig.S2B and Fig.S2G, increase of heterochromatin lamps inside the LBR-KO nuclei is obvious. E.g., we do not see a difference in DNA distribution between Fig.S2B and Fig.S2D. We are aware that the authors undertook quantification of the DNA signal (Fig.S2E), however, in our view the correct analysis based on a single optical section is difficult for mESCs because of the AT-rich chromocenters and large nucleolus. Therefore, we suggest the authors to prepare a Supplementary Information file showing mid-confocal sections of DNA staining for all collected nuclei and for all conditions.

We agree with the reviewers that LBR knockout alone seems to affect chromatin organization within the nucleus (see our response to point 8 above). The images shown in Extended Data Fig. 3 are in fact mid-confocal sections, and the quantification (Extended Data Fig. 3E) indicates a significant difference between DNA positioning in “shells” 14-18 and 21-25 of LBRKO mESCs compared to WT mESCs (see legend). We do agree with the reviewer’s astute inference that heterochromatic chromocenter organization is altered in mutant genotypes, in particular in LBRKO and QKO mESCs. However, we chose to emphasize the major changes to genome organization that we detect by LAP2 β CUT & RUN (Fig. 2E-G) rather than using microscopy data to make this point.

10. Line 118: Fig.S3F: in this figure, as examples for TKO+LBR miR-E the pairs of nuclei are shown (two panels in the right low corner). These pairs are apparently sister cells in early G1, the stage when chromatin is partially condensed and nuclear architecture is not yet established. They have to be replaced by examples of later cell cycle stages.

The reviewer has sharp eyes! We have replaced these examples with other cells that appear based on size and morphology to be more representative of mid-interphase chromatin organization following acute depletion of LBR.

11. Lines 121-123: Could the authors comment whether the movement of H3K9me2-positive heterochromatin inside and outside occurs within one cell cycle or after cell division? In other words, do cells have to go through mitosis with following nuclear formation in order to rearrange heterochromatin?

Unfortunately, it is very difficult to be conclusive on this point with the tools we were able to generate. The miR-E system can achieve strong – though incomplete – knockdown of LBR by 48 hours, which is too slow to make inferences about cell division. Related to the reviewer's above point, we may find examples of sister nuclei with abnormal chromatin organization but we cannot know with certainty whether LBR was depleted to induce this effect within the last cell cycle. We attempted to engineer LBR degron lines that would make it possible to degrade LBR in less than one cell cycle and also to visualize LBR directly by live imaging. While we were able to generate these lines, basal degradation and/or tag effects impaired the heterochromatin tethering function of LBR, such that heterochromatin was displaced even without degron induction. We intend to revisit this approach with new degron tag variants and/or different tag integration sites in the future.

12. Line 140-141: The distribution of H3K9me2 appears similar between all samples. However, absolute levels of signal cannot be compared in the absence of a spike-in control. The materials and methods states this is the case with all Cut&Run experiments, but the text implies Spike-in was only performed in Figure 7. Can the authors demonstrate that there is no difference in overall H3K9me2 signal in mutant vs wildtype ESCs via Spike-in normalised Cut&Run? We feel this is essential to be able to make this claim.

All Cut & Run experiments were performed with a spike-in control, and this is stated in the corresponding Results sections for Fig. 2 and Fig. 6. To increase clarity, we now state that a spike-in control was used in the corresponding figure legends as well. In the revised manuscript, we show that the ratio between spike-in DNA and CUT & RUN H3K9me2 library DNA is consistent across genotypes in Fig. 2J; and show by profile plots of spike-in-controlled RPKM signal that H3K9me2 levels are comparable on WT H3K9me2 domains (Fig. 2I).

13. Line 183-192: There is a strong discordance between number of mis-expressed genes in QKO knockout cells versus those with the inducible LBR knockdown. Naturally this could just be the result of incomplete loss of LBR or a time delay before H3K9me2-repressive fails. Could the authors confirm that the genes that are mis-expressed in knockdown cells are also mis-expressed in knockout cells (and in the same direction)?

We now include Venn diagrams showing the overlap of DEGs in the inducible LBR knockdown conditions versus in QKO cells in Extended Data Fig. 10I-J. The small number of DEGs we observe upon inducible LBR knockdown do overlap extensively with the larger group of genes dysregulated in QKOs. The much larger effect in QKOs could indicate secondary effects of the KO and/or the difference between partial and complete loss of LBR. LBR knockdown is not complete in the miR-E condition, as is now shown in Extended Data Fig. 10H.

14. Related to this, what is the overlap between gene mis-regulation in QKO cells and those where G9a/GLP is inactivated? A difference in the latter groups of genes may indicate that some H3K9me2-marked loci are more sensitive to LAD loss than others. Alternatively, if no overlap is

observed, this would indicate that the connection between lost lamina-association and gene expression is not intrinsically linked to H3K9me2.

We thank the reviewer for this insightful suggestion. We had noted the dysregulation of some well-described G9a target genes in our datasets (*Mage* genes, *Rhox* genes, etc; described in PMID 15774718). Inspired by this suggestion, we undertook a more systematic analysis by comparing our DEGs in QKO EpiLCs to a published RNAseq dataset of G9a KO mouse epiblasts at E6.25. (As we note in the text, G9a K.O.s exhibit a similar phenotype to our QKOs – poor survival of epiblast cells.) We now show that genes upregulated in QKO EpiLCs are also upregulated in G9a KO epiblasts (Fig. 7H). This strengthens our interpretation that a functional relationship exists between the lamina and H3K9me2-mediated repression in early development.

15. Line 284: Fig.S7J: are the values of EM density in the graph significant? In our view, such evaluation of chromatin density after TEM makes no sense because areas in vicinity to nuclear envelope and in nucleoplasm can unavoidably be selected arbitrary.

The lack of statistical analysis in Extended Data Fig. 15J was an oversight. We have now included statistics, which indicate that NE/nuclear chromatin intensity ratios differ significantly between WT and TKO as well as between WT and QKO conditions (as judged by ANOVA followed by Dunnett's T3 multiple comparisons test, which makes comparisons between control group (WT) and each other genotype without assuming equal variance for each sample). The figure and legend have been updated accordingly.

We apologize for the lack of clarity in how the TEM images were quantified. We have now clarified the methods, which were as follows: for each cell, 40 non-overlapping ROIs beneath the nuclear envelope and 40 non-overlapping ROIs inside the nucleoplasm were manually selected. The average signal in each of these squares was used to calculate an average NE/nucleoplasm intensity ratio per cell. The N in the TEM quantification plots (Fig. 1M and Extended Data Fig. 15J) reflects the number of individual nuclei analyzed in this way. While we acknowledge that manual selection of coordinates could introduce bias, we made an effort to sample each nucleus densely with ROIs to make the analysis as unbiased as possible.

16. There is no reference to the Fig.S8 through the entire MS.

In consultation with the editorial staff, we may move these uncropped TEM images (now Extended Data Fig. 16) into Source Data.

17. Line 301: Plot 6C indicates that the vast majority of the genome (18729 genes) become incorporated into H3K9me2 regions. This far exceeds previous estimates, including cell-types that arise after EpiLCs. Do the authors have an explanation for this? We feel it is necessary to rule out technical artefacts to make this claim. Could the authors also show the IgG control from both cell-types and confirm there is also not the same increase in non-specific signal in EpiLCs? For the reviewers, it would be key to see the IgG control alone, as well as a log2 ratio.

We apologize for causing confusion on this point - please see response to Reviewer 1 point 2 above. In short, we have made some improvements to our H3K9me2 domain HMM calling methods; and importantly, our datasets include both protein-coding and noncoding loci while some previously published datasets report only the numbers of protein-coding genes residing in domains. (We mapped our data to the GRCm39 mouse genome assembly, which now includes over 70,000 coding and non-coding loci according to the Ensembl database.)

IgG controls are not typically performed for CUT & RUN experiments, and CUT & RUN signal is not typically shown as a log2 ratio to a negative control. This is because, compared to ChIP-seq, the background generated by CUT & RUN is far lower. Nevertheless, we have run IgG controls and show example tracks in WT ESCs and WT EpiLCs here. For comparison and at the same scale, we show Lamin B1 CUT & RUN samples that were performed in parallel.

Reviewer Figure 2

18. Line 425-428: these conclusions by Smith et al are highly debatable and in conflict with other existing literature about H3K9me2 as a repressive mark of heterochromatin positioned also inside the nucleus, e.g. around nucleoli. As for mouse rod nuclei, to which the authors refer themselves, other researchers have shown absence of H3K9me2 at the nuclear periphery and its presence in the central heterochromatin blob [10.1007/s10577-013-9375-7]. We believe, the authors have to refer to and discuss literature in a more balanced manner.

Please see our response to point 5 above.

Response to Reviewer 3

Major points:

1. The light- and EM-microscopy data show a striking repositioning of heterochromatin away from the nuclear lamina. This observation would benefit from complementary analysis through genomic means. For instance, through DamID-seq, pA-DamID or CUT&RUN against NL-components such as Emerin or Lap2B.

Thank you for raising this point; we now include LAP2β CUT & RUN. Please see our response to Reviewer 1 point 1 and Reviewer 2 point 3 above.

2. The authors describe using a 4-state Hidden Markov Model. However, a justification for choosing this number of states is lacking. This is significant because the states are the basis for subsequent analyses on K9 decoration and gene expression. The authors should 1) explain their choice and methods in more detail, 2) show the other two states in the figure panels as well, and 3) compare the overlap between the states and blacklisted regions rigorously.

We apologize for the lack of justification and methods regarding our HMM analysis. In response to the astute points raised by the reviewer, we have overhauled our HMM analysis to be more in line with other genomic analyses of LADs and other large chromatin domains and have updated the methods section with more details. First, instead of relying on a fourth state for blacklisted regions, we used a published BED file (PMID: 37067481) to filter out blacklisted regions in the

mm39 mouse genome before HMM analysis. Second, we assessed the fit of 2- to 5-state models to the same training data by calculating the Akaike information criterion (AIC) and Bayesian information criterion (BIC) values (Extended data Figure 6B, 7B, 8B). Furthermore, we calculated the difference between these metrics with increasing the number of states (Extended data Figure 6C, 7C, 8C) as has been done previously (PMID: 36691074). While increasing the number of states lowers AIC and BIC metrics, these are marginal differences even between the 2-state versus 3-state model, which is less than half of the 2-state AIC or BIC base values. Additionally, we now show the overlap in HMM domains called between replicates in 2-state models of each CUT & RUN dataset (Extended data Figure 6E, 7E, 8E); we found that domain calls overlapped across replicates significantly better when a 2-state model was used than when a 3-state model was used. Since the goal of our study was to identify true CUT&RUN signal of H3K9me2, LMNB1 and LAP2 β versus background signal, we decided to take a conservative approach and use a 2-state HMM model.

3. Fig 6 is dependent on the 4-state HMM, which the authors rightfully mention that the HMM is found to be ineffective in describing the K9me2/3 domains in EpiLCs. This should be investigated in greater detail in light of our previous comments.

As described above, we now use a simplified 2-state HMM to call H3K9me2 domains, which performs more robustly across cell states and genotypes. This model identifies similar numbers of H3K9me2 domains across genotypes and actually identifies moderately fewer domains in EpiLCs than ESCs. While an increase in bulk H3K9me2 abundance during this transition has been demonstrated by others (including PMID 26551560), and we also see evidence of a bulk increase in H3K9me2 levels by quantitative immunostaining and Western blotting (Fig. 1; Extended Data Fig. 14E-F), whether or not our genomic analyses support the claim that H3K9me2 domains cover a greater genomic distance in EpiLCs depends on the specifics of the domain calling parameters used. For this reason, we no longer focus on the expansion of H3K9me2 across the genome during the transition from naïve to primed pluripotency. Instead, we focus on the major remodeling of H3K9me2 both within and outside of LADs during this transition, and the enrichment of dysregulated genes within new H3K9me2-only domains (KODs) in mutant EpiLCs.

4. The analysis of genes upregulated upon QKO is cursory and lacks depth. It would be interesting if the authors could look more into gene regulatory networks, regulation of major transcription factors and classes of genes that are affected upon manipulation in the context of genomic location. Now, too often, the narrative is dictated by cherry-picked examples and GO-term analyses are lacklustre.

Thank you for raising this point. We have overhauled our gene expression analyses; please see our detailed responses to Reviewer 1 point 4a and 4b above.

5. It would be interesting to cross reference the differentially expressed genes upon the knockout and the acute RNAi-mediated knockout condition.

We now include Venn diagrams showing the overlap of DEGs in the inducible LBR knockdown conditions versus in QKO cells in Extended Data Figure 10. The small number of DEGs we observe upon inducible LBR knockdown do overlap extensively with the larger group of genes dysregulated in QKOs.

6. It has already been shown that LBR and Lamin A/C depletion alone can cause massive chromatin repositioning. It is unclear if the QKO-results are thus driven by absence of LBR and Lamin A/C or that there is a true (novel) effect of the quadruple depletion. It would be very

interesting to see what the contribution of the B-type lamins is to the phenotype, or whether the results are caused primarily or exclusively by the absence of LBR and Lamin A.

We agree that it will be interesting to explore the unique contributions of lamin isoforms and LBR to chromatin positioning in the future, and we intend to do this by performing rescue experiments in QKO cells and also by generating our own Lamin A / LBR DKO mESCs. Our attempts to reintroduce Lamin A into the QKO cells were not interpretable, as Lamin A remained mislocalized in the nucleoplasm (not shown). We think this may be due to Lamin A's dependency on Lamin B1 for its incorporation into the nuclear lamina (PMID 24523288).

7. Looking at the microscopy, detachment is still preferentially located to the NL. An interesting alternative hypothesis is that there is no real detachment but rather decompaction of heterochromatin. The authors should investigate the distinction between detachment and decompaction.

We now include LAP2 β Cut & Run to track contacts between the genome and the nuclear envelope. These analyses demonstrate that pervasive detachment / displacement of LAD loci from the nuclear periphery occurs in QKO mESCs. We speculate that the high proliferation rate of mESCs prevents the complete coalescence of heterochromatin into a single intranuclear focus. Alternatively, additional chromatin modifications and/or distinct H3K9me2/3 binding proteins may promote heterochromatin coalescence in some contexts (such as in some types of neurons) but not in mESCs.

Minor points:

8. The Venn-diagrams in the supplemental show that this manuscript's domains cover much more genomic loci than the data they are comparing to. The authors should show that their calls are not inflated by false positives (i.e., background-signal).

Please see our response to Reviewer 1's point 2 above, and our Reviewer Figure 2 in response to Reviewer 2's point 17 above.

9. The mechanism of LBR binding heterochromatin through HP1 is not mentioned until the discussion. This context would be valuable to mention in the introduction.

We now include this information in the revised introduction.

10. Throughout the manuscript, the authors don't always mention the sample size for presented analyses. For example, Fig. S2E, 4A/B, 7A and more.

Thank you for catching this omission. We double-checked and added missing sample size information to figure legends.

11. The authors find moderate effects on LBR KO, and the TKO conditions but don't discuss them much. Some reflections on these observations could be helpful to give context to their findings.

We now include 3-way Venn diagrams of RNAseq DEGs that make the distinctions between LBRKO, TKO, and QKO DEGs more clear.

12. Figure 3A is counterintuitive: why are the numbers shown for QKO-vs-TKO not the difference between QKO-vs-WT and TKO-vs-WT?

There are some genes that are upregulated to some sub-threshold extent in TKO mESCs and upregulated more highly in QKO mESCs. Thus, the fold change difference between QKO and WT may be larger than 2-fold while the fold change difference between either TKO and WT or QKO and TKO may each be less than 2-fold. We recognize that the QKO versus TKO comparison was confusing and have omitted it from the revision.

13. Figures 3D and 4I mask gene-specific effects and clusters. It would be better to visualise this with enrichment-heatmaps (made with tools such as Deeptools2).

We now focus specifically on subsets of genes with shared chromatin features for these analyses. For instance, in Fig. 3D,G, and H now focus specifically on dysregulated LAD genes while Fig. 7G focuses on dysregulated KOD genes.

14. Some figures could be merged (fig 5 & 6) and some panels are not interesting enough for main text figures. E.g. 2E/F, 3C, 3G (generally the GO terms are uninteresting filler). Fig 6 B/C/D are largely uninteresting.

We respectfully disagree with the reviewer on this point, especially in light of the added data in our revised manuscript. Figure 5 demonstrates the consequences of heterochromatin displacement on differentiation, while Figure 6 details how LADs and H3K9me2 are remodeled during differentiation as well as the disruptive effect of heterochromatin untethering on chromatin remodeling. These are two important new observations that we feel each deserve their due. The revised figures also now contain key new data showing rescue of differentiation (Fig. 5) and LAD and H3K9me2 remodeling during differentiation (Fig. 6).

15. Line 166 onwards, the authors mention that upregulation is the main effect of QKO. Although we agree, there's a considerable fraction of genes downregulated as well. It would be interesting to give these genes more context.

In the revised manuscript, we discuss the strong GO terms identified by GSEA in downregulated genes, and we compare genes dysregulated upon acute LBR depletion to genes dysregulated in stable KO clones to infer primary *versus* secondary effects of heterochromatin displacement.

16. The authors decide to focus on the L1MdA_I LINE1 family. No justification is given why this family is particularly interesting.

We have now used LAP2 β and H3K9me2 Cut & Run to identify TE families strongly enriched in LADs and/or strongly modified by H3K9me2 (Figure 4A). This analysis highlights several L1 LINE elements including the L1MdA_I family. L1MdA_I is also among the TE families upregulated in TKO and QKO mESCs (Figure 4C,D). We then focused our further analyses on the L1MdA_I family due to its large size (>4500 copies in the mouse genome) and the fact that it is the evolutionarily youngest LINE1 family in the mouse genome. This latter point has two implications: (1) its individual copies have high sequence homology making our analysis of chromatin state of individual TE copies more tractable; and (2) L1MdA_I lines retain the ability to retrotranspose, meaning that there are major consequences for the genome when they are de-repressed. We have clarified these points in the revised manuscript and also now include data showing that we can detect the increased expression of the L1-encoded ORF1 protein in TKO and QKO mESCs, which indicates that LINE1 activity is elevated by lamina disruption (Figure 4K-O).

17. Is the morphology of 2i/LIF mESCs normal in QKO conditions? In S1G, they look less dome-shaped?

We do consistently notice that QKO mESCs have slightly less uniform colony borders and are somewhat more spiky than WT mESCs.

18. Fig 5B is missing a scale bar.

Fixed.

19. Fig 6G, H, the authors conclude that K9 is differentially enriched on domains. But it seems to me the enrichment is the same, rather the baseline is just higher?

We do see high H3K9me2 signal in QKO EpiLCs genome-wide; the best genome-wide summary of this is the kernel density plot showing the distribution of all spike-in-controlled signal (Extended Data Fig. 17D). This distribution is right-shifted for QKO EpiLCs compared to other genotypes. We do not see this effect in mESC data (Extended Data Fig. 8J), implying that this effect is EpiLC-specific. Separately, we see high H3K9me2 signal within domains called in WT EpiLCs. In the revised manuscript, we note two phenomena: (1) H3K9me2 is eroded from WT cLADs in WT EpiLCs, but more H3K9me2 remains on these domains in QKO EpiLCs (Fig. 6G); and (2) H3K9me2 accumulates at higher levels in H3K9me2 domains that are deposited in WT EpiLCs (Fig. 6H).

20. Fig 7H colours are super confusing and it is unclear if these are all the genes of interest or that these are cherry-picked.

We apologize for the confusion caused by this figure panel. We analyze a curated list of cell fate regulators and markers for naïve and primed (epiblast) states, as well as for alternative cell fates including the primitive endoderm. We focus on these genes to survey the effects of heterochromatin displacement on the decision to leave naïve pluripotency and enter primed pluripotency, and this focused analysis is motivated by the established role of G9a and H3K9me2 in surviving this stage of development (PMID 12130538, PMID 26551560). We endeavor to make this rationale clearer in the revised text.

21. The authors hypothesize that division is preventing nuclear inversion. Could they test this?

Please see our response to Reviewer 2's point 11 above.

Author Response

We sincerely thank the reviewers and the editorial staff at *NCB* for their thorough and thoughtful evaluation of our study. We are grateful that comments from the reviewers helped us to refine and improve the manuscript. Our specific responses follow in-line below.

Reviewer #1 (Remarks to the Author):

I am satisfied with the authors response and feel the manuscript should be accepted. Congratulations to the authors on a fantastic study!

Thank you!

Reviewer #2 (Remarks to the Author):

The authors have made significant effects to address our concerns and we believe, as we already stated in our first review, that this is a highly timely and exciting paper for the field. In particular, the authors have (a) edited the text to clarify their focus on H3K9me2 and controversies regarding its peripheral localization, (b) profiled breaking genome-nuclear envelope interactions via Lap2 β Cut&Run, (c) re-focused their gene expression analysis.

Thank you!

We have only two remaining concerns that should be addressed before publication.

Lines 102-106: The authors clarify that all QKO data derives from a single clone because others have not survived after they were thawed for expansion. This is naturally a concern, as the single successfully isolated clone may be an aberration. The authors make significant effort to address this by demonstrating similar effects through other strategies (e.g., inducible KDs and rescue experiments). As such, we feel their conclusions are valid. Nevertheless, we feel they should make the fact that only a single clone could be analyzed clearly articulated in the main text, as it is an important caveat that should not be overlooked.

We have adjusted the text (second paragraph of Results) to clarify that “we generated a single lamin + LBR quadruple knockout (QKO) clone and several LBR knockout (KO) clones...” We add the statement that “the survival of only one QKO clone implies that disrupting both the lamins and LBR severely impairs viability.”

We attempt to make clear in subsequent sections where orthogonal approaches (acute nonclonal KO, inducible RNAi, rescue experiments) are used to strengthen our conclusions. For instance, in the following paragraph (around line 120) we state “these observations were confirmed by acute CRISPR/Cas9 targeting of LBR (Extended Data Fig. 1E-H)”.

Fig. 2: The authors did excellent and careful work validating Lap2 β as a target for peripheral DNA mapping by Cut&Run. Nevertheless, we see clear indications of a very large scale increasing in non-LAD contacts in the KO cells. Specifically, LBR-KO cells show a strong redistribution of signal into previously non-LAD regions (left hand side of Fig.2F). This indicates that ectopic signal is diminished and more localized to specific loci in the QKO cells. To us, this strongly suggests that there is at least some form of chromatin inversion and so is an important point to clarify before publication.

Specifically, how extensive is this signal redistribution in each successive mutant? Do the gained regions correspond with a specific chromatin signature (e.g. H3K27ac, H3K27me3, H3K9me3)? Quantifying this globally as well as showing extended figure views of broader chromosome scale plots would be important.

We agree with the reviewer that previously non-LAD regions appear to gain association with the nuclear periphery in mutant mESCs. We now include an additional profile plot showing average spike-in-controlled LAP2 β intensity within domains identified in each genotype (Figure 3D). This analysis shows that in the LBRKOs, new lamina contacts do form with a similar contact frequency to WT LADs. While new “LADs” are also found in TKO and QKO mESCs, they have far lower contact frequency. Therefore, in our evaluation, a significant inversion effect only occurs in the LBRKO mESCs. We interpret the very low overall contact frequency of LADs found in TKO and QKO mESCs to indicate that chromatin position is significantly more random in these genotypes.

We did observe that new LADs/inverted LADs that form in LBRKO cells do not overlap with H3K9me2; it is likely that these inverted domains are instead modified with other marks, but this point is beyond the scope of this current study, and we do not go into this level of detail in the manuscript due to space limitations.

We appreciate the reviewer’s point about showing more zoomed-out chromosome plots and have adjusted all tracks of genomic data to show a ~50 Mb segment of a chromosome.

This going along with our previous minor point 9. We do appreciate all the efforts the authors made for image quantification. However, following saying “seeing is believing”, we still think it would be useful for readers and followers of this exciting study to see a broad collection of mid-sections for WT, TKO, LBR-KO and QKO. Relatively low number of analyzed nuclei – WT (n = 17), TKO (n = 25), LBR KO (n = 26), and QKO (n = 18) – allows to show them all in a Supplementary Figure without new experiments.

We now include these additional requested images in Supplementary Data Figure 2.

Additional Comments in response to Reviewer #3's remaining points:

1. The distinction between chromatin decompaction and NL detachment is underemphasized. The newly added Lap2 β data suggest substantial detachment, yet Ext. Data Fig. 1D/H indicates that H3K9me2-decorated chromatin remains near the NL: while the rim-staining is lost, the chromatin still appears in close proximity to the NL. This distinction is critical, as it affects the interpretation of gene expression changes, H3K9me2 alterations, and other downstream effects.

- In my view, the comment is not valid, because the reviewer refers to H3K9me2 staining, which might be not always successful, however, accompanying DAPI images clearly show that chromatin does dissociate from the lamina in QKO.

2. Due to technical limitations, Lap2 β CUT&RUN data in EpiLCs is missing. As a result, the authors rely on H3K9me2 as a proxy for DNA-NL tethering, which muddles the interpretation of the data and makes it challenging to infer an underlying mechanism.

- It would indeed be more ideal if the authors could successfully show Lap2 β CUT&RUN data in

KO EpiLCs. However, there are 3 points that support the author's conclusions. (1) the authors have already profiled LADs in WT EpiLCs, and so have determined what loci are normally at the nuclear periphery. (2) They also performed IF in EpiLCs, confirming the H3K9me2 is significantly released from the periphery in the QKO condition. (3) They are careful in their wording when describing the CUT&RUN QKO data to avoid describing.

E.g., "We show that H3K9me2 domains are extensively remodeled during the transition from naïve to primed pluripotency (Fig. 6). While LADs are overwhelmingly modified by H3K9me2 in the naïve state (Fig. 2B), H3K9me2 is removed from many LADs in the primed state (Fig. 6A-C). Interestingly, LADs remain strongly repressive despite this remodeling (Fig. 6D), suggesting that other chromatin modifications and/or cofactors maintain repression of primed LADs." Thus, though a limitation, it does not alter the paper's findings.

3. The manuscript lacks mechanistic insight. The analyses remain relatively superficial, missing an opportunity to explore the biological implications of the interesting QKO phenotype in greater depth.

- We disagree with this comment. If the mechanistic insight would be possible, the paper will be published in Nature. This is an important advance in describing the factors that mediate lamina-attachment and so should be made available to the community immediately.

Specific remaining points:

1. The IGV figures display only a few megabases of the genome. Since LADs are large, multi-megabase structures, it would be beneficial to present data across an entire chromosome to better assess the effects of QKO and the overall data quality.

- I agree that showing multiple scales would provide important context as to the local and large-scale effects on LADs.

We have adjusted all genomic data figures to show ~50 Mb segments of chromosomes.

2. The CUT&RUN data for H3K9me3 shows unexpectedly poor overlap with H3K9me2 and LADs. This apparent anti-correlation is puzzling to us, as previous studies have reported significant overlap between H3K9me3 and LADs. The authors need to further clarify and/or discuss this discrepancy.

- H3K9me3 has indeed been reported to correlate with LADs. However, our unpublished data shows that the extent of this correlation strongly varies with cell-type. For example, ESCs show limited "peaky" H3K9me3 while multiple cancer cell lines and differentiated tissues possess broader H3K9me3 that often, but not always, overlaps with LADs. This can be seen in available encode H3K9me3 ChIP-seq data.

However, we agree that this point is not obvious in the field and so the authors should discuss it to prevent confusion in the field.

We agree that this point is interesting and under-appreciated in the field. To more clearly convey this point, we have moved the H3K9me3 CUT & RUN data into main Figure 2 to enable readers to more easily see the direct comparisons between LADs, H3K9me2, and H3K9me3 in mESCs. We do devote a paragraph to these comparisons in the Results section, and cannot go into further detail on this point due to space constraints.

3. In Fig. 2F, some regions gain contacts with the NL upon QKO. These regions could provide insight into the underlying mechanism—are they actively transcribed or repressed? This is an example of a missed opportunity to really dive into the mechanistic underpinnings of their observation.

- We do agree, with this point and highlighted in our own review. The authors stated that these gained peaks were minimal and not significant. However, they appear highly significant to us and so we believe the authors should perform a list a general analysis of their distribution in the genome and the properties of the genes/loci they contain.

Please see our response to Reviewer 2 above; we also agree with the interpretation that new domains do appear with a meaningful contact frequency in LBR KO mESCs, but not in QKO mESCs, where the contact frequencies are always far lower than in WT cells.

4. The role of Lmna and LBR in driving the phenotype remains unclear. The authors attempted to re-express Lmna in the TKO cell line, but this experiment was unsuccessful. We would appreciate seeing these data and understanding why this approach was chosen instead of knocking out Lmna in the existing LBR KO cell line, which would more directly address this question.

- We have not found experiments, successful or unsuccessful, with re-expression of Lmna in TKO in the paper. However, ED.Fig.4F shows depletion of LBR in TKO, and results indicate the major role of LBR in chromatin tethering. Hence, generation of QKO. The exact mechanisms of chromatin tethering by LMNA and LBR, indeed, remain to be elucidated, but the phenomenon is clearly described.

5. The hypothesis that cell cycle dynamics prevent complete chromatin inversion in QKO is intriguing, but no supporting data are provided. If this hypothesis is retained, additional supporting evidence would be valuable; otherwise, it may be best to remove it.

- The hypothesis was proposed not by these authors but was first discussed in Solovei et al (2013) and Falk et al (2019), where authors claimed that in difference to postmitotic cells, quickly cycling cells cannot complete nuclear inversion.

6. Our previous concern regarding the underexploration of gene expression changes in TKO and QKO remains (see Major Point 3).

- In our opinion, the transcriptome analysis of mutants is sufficient.

7. Our previous comment on boxplots also still holds: it is difficult to determine whether effects are normally distributed or exhibit a different pattern (e.g., bimodal). A heatmap would be ideal, but a violin plot would also improve clarity.

- We cannot comment on this.

8. Similarly, the lack of analysis into upregulated genes remains an issue (see Major Point 3).

- As mentioned above, we do not think more deep analysis is needed.

9. Previous minor point 19 still holds true in our opinion: the kernel plot seems to support our

comment, showing higher counts of K9me2 across the genome. This would point to a higher background. We'd like the authors to respond to this.

- If the reviewer means Fig.5J, we do not see controversy here: the plot shows shift of H3K9me2 chromatin from the periphery in the interior although not all heterochromatin shifted.

10. A recent bioRxiv preprint from the Kuntay lab (December 2024) reports a similar phenotype in mESCs but highlights the essential role of Lap2 β and LBR in DNA-NL tethering. It would be valuable for the authors to discuss these findings and address any discrepancies between the two studies.

- This is a very good suggestion: the authors should cite Levis et al manuscript published recently on BioRxiv

Thank you for raising this point – we are aware of this exciting work. We now cite and briefly discuss this recent work in the Discussion.